# Duality and Policy Evaluation in Distributionally Robust Bayesian Diffusion Control

## Abstract

We consider a Bayesian diffusion control problem of expected terminal utility maximization. The controller imposes a prior distribution on the unknown drift of an underlying diffusion. The Bayesian optimal control, tracking the posterior distribution of the unknown drift, can be characterized explicitly. However, in practice, the prior will generally be incorrectly specified, and the degree of model misspecification can have a significant impact on policy performance. To mitigate this and reduce overpessimism, we introduce a distributionally robust Bayesian control (DRBC) formulation in which the controller plays a game against an adversary who selects a prior in divergence neighborhood of a baseline prior. The adversarial approach has been studied in economics (Hansen & Sargent, 2008) and efficient algorithms have been proposed in static optimization settings (Rahimian & Mehrotra, 2019). We develop a strong duality result for our DRBC formulation. Combining these results together with tools from stochastic analysis, we are able to derive a loss that can be efficiently trained (as we demonstrate in our numerical experiments) using a suitable neural network architecture. As a result, we obtain an effective algorithm for computing the DRBC optimal strategy. The methodology for computing the DRBC optimal strategy is greatly simplified, as we show, in the important case in which the adversary chooses a prior from a Kullback-Leibler distributional uncertainty set.

## 1 Introduction

Decision-making under uncertainty is a core challenge in reinforcement learning and control. In many practical settings, agents must act without knowing key environment parameters (e.g., transition dynamics, reward biases). We consider a diffusion control problem for which a controller aims at maximizing expected terminal utility by making decisions informed by observations. Since the controller cannot directly observe model parameters, it is natural to consider a Bayesian approach to learn while optimizing. Thus, the unknown parameter or factor is modeled as an unobservable random element with a prior distribution. This approach, known as Bayesian control, is well studied in the control literature and gives rise to sophisticated policies that naturally work well if the full Bayesian model is well specified.

However, if the model is not well specified, the Bayesian policy will often deliver suboptimal results. Adversarial approaches have been used to mitigate the impact of model misspecification. The controller interacts with a fictitious adversary to maximize value function, while the adversary selects a worst-case probability to ensure policy robustness. Distributionally robust control (DRC) (Hansen & Sargent, 2001; 2008) is one of such method, and it is built on finding robust formulations that are tractable in the sense of leading to a dynamic programming principle. In exchange of this type of tractability, the approach leads to very pessimistic policies, because the adversary's power is replenished at every point in time. This also makes calibrating the size of the distributional uncertainty difficult, because small variations on this parameter have a significant impact on performance.

To address over-conservatism, we consider a Distributionally Robust Bayesian Control (DRBC) formulation in which we only build distributional robustness around the prior distribution in the Bayesian control formulation. This allows us to combat pessimistic policies at the expense of loosing the dynamic programming principle, yet we need to develop alternative methods for computing the

optimal policy. Our motivating application is continuous-time control with unknown dynamics, but DRBC applies broadly to problems where robustness to prior misspecification is critical.

## 1.1 OUR CONTRIBUTIONS

Our contributions are summarized as follows.

- We formulate a DRBC problem for misspecified priors in the context of diffusion control using a $\phi$-divergence uncertainty set around the prior (which is imposed on the drift of the diffusion) (Section 2.2).

- We prove a strong duality result that reformulates DRBC into a tractable optimization problem. This connects our framework to smooth ambiguity models while making it directly applicable to machine learning settings (Theorem 2).

- We provide sample complexity results showing that policy evaluation is possible with the canonical rate $\mathcal{O}_p\left(n^{-1/2}\right)$ via a novel randomized multi-level Monte Carlo (rMLMC) unbiased estimator (Algorithm 2 and Theorem 13).

- We introduce a class of policies that contains the optimal policy for a large class of utility maximization problems of interest in the Bayesian control case and that are learnable efficiently via deep learning methods (Theorem 17). In the important special case of KL divergence, we find semi-closed form expressions to simplify the learning (Section 5.1).

- We present simulation results to demonstrate the accuracy of our deep learning method (Section 6.1) and confirm the $\mathcal{O}_p\left(n^{-1/2}\right)$ convergence rate (Section H.1). Numerical results show the robustness of DRBC and it overcomes overpessimism (Section H.2).

## 1.2 RELATED WORK

Distributionally Robust Methods have been well-studied in a wide range of areas. For example, Distributionally Robust Optimization (DRO) (i.e. the supervised learning case) can be shown to recover a wide range of successful statistical estimators (including sqrt-Lasso, AdaBoost, group Lasso etc.), by carefully choosing the uncertainty set, often in terms of $\phi$-divergence or Wasserstein sets (see Blanchet et al. (2021b; 2024)). Refer also to Rahimian & Mehrotra (2019); Bayraksan & Love (2015) for comprehensive reviews on DRO.

Motivated by problems in areas such as economics and finance, among others, (Hansen & Sargent, 2001; 2008; Denis & Kervarec, 2013; Bartl et al., 2021) , DRO has been generalized to the setting of dynamic decision-making with model uncertainty. This situation is significantly more complicated and the literature has focused mostly on developing formulations that are amenable to dynamic programming (DP), giving rise to Distributionally Robust Control (DRC) and Distributionally Robust Markov Decision Processes (DRMDP). The availability of a dynamic programming principle facilitates the development of Distributionally Robust Reinforcement Learning (DRRL) and related settings (Si et al., 2023; Wang & Zou, 2022; Wang et al., 2023a; Liu et al., 2022; Zhou et al., 2021; Lu et al., 2024).

However, to develop a DP in the DRC, DRMDP, and DRRL settings, the adversary gets its power replenished at every point in time, making the formulations pessimistic. That occurs at every point in time, thus making these formulations overconservative.

*In contrast, our formulation combines Bayesian stochastic control with DRO by introducing a single distributional uncertainty set in the prior distribution. This combats overconservative solutions, but at the expense of the DP. However, we develop a formulation and techniques that make the optimal solution learnable by exploiting continuous time stochastic analysis.*

Computations of static DRO problems has been studied by Levy et al. (2020); Blanchet & Kang (2020); Wang et al. (2021). Compared with us, they consider different uncertainty sets and the estimators are biased. We also mention the literature on RL in finance (Hambly et al., 2023), Bayesian Optimization (Daulton et al., 2022) and Distributionally Robust Bayesian Optimization (DRBO, (Kirschner et al., 2020)). Our setting is different from DRBO. We work in continuous time, which allows us to use stochastic analysis (via the martingale method and other techniques) to obtain convenient expressions to define a suitable loss for the optimal strategy. Our formulation is also offline

and not online as in DRBO. We wish to efficiently evaluate policies for the game corresponding to our formulation. In addition, the work on Bayesian Distributionally Robust Optimization (BDRO, Shapiro et al. (2023)) is more related to our setting here, but it focuses on a static setting.

## 2 SETTINGS AND FORMULATIONS

### 2.1 CLASSICAL BAYESIAN CASE

Let $(\Omega, \mathcal{F}, P)$ be a complete filtered probability space, and $W$ is a standard Brownian motion under $P$. $B : \Omega \to \mathbb{R}$ is a real-valued random variable such that $B$ and $W$ are independent. We denote $\sigma(B)$ as the $\sigma$-algebra generated by $B$ and assume that $B$ has a distribution $\mu \in \mathcal{P}(\mathbb{R})$ (the space of all Borel probability measures on $\mathbb{R}$). We have the risk-free asset $S_{0,0} = s_0 > 0$ and $dS_{0,t} = rS_{0,t}dt$, $0 \le t \le T$, with interest-free rate $r > 0$ and a risky asset $S$ with drift $B$

$$dS_t = S_t(Bdt + \sigma dW_t), \ \ 0 \le t \le T. \tag{1}$$

We next define filtrations: $\mathcal{F}^S$ is the natural filtration of $S$, and $\mathcal{G}$ is the union of $\sigma(B)$ and $\mathcal{F}^W$ (the natural filtration of $W$). We denote the $\mathcal{F}^S$-adapted (the decisions are made based on only the observations of the stock prices) stochastic process $\pi = \{\pi_t\}_{t \in [0,T]}$ as the amount of money invested in the risky asset. This induces the dynamics of a controlled wealth process with $X_0 = x_0$ (we simplify the notation so that $X^\pi$ is written as $X$)

$$dX_t = (X_t - \pi_t)r\,dt + \pi_t \left(Bdt + \sigma dW_t\right). \tag{2}$$

In this paper, control, policy, and strategy are used interchangeably. We call an $\mathcal{F}^S$-progressively measurable stochastic processes (control) $\pi = \{\pi_t\}_{t \in [0,T]}$ admissible if $X_0 = x_0$, $\int_0^T \|\pi_t\|_2^2 \, dt < \infty$, and Equation (2) admits a unique strong solution. The collection of all admissible controls is denoted as $\mathcal{A}(x_0)$. The Bayesian diffusion control problem is defined as

$$V(x_0) = \sup_{\pi \in \mathcal{A}(x_0)} E^P \left[u(X_T)\right], \tag{3}$$

where the utility function $u : (0, \infty) \to \mathbb{R}$ is strictly concave and strictly increasing. Without loss of generality, in this paper, we consider the utility function $u(x) = \frac{1}{\alpha}x^\alpha$ with $\alpha \in (0, 1)$ and $X_t > 0$ for any $t \in [0, T]$. The optimal solution of Problem (3) is given by Theorem 28 in Appendix B (Karatzas & Zhao, 1998). The takeaway is that the optimal solution $V(x_0)$ is a functional of the prior distribution $\mu$, and the optimal policy is a random variable depending on $\mu$ and the observations. In practice, the prior distribution is chosen by experts and other available information, and the fraction of investment into risky asset is computed via the formula provided by Theorem 28 with real observations.

### 2.2 AMBIGUITY SET FOR THE DISTRIBUTIONALLY ROBUSTNESS

Given a convex function $\phi : [0, \infty) \to \mathbb{R}$ with $\phi(1) = 0$, $\phi$-divergence of $Q$ from $P$ is $D_\phi(P\|Q) = \int_\Omega \phi\left(\frac{dP}{dQ}\right) dQ$, where $\frac{dP}{dQ}$ is the Radon-Nikodym derivative of $P$ with respect to $Q$. If $\phi(x) = x\log(x) - x + 1$, then $D_\phi$ is the Kullback–Leibler (KL) divergence, denoted as $D_{\text{KL}}$.

In the Bayesian problem (3), the imposed prior may not be exactly the same as the underlying drift. Distributionally Robust Control (DRC) methods relying on dynamic programming principles to mitigate this model misspecification (Hansen & Sargent, 2001) are often too pessimistic since in every step the worst case is chosen. Thus we consider a distributionally robust Bayesian control (DRBC) formulation where the controller engages in a game against an adversary who chooses a prior for the drift at the beginning from a $\phi$-divergence neighborhood (we call this ambiguity set or uncertainty set) around the baseline prior to overcome this overpessimism. To rigorously define the uncertainty set, we need to make sure only distribution of $B$ is changed and all other conditions (e.g. measurability) are kept the same. We denote $\mathcal{P}(\Omega, \mathcal{F})$ as the collection of all Borel probability measures on the measurable space $(\Omega, \mathcal{F})$. Then we define the following set ($B \perp W$ means that the two random variables $B$ and $W$ are independent)

$$\mathcal{Q}_\delta = \left\{ \begin{array}{l} Q \in \mathcal{P}(\Omega, \mathcal{F}), \\ Q \ll P \end{array} \middle| \begin{array}{l} Q(B \in A) = \nu(A) \text{ for some } \nu \in \mathcal{P}(\mathbb{R}), \forall A \in \mathcal{B}(\mathbb{R}), D_\phi(\nu\|\mu) \le \delta, \\ B \perp W, W \text{ is a standard Brownian motion under } Q. \end{array} \right\}.$$

The set $\mathcal{Q}_\delta$ is endowed with the topology of weak convergence (Billingsley, 1986). We define a relation on $\mathcal{Q}_\delta$:

$$Q_1 \sim Q_2 \iff \text{for any } A \in \mathcal{B}(\mathbb{R}), Q_1(B \in A) = Q_2(B \in A). \tag{4}$$

We use the quotient space $\mathcal{U}_\delta = \mathcal{Q}_\delta / \sim$ as the uncertainty set for the DRBC problem. The following theorem gives an intuition of this.

**Theorem 1.** $\mathcal{U}_\delta$ *is well-defined. For all* $Q \in \mathcal{U}_\delta$*, there exists* $\nu \ll \mu$ *such that* $\frac{dQ}{dP} = \frac{d\nu}{d\mu}(B)$ *$P$-almost surely and* $D_\phi(Q \parallel P) = D_\phi(\nu \parallel \mu)$.

With Theorem 1, we define the goal of the DRBC problem

$$\sup_{\pi \in \mathcal{A}(x_0)} \inf_{Q \in \mathcal{U}_\delta} E^Q[u(X_T)]. \tag{5}$$

We denote the optimal solution of Problem (5) as $\pi_{\text{DRBC}}$. Compared with the DRC approach, Problem (5) looses the tractability in terms of dynamic programming principle (see Appendix I.3.2), thus we need to develop another method to compute $\pi_{\text{DRBC}}$. The first step is to use duality to simplify the representation of the original problem. Note that for simplicity, all the settings above are one dimensional case. High dimensional case is a natural analogy and can be found in B.2.

## 3 STRONG DUALITY

We begin with some notations. Suppose $f : \mathbb{R} \to \mathbb{R}$ is a real-valued function, we denote $f^*$ as the convex conjugate of $f$, which is defined as $f^*(x^*) = \sup_{x \in X} (\langle x^*, x \rangle - f(x)) = -\inf_{x \in X} (f(x) - \langle x^*, x \rangle)$ and $\langle ., . \rangle$ is the standard inner product in $\mathbb{R}$ (Rockafellar, 1970). Moreover, there exists unique class of probability measures $\{P^b\}_{b \in \mathbb{R}} \subset \mathcal{P}(\Omega, \mathcal{F})$ such that for any $A \in \mathcal{F}, P(A) = \int_{\mathbb{R}} P^b(A) d\mu(b)$. We then prove an extension of the strong duality result in Shapiro (2017), which transforms the original infinite-dimensional problem to finite-dimensional.

**Theorem 2.**

$$\inf_{Q \in \mathcal{U}_\delta} E^Q[u(X_T)] = \sup_{\lambda \geq 0, \beta \in \mathbb{R}} \left\{ \beta - \lambda \delta + \int_{\mathbb{R}} \Phi_{\lambda,\beta} \left( E^{P^b}[u(X_T)] \right) d\mu(b) \right\},$$

*where for fixed* $\lambda \geq 0, \beta \in \mathbb{R}$,

$$\Phi_{\lambda,\beta}(x) := -(\lambda\phi)^*(\beta - x).$$

In general, as a function of $x$, $\Phi_{\lambda,\beta}$ is always concave due to the convexity of $\phi$, but it is not always increasing. As a function of $(\lambda, \beta)^T$, $\Phi_{\lambda,\beta}$ is always concave. For the convenience of the notations, we denote $Z^b := E^{P^b}[u(X_T)]$. If $\phi$ induces the KL divergence, then we denote the uncertainty set as $\mathcal{U}_{\text{KL},\delta}$ and have the following duality result.

**Theorem 3.**

$$\inf_{Q \in \mathcal{U}_{KL,\delta}} E^Q[u(X_T)] = \sup_{\lambda \geq 0} \left\{ -\lambda\delta - \lambda \log \left( \int_{\mathbb{R}} \exp \left( \frac{-Z^b}{\lambda} \right) d\mu(b) \right) \right\},$$

*where* $\lambda \mapsto -\lambda\delta - \lambda \log \left( \int_{\mathbb{R}} \exp \left( \frac{-Z^b}{\lambda} \right) d\mu(b) \right)$ *takes the value ess inf* $Z^B$ *when* $\lambda = 0$ *and ess inf denotes the essential infimum.*

The analysis of an extension which is called Cressie-Read divergence is provided in Appendix D.3. We give assumptions on the $\phi$-divergence to ensure the attainability of an optimal pair $(\lambda^*, \beta^*)^T = (\lambda^*(\pi), \beta^*(\pi))^T$ with $\lambda^*(\pi) > 0$ for a fixed $\pi \in \mathcal{A}(x_0)$. The precise discussion can be found in Appendix D.4. We notice that the duality can be further written as a univariate function $\Phi_\lambda$ or $\Phi_\beta$ in terms of the dual variable (as in Theorem 3). For the notational convenience, if $\Phi_\lambda$ or $\Phi_\beta$ is strictly concave in $\lambda$ or $\beta$, we still say $\Phi_{\lambda,\beta}$ is strictly concave. This will not affect the rate of the asymptotic analysis, and the impacts on explicit computation are discussed in Appendix D.4 and E.2.

**Assumption 4.** For a fixed control $\pi \in \mathcal{A}(x_0)$, the norm of the pair of the optimal multipliers $(\lambda^*, \beta^*)^T$ has finite upper and lower bounds. In particular, $\lambda^* \neq 0$.

**Assumption 5.** $\Phi_{\lambda,\beta}$ is strictly increasing and strictly concave in $x$, and it is strictly concave in $(\lambda, \beta)^T$. For any fixed $\lambda \geq 0$ and $\beta \in \mathbb{R}$, $\Phi_{\lambda,\beta}$ is continuously differentiable in a neighborhood of $Z^b$ $\mu$-almost surely, satisfies the linear growth condition, and $D\Phi_{\lambda,\beta}$ is locally Hölder continuous with parameter $w \geq 1$ and bounding constant $\mathcal{K}$.

**Assumption 6.** For any fixed $b \in \mathbb{R}$, the value $V_b(x_0) = \sup_{\pi \in \mathcal{A}(x_0)} E^{P^b}[u(X_T)] < \infty$. This assumption is standard in portfolio optimizations problem (Karatzas et al., 1991). Moreover, we denote for a fixed control $\pi \in \mathcal{A}(x_0)$, ess inf $X_T = m \geq 0$. We assume that $\Phi_{\lambda^*,\beta^*}(0) > 0$.

Both of the following assumptions will imply Assumption 4 in the KL divergence case.

**Assumption 7.** The prior $\mu$ is a light-tailed distribution such that $0 < M := \int_{\mathbb{R}} V_b(x_0)^{3(1+w)} d\mu(b) < \infty$.

**Assumption 8.** The density (or probability mass function) of $E^{P^B}[u(X_T)]$ has a uniform lower bound $b > 0$ and the domain of density (or probability mass function) is compact.

As mentioned in Si et al. (2023) and Duchi & Namkoong (2021), Assumption 4, 5, and 6 are satisfied with mild conditions for KL and Cressie-Read divergences. From strong duality, the general form of Problem (5) becomes

$$\sup_{\pi \in \mathcal{A}(x_0)} \inf_{Q \in \mathcal{U}_\delta} E^Q[u(X_T)] = \sup_{\pi \in \mathcal{A}(x_0)} \sup_{\substack{\lambda > 0 \\ \beta \in \mathbb{R}}} \left\{ \beta - \lambda\delta + \int_{\mathbb{R}} \Phi_{\lambda,\beta}(Z^b) d\mu(b) \right\}.$$

Solving Problem (5) is hard in general. We provide a heuristic example of closed-form computation in Appendix F.5 to illustrate this difficulty. In particular, the difficulties are two-fold: the double supremum is highly nonconvex and even doing policy learning with fixed $\lambda$ and $\beta$ is hard. Inspired by the typical choices in distributionally robust contextual bandit (Si et al., 2023), we use an alternative optimization algorithm to get the well-approximated optimal solution $\hat{\pi}_{\text{DRBC}} \approx \pi_{\text{DRBC}}$ (See Algorithm 3 for the KL case). We learn $\hat{\pi}_{\text{DRBC}}$ by first fixing $\pi$ and optimizing $(\lambda, \beta)^T$ (we call this step policy evaluation) and then by fixing $(\lambda, \beta)^T$ and optimizing $\pi$ (we call this step policy learning). We continue this alternative updating until the values of $(\lambda, \beta)^T$ converges. Simulation results show that this alternative iterative algorithm works well in practice (see Section 6).

## 4 POLICY EVALUATION STEP

We first fix a control $\pi \in \mathcal{A}(x_0)$ and aim to compute

$$Q_{\text{DRBC}}(\pi) = \sup_{\substack{\lambda > 0 \\ \beta \in \mathbb{R}}} \left\{ \beta - \lambda\delta + \int_{\mathbb{R}} \Phi_{\lambda,\beta}(Z^b) d\mu(b) \right\} \tag{6}$$

for the general case, while for the KL divergence case, we have the policy evaluation

$Q_{\text{DRBCKL}}(\pi) = \sup_{\lambda > 0} \left\{ -\lambda\delta - \lambda \log \left( \int_{\mathbb{R}} \exp \left( \frac{-Z^b}{\lambda} \right) d\mu(b) \right) \right\}$. The basic idea is to first derive an unbiased estimator for the nested expectations (the inner samples depend on the outer samples) $\int_{\mathbb{R}} \Phi_{\lambda,\beta}(Z^b) d\mu(b)$ and $\log \left( \int_{\mathbb{R}} \exp \left( \frac{-Z^b}{\lambda} \right) d\mu(b) \right)$ and then use the standard Newton-Raphson method or gradient descent methods to find the optimal $\lambda^*$ and $\beta^*$ (Theorem 37, 39).

We focus on the general case. Assume that for a fixed $\pi \in \mathcal{A}(x_0)$, we have the access to the simulator $\mathcal{S}$ which is able to generate samples from the distribution of $\mu$ and take one sample $b$ from $\mu$ as an input to generate unbiased samples from $Z^b$. This is a standard assumption (Syed & Wang, 2023). For the KL divergence case, see the discussion and Algorithm 1 in Appendix E.2.

We quickly review an important method that we will use in the approximation of the solution. Multilevel Monte Carlo (MLMC) methods are designed to reduce the total computational complexity in the Monte Carlo estimations (Giles, 2008; 2015). Rhee & Glynn (2015) proposes a randomized MLMC (rMLMC), which in addition produces unbiased estimates. The rMLMC estimator is also used to produce unbiased estimates of single-layer nested expectations and solutions of stochastic optimization problems (among other problems) (Blanchet & Glynn, 2015; Blanchet et al., 2019).

In our case, the randomized (rMLMC) estimator first samples $b \sim \mu$, then independently samples a random $N^b = \tilde{N}^b + n_0$, where $n_0$ is a fixed non-negative integer and $\tilde{N}^b \sim \text{Geo}(R)$ with $R \in \left(\frac{1}{2}, \frac{3}{4}\right)$, finally generates $2^{N^b+1}$ unbiased estimates $\left\{\hat{Z}_i^b\right\}_{1 \le i \le 2^{N^b+1}}$ of $Z^b = E^{P^b}[u(X_T)]$, and the random estimator of $\int_{\mathbb{R}} \Phi_{\lambda,\beta}\left(Z^b\right) d\mu(b)$ is given by $\mathcal{E}^b = \mathcal{E}_{\lambda,\beta}^b = \frac{\Delta_{N^b}^b}{p(N^b)} + \Phi_{\lambda,\beta}\left(\frac{S_{2^{n_0}}^b}{2^{n_0}}\right)$, where $p(.)$ is the probability mass function of $N^b$ and $S_l^b = \sum_{i=1}^{l} \hat{Z}_i^b$. For $N^b \ge n_0$, we define $\Delta_{N^b}^b = \Phi_{\lambda,\beta}\left(\frac{S_{2^{N^b+1}}^b}{2^{N^b+1}}\right) - \frac{1}{2}\left(\Phi_{\lambda,\beta}\left(\frac{S_{2^{N^b}}^{O,b}}{2^{N^b}}\right) + \Phi_{\lambda,\beta}\left(\frac{S_{2^{N^b}}^{E,b}}{2^{N^b}}\right)\right)$, where $S_l^{O,b} = \sum_{i=1}^{l} \hat{Z}_i^{O,b}$ and $S_l^{E,b} = \sum_{i=1}^{l} \hat{Z}_i^{E,b}$ with $\left\{\hat{Z}_i^{E,b}\right\}_{1 \le i \le 2^{N^b+1}}$ and $\left\{\hat{Z}_i^{O,b}\right\}_{1 \le i \le 2^{N^b+1}}$ denotes the estimates indexed by even and odd values, respectively. Our overall estimator $\mathcal{E}$ is (for fixed $\pi$ and $\delta$)

$$\mathcal{E} = \mathcal{E}_{\pi,\delta,\lambda,\beta} = \frac{1}{n}\sum_{i=1}^{n} \mathcal{E}^{b_i} = \frac{1}{n}\sum_{i=1}^{n}\left(\frac{\Delta_{N^{b_i}}^{b_i}}{p(N^{b_i})} + \Phi_{\lambda,\beta}\left(\frac{S_{2^{n_0}}^{b_i}}{2^{n_0}}\right)\right) \tag{7}$$

given $n$ i.i.d. samples $\{b_i\}_{1 \le i \le n}$ generated from $\mu$.

*Remark* 9. When $n_0 = 0$, then our estimator is exactly the same as the estimator in Blanchet & Glynn (2015) and Blanchet et al. (2019). Note that $N^{b_i}$ does not depend on $b_i$, and the notation represents the i.i.d. copies of the shifted geometric random variable. The reason to introduce the non-negative shift $n_0$ is that the estimator $\mathcal{E}$ loses the concavity due to the definition of $\Delta_{N^b}^b$, but if we make $n_0$ properly large, then $\Delta_{N^b}^b$ looks more like a concave function, thus help the numerical optimization steps in terms of $(\lambda, \beta)^T$.

We make the following assumption that will be important in the analysis of the variance of the estimator when the prior has a continuous density. The case when the prior is finitely supported can be analyzed similarly.

**Assumption 10.** The map $b \mapsto Z^b$ is continuously differentiable and injective on the support of $B$.

**Theorem 11.** *Suppose Assumption 5, 6 and 7 hold, then $\mathcal{E}_{\pi,\delta,\lambda,\beta}$ is an unbiased estimator for $\int_{\mathbb{R}} \Phi_{\lambda,\beta}\left(Z^b\right) d\mu(b)$. If Assumption 10 is satisfied, $\mu$ is compactly supported with a continuous density, then $\text{Var}\left(\mathcal{E}_{\lambda,\beta}^{b_1}\right)$ and $\text{Var}\left(\mathcal{E}_{\pi,\delta,\lambda,\beta}\right)$ are both finite.*

We define the DRBC policy evaluation estimator $\hat{Q}_{\text{DRBC}}(\pi) = \sup_{\substack{\lambda > 0 \\ \beta \in \mathbb{R}}} \{\beta - \lambda\delta + \mathcal{E}_{\pi,\delta,\lambda,\beta}\}$ (see Algorithm 2 in Appendix G.1). Note that this empirical version of optimization problem is no longer strictly concave in $(\lambda, \beta)^T$ for finite sample size (it is strictly concave for $n \to \infty$). It is natural to have the following assumption, thus have the following $\mathcal{O}_p\left(n^{-1/2}\right)$ convergence rate guarantee.

**Assumption 12.** For a fixed $\pi \in \mathcal{A}(x_0)$, with probability 1, $\arg\max \hat{Q}_{\text{DRBC}}(\pi)$ has the same bounds as the bounds for $\arg\max Q_{\text{DRBC}}(\pi)$ in Assumption 4.

**Theorem 13.** *Suppose Assumption 4, 5, 6, 7, 10, and 12 hold, $\mu$ is compactly supported with a continuous density. For fixed $\pi \in \mathcal{A}(x_0)$, let $n$ denote the number of i.i.d. samples of $\{E_{\lambda,\beta}^{b_i}\}_{1 \le i \le n}$, then*

$$\sqrt{n}\left(\hat{Q}_{DRBC}(\pi) - Q_{DRBC}(\pi)\right) \Rightarrow \mathcal{N}\left(0, \text{Var}\left(\mathcal{E}_{\lambda^*,\beta^*}^{b_1}\right)\right),$$

*where $\Rightarrow$ denotes convergence in distribution, $(\lambda^*, \beta^*)^T$ is defined in Section 3, and $\mathcal{N}(0, \sigma^2)$ represents a normal distribution with mean 0 and standard deviation $\sigma > 0$.*

## 5 POLICY LEARNING STEP

This section concentrates on the policy learning step. Section 5.1 focuses on the KL divergence case with a finitely-supported prior. In Section 5.2, we slightly modify the class of admissible controls and propose a general deep learning method, utilizing techniques from stochastic analysis. Interestingly, when the Lagrangian multipliers $\lambda$ and $\beta$ are fixed, the policy learning step is equivalent to solve a smooth ambiguity problem with functional parameters from the strong duality (Theorem 2).

## 5.1 A Finite Prior Example: Duality Theory

In the KL divergence case, if we fix $\lambda > 0$, then we essentially need to solve

$$\sup_{\pi \in \mathcal{A}(x_0)} \left\{ -\lambda \log \left( \int_{\mathbb{R}} \exp \left( \frac{-Z^b}{\lambda} \right) d\mu(b) \right) \right\}. \tag{8}$$

In fact, Problem (8) is equivalent to a problem in mathematical finance called the ambiguity aversion measured by the entropic risk measure (Schied, 2007). Because of the special form of the KL divergence, we obtain the duality form and thus a semi-closed form of the optimal solution. We define a class of equivalent probability measures $\mathcal{AC}$ on the set (quotient space w.r.t the weak topology)

$$\mathcal{AC}_{\text{pre}} = \left\{ \begin{array}{l} \tilde{Q} \in \mathcal{P}(\Omega, \mathcal{F}), \\ \tilde{Q} \ll P \end{array} \middle| \begin{array}{l} \tilde{Q}(B \in A) = \nu(A) \text{ for some } \nu \in \mathcal{P}(\mathbb{R}), \forall A \in \mathcal{B}(\mathbb{R}), \\ W \text{ is a standard Brownian motion under } \tilde{Q} \text{ and } B \perp W. \end{array} \right\}$$

with respect to the equivalent relation $\sim$ defined in Section 2.2 That is, $\mathcal{AC} = \mathcal{AC}_{\text{pre}}/\sim$. Then obviously, $\mathcal{U}_\delta \subset \mathcal{AC}$.

**Lemma 14.** *For all $\tilde{Q} \in \mathcal{AC}$, there exists $\nu \ll \mu$ such that $\frac{d\tilde{Q}}{dP} = \frac{d\nu}{d\mu}(B)$ $P$-almost surely and $D_\phi \left( \tilde{Q} \parallel P \right) = D_\phi \left( \nu \parallel \mu \right)$.*

**Theorem 15.**

$$\sup_{\pi \in \mathcal{A}(x_0)} \left\{ -\lambda \log \left( \int_{\mathbb{R}} \exp \left( \frac{-Z^b}{\lambda} \right) d\mu(b) \right) \right\} = \sup_{\pi \in \mathcal{A}(x_0)} \inf_{\tilde{Q} \in \mathcal{AC}} \left\{ E^{\tilde{Q}} \left[ u(X_T) \right] + \lambda D_{KL} \left( \tilde{Q} \parallel P \right) \right\}.$$

Recall that for a fixed $\tilde{Q} \in \mathcal{AC}$, the problem becomes Problem (3), thus it would be easier to solve the problem if we could interchange the sup and inf. In order to achieve this, we would like to apply Sion's min-max theorem (see Theorem 27 in Appendix A) to the functional $f(\tilde{Q}, \pi) := E^{\tilde{Q}} \left[ u(X_T) \right] + \lambda D_{KL} \left( \tilde{Q} \parallel P \right)$, which is defined on $\mathcal{AC} \times \mathcal{A}(x_0) \subset \mathcal{P}(\mathbb{R}) \times \mathcal{A}(x_0)$.

**Theorem 16.** *Assume that $\mu$ is finitely supported, $m = \text{ess inf } X_T$ exists, $P(X_T = m) > 0$, then there exists an optimal pair $\left( \tilde{Q}^*, \pi^* \right)$ such that $\pi^*$ is the optimal solution for Problem (3) with underlying probability measure $\tilde{Q}^*$ (see Theorem 28) and*

$$\sup_{\pi \in \mathcal{A}(x_0)} \left\{ -\lambda \log \left( \int_{\mathbb{R}} \exp \left( \frac{-Z^b}{\lambda} \right) d\mu(b) \right) \right\} = \sup_{\pi \in \mathcal{A}(x_0)} \inf_{\tilde{Q} \in \mathcal{AC}} \left\{ E^{\tilde{Q}} \left[ u(X_T^\pi) \right] + \lambda D_{KL} \left( \tilde{Q} \parallel P \right) \right\}$$

$$= \inf_{\tilde{Q} \in \mathcal{AC}} \sup_{\pi \in \mathcal{A}(x_0)} \left\{ E^{\tilde{Q}} \left[ u(X_T^\pi) \right] + \lambda D_{KL} \left( \tilde{Q} \parallel P \right) \right\} = f \left( \tilde{Q}^*, \pi^* \right).$$

We focus on a commonly used example when the prior distribution takes value on a finite set of points $\{b_1, \ldots, b_d\}$ with probability mass function $P(B = b_i) = p_i \in (0, 1)$, for $i = 1, 2, \ldots, d$, where $d \geq 1$. For the fixed prior $\mu$, the optimal value function becomes $\tilde{V}(\boldsymbol{p}) := \frac{(x_0 e^{rT})^\alpha}{\alpha} \left( \int_{\mathbb{R}} \left( \tilde{F}_{\boldsymbol{p}}(T, z) \right)^{\frac{1}{1-\alpha}} \varphi_T(z) dz \right)^{1-\alpha}$, where $\tilde{F}_{\boldsymbol{p}}(t, z) = \sum_{i=1}^d p_i L_t(b_i, z)$. If we want to find the optimal probability measure $\tilde{Q}^*(\lambda)$, then it suffices to solve the convex problem

$$\inf_{\substack{\boldsymbol{q}=(q_1,\ldots,q_d) \\ \sum_{i=1}^d q_i=1, \text{and } q_i \geq 0}} V(\boldsymbol{q}) = \inf_{\substack{\boldsymbol{q}=(q_1,\ldots,q_d) \\ \sum_{i=1}^d q_i=1, \text{and } q_i \geq 0}} \tilde{V}(\boldsymbol{q}) + \lambda \sum_{i=1}^d q_i \log \frac{q_i}{p_i}. \tag{9}$$

The DRBC algorithm to iteratively solve Problem (5) (Algorithm 3) (we use (stochastic) gradient descent here) can be found in Appendix G.2. The precise algorithm for updating $\pi$ is given in Algorithm 4 in Appendix G. We remark that the DRBC learning steps are done via simulated (training) samples $\{S_t\}_{t \in [0,T]}$ before we observe the real market data $(\{\tilde{S}_t\}_{t \in [0,T]})$. In other words, we derive optimal policies of the form in Theorem 28 with a worst-case probability, and $\hat{\pi}_{\text{DRBCKL}}$ is computed by this worst-case probability and $\{\tilde{S}_t\}_{t \in [0,T]}$. Note Algorithm 4 is just an example for low dimensional case, see Section H.3 and Appendix B for high dimensional discussions.

### 5.2 GENERAL CASE: DEEP LEARNING APPROACH

In this section, we consider the case for the general $\phi$-divergence, we solve for fixed $(\lambda, \beta)^T$,

$$\sup_{\pi \in \mathcal{A}(x_0)} \int_{\mathbb{R}} \Phi_{\lambda, \beta} \left( E^{P^b} [u(X_T)] \right) d\mu(b). \tag{10}$$

Under Assumption 5, $\Phi_{\lambda, \beta}$ is strictly concave and strictly increasing in $x$, thus Problem (10) is a continuous-time smooth ambiguity problem (Klibanoff et al., 2005). Such problems can be solved recursively in discrete-time (Klibanoff et al., 2009), but in continuous-time only few blue cases can be solved in closed-form, and there is few literature in numerical methods for this with generality.

We utilize tools from stochastic calculus and results from Guan et al. (2022) to design a deep learning method to solve Problem (10). This motivates an alternate definition of the set of admissible controls $\tilde{\mathcal{A}}(x_0)$ that contains the optimal policy for a large class of Bayesian control problems (see Appendix F.4): the collection of all alternate admissible controls $\pi$, which is a subset of $\mathcal{A}(x_0)$ such that there exists a function $v \sim u$ such that there exists corresponding function $h : \mathbb{R} \to \mathbb{R}$ and a functional $\rho : L^1 \to \mathbb{R}$ such that for the controlled terminal wealth $X_T^\pi$, $\int_{\mathbb{R}} E^{P^b} [v(X_T^{b*})] \lambda(b) d\mu(b) = h(x_0)\rho(\lambda)$ and $h(1) = v(e^{rT})$. For a fixed $b \in \mathbb{R}$, define $\eta_t^{b*} = \exp\left(-\frac{B-b}{\sigma} W_t - \frac{(B-b)^2}{2\sigma^2} t\right)$, $\left.\frac{dQ^b}{dP}\right|_{\mathcal{F}_t^S} = \eta_t^{b*}$, and $W_t^b = W_t + \frac{B-b}{\sigma} t$ for $t \in [0, T]$, then under $Q^b$, the process $W^b$ is an $\mathcal{F}^S$-Brownian motion that is independent from $B$, hence under $Q^b$, the stock price $S$ evolves as $dS_t = S_t \left(bdt + \sigma dW_t^b\right)$. Moreover, $dX_t = (X_t - \pi_t)r \, dt + \pi_t \left(bdt + \sigma dW_t^b\right)$. For a fixed $b \in \mathbb{R}$, $E^{P^b} [u(X_T)] = E^{Q^b} [u(X_T)]$, therefore it is equivalent to study the problem $\sup_{\pi \in \tilde{\mathcal{A}}(x_0)} \int_{\mathbb{R}} \Phi_{\lambda, \beta} \left( E^{Q^b} [u(X_T)] \right) d\mu(b)$, which is solved in Guan et al. (2022). Let $b_0 = E^P[B]$ and if $b = b_0$, then we denote $W^{b_0}$ as $\hat{W}$. We also define for $t \in [0, T]$, $\eta_t = \exp\left(-\nu \hat{W}_t - \frac{1}{2}\nu^2 t\right)$, where $\nu = \frac{b_0 - r}{\sigma}$. For a fixed $b \in \mathbb{R}$, we define $t \in [0, T]$, $\eta_t^b = \exp\left(-\nu_b \hat{W}_t - \frac{1}{2}\nu_b^2 t\right)$, where $\nu_b = \frac{b_0 - b}{\sigma}$. The families of measures $\{P^b\}_{b \in \mathbb{R}}$ and $\{Q^b\}_{b \in \mathbb{R}}$ are quite different. For example, the distribution of $B$ under $Q^b$ is still $\mu$ (see Appendix F.6), where under $P^b$, $B$ is a constant $b$. In the rest of this section, we assume that $\mu \sim \mathcal{N}\left(\mu_0, \sigma_0^2\right)$. Now, we are able to derive sufficient conditions that an optimal terminal wealth satisfies. Theorem 17 motivates a loss function that also ensures numerical stability (Appendix G.5).

**Theorem 17.** *If Assumption 5 holds, $X_T$ is a terminal wealth such that there exists a constant $\kappa \in \mathbb{R}$ with $E^{Q^r}[X_T] = x_0 e^{rT}$ and $L_\kappa(X_T) = 0$ with*

$$L_\kappa(X_T) = \kappa - \log\left(u'(X_T)\right) - \frac{K_2^2}{4K_1} + \log\left(\frac{\sigma_0}{\sigma_Q}\right) + \frac{r - b_0}{\sigma}\left(W_T + \frac{B - b_0}{\sigma} T\right) - \frac{(b_0 - r)^2}{2\sigma^2} T$$

$$+ K_3 + \log\left(\int_{\mathbb{R}} \Phi'_{\lambda, \beta}\left(E^{Q^b}[u(X_T)]\right) d\mu(b)\right) - \log\left(\int_{\mathbb{R}} \Phi'_{\lambda, \beta}\left(E^{Q^a}[u(X_T)]\right) d\mu_A(a)\right)$$

$D = W_T + \frac{B - b_0}{\sigma} T$, $K_1 = \frac{T}{2\sigma^2} + \frac{1}{2\sigma_0^2}$, $K_2 = \frac{Tb_0}{\sigma^2} + \frac{D}{\sigma} + \frac{\mu_0}{\sigma_0^2}$, $K_3 = \frac{Tb_0^2}{2\sigma^2} + \frac{b_0 D}{\sigma} + \frac{\mu_0^2}{2\sigma_0^2}$,

$\mu_A \sim \mathcal{N}\left(\mu_Q, \sigma_Q^2\right)$, $\sigma_Q^2 = \left(\frac{T}{\sigma^2} + \frac{1}{\sigma_0^2}\right)^{-1}$, *and* $\mu_Q = \sigma_Q^2\left(\frac{W_T}{\sigma} + \frac{BT}{\sigma^2} + \frac{\mu_0}{\sigma_0^2}\right)$, *then $X_T$ is the optimal terminal wealth for Problem (10).*

From Theorem 17, we guess that the optimal terminal wealth has the form $X_T^* = h(W_T, B)$, where $h$ is a function that we plan to use neural network $h_\theta$ to approximate ($\theta \in \mathbb{R}^d$). $\kappa \in \mathbb{R}$ is a learnable scalar. We replace $X_T$ in Theorem 17 by $h_\theta(W_T, B)$, denote all the learnable parameters as $\boldsymbol{\theta} = (\theta, \kappa)^T \in \mathbb{R}^{d+1}$, and design the loss function as for a choice of $b_1 \in \mathbb{R}$ (see Appendix G.5),

$$\mathcal{L}(\boldsymbol{\theta}) = E^{Q^{b_1}}\left[\left\|L_\kappa(h_\theta(W_T, B)))\right\|_2^2\right] + \left(E^{Q^r}[h_\theta(W_T, B)] - x_0 e^{rT}\right)^2. \tag{11}$$

If $X_T^* = h_\theta(W_T, B)$, then $\mathcal{L}(\boldsymbol{\theta}) = 0$. The Algorithm 6 (Appendix G.5) can be used to minimize $\mathcal{L}(\boldsymbol{\theta})$ to find the optimal numerical solution $\boldsymbol{\theta}^*$. Discussion of DRBC algorithm in general case is in Appendix G.6. Usage of $h_\theta$ enables scaling with dimension $n$, for simplicity we don't show it here.

Table 1: Comparisons of learning results and closed-form solutions. Here $b_1 = 0.1$, $b_2 = 0.3$; $r_1 = 0.05$, $r_2 = 0.1$. We run evaluations 100 times.

| COMPARING TERM | LEARNING RESULT | CLOSED-FORM |
|---|---|---|
| $E^{Q^{b_1}}[u(X_T^*)], r_1$ | $3.174 \pm 0.013$ | $3.226$ |
| $E^{Q^{b_2}}[u(X_T^*)], r_1$ | $3.460 \pm 0.030$ | $3.380$ |
| $E^{Q^{b_1}}[u(X_T^*)], r_2$ | $3.179 \pm 0.014$ | $3.267$ |

Table 2: Comparison of mean Sharpe Ratios across methods for part of $S\&P$ 500 data.

| METHOD | MEAN OF SHARPE RATIO ($\uparrow$) |
|---|---|
| MERTON | $0.015 \pm 0.301$ |
| BAYESIAN WITH NO AMBIGUITY (PRIOR 1) | $0.493 \pm 0.281$ |
| BAYESIAN WITH NO AMBIGUITY (PRIOR 2) | $0.655 \pm 0.282$ |
| DRC (PRIOR 1) | $-0.220 \pm 0.292$ |
| DRC (PRIOR 2) | $-0.237 \pm 0.293$ |
| DRBC (PRIOR 1) | $\mathbf{0.818 \pm 0.308}$ |
| DRBC (PRIOR 2) | $\mathbf{1.147 \pm 0.311}$ |

## 6 NUMERICAL EXPERIMENTS

In this section, we present two numerical experiments to illustrate our theoretical findings. In Section 6.1, we compare the performance of the neural network approach with the closed-form solution. In Section 6.2, we apply our method and baselines to real stock data and use Sharpe Ratio as the evaluation metric to compare the performances. More details on the implementation, choices of performance measures and parameter settings are in Appendix I. Additional experiments on validation the rate of convergence and comparisons between DRBC and baseline methods using simulated data with user specified finite prior and KL uncertainty set and high dimensional case are provided in Appendix H. We find all experiment results align with our theoretical arguments in previous sections.

### 6.1 COMPARE WITH CLOSED-FORM SOLUTIONS

If we replace $\Phi_{\lambda,\beta}$ with a power function $\Phi$, Problem (10) admits a closed-form solution (Guan et al., 2022). This allows us to explicitly evaluate the performance of Algorithm 6 in this specific scenario. In Table 1, we compare the closed-form optimal value $E^{Q^{b_1}}[u(X_T^*)]$ with the learned optimal value $E^{Q^{b_1}}[u(h_{\theta^*}(W_T, B))]$ across various market parameters $r$ and different values of $b_1$. To estimate the learned optimal value $E^{Q^{b_1}}[u(h_{\theta^*}(W_T, B))]$, we employ Monte Carlo approach by conducting 100 independent experiments, each utilizing 2000 samples of the pair $(W_T, B)$.

### 6.2 REAL DATA EXPERIMENTS

This experiment is motivated by Blanchet et al. (2021a). We use $S\&P$ 500 constituents data from 2015 to 2024 and evaluate different methods using average annualized Sharpe Ratio for all stocks. We use a rolling window of one year to get the required parameters for all methods like interest rate $r$ and an estimation of $\sigma$. For the ease of computation, we choose two fixed priors across time based on Wang & Zhou (2020), prior 1 is more deviated and prior 2 is less deviated. The uncertainty set radius $\delta$ is chosen following the cross-validation type in Si et al. (2023). Results in Table 2 show that the DRBC is substantially better than benchmarks and it reduces the overpessimism in real data.

## 7 CONCLUSION AND FUTURE WORK

We provided the DRBC model to mitigate misspecification and overpessimism. Though effecitve in the $\phi$-divergence uncertainty case, we believe that other efficient numerical methods under different unccertainty measure can be found to reduce the overpessimism. We leave them for future work.

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

APPENDIX

# A    PRELIMINARIES

In this section, we present preliminary definitions and results that will be used for the proofs.

## A.1    PRELIMINARY DEFINITIONS

**Definition 18.** The *Cressie-Read divergence* is a $\phi$- divergence where the convex function is taken by for a fixed $k > 1$,

$$f_k(t) = \frac{t^k - kt + k - 1}{k(k - 1)}.$$

**Definition 19.** Suppose $g : \mathbb{R}^d \to \mathbb{R}$ is a function, then we say $g$ satisfies the *linear growth condition* if there exists a constant $c > 0$ such that

$$|g(x)| \leq c \left( 1 + \|x\| \right),$$

where $\|.\|$ denotes a norm in the Euclidean space.

**Definition 20.** Suppose $f : D \subset \mathbb{R}^d \to \mathbb{R}$ is a function, then we say $f$ is *Hölder continuous* with parameter $w$ and bounding constant $K$ if there exists $K > 0$ and $w > 0$ such that for any $x, y \in D$,

$$|f(x) - f(y)| \leq K \|x - y\|^w. \tag{12}$$

We say $f$ is *locally Hölder continuous* with parameter $w$ and bounding constant $K$ if Equation (12) holds inside each compact neighborhood.

**Definition 21.** Let $B_1$ and $B_2$ be Banach spaces and $G : B_1 \to B_2$ be a mapping. It is said that $G$ is directionally differentiable at a considered point $\mu \in B_1$ if the limits

$$G'_\mu(d) = \lim_{t \downarrow 0} \frac{G(\mu + td) - G(\mu)}{t}$$

exist for all $d \in B_1$.

Furthermore, it is said that $G$ is *Gâteaux directionally differentiable* at $\mu$ if the directional derivative $G'_\mu(d)$ exists for all $d \in B_1$ and $G'_\mu(d)$ is linear and continuous in $d$. For ease of notation, we also denote $D_\mu(\mu_0)$ the operator $G'_\mu(\cdot)$.

Finally, it is said that $G$ is *Hadamard directionally differentiable* at $\mu$ if the directional derivative $G'_\mu(d)$ exists for all $d \in B_1$ and

$$G'_\mu(d) = \lim_{t \downarrow 0} \frac{G(\mu + td') - G(\mu)}{t}, \quad d' \to d.$$

## A.2    AUXILIARY RESULTS

**Theorem 22.** *(Abstract Bayes Theorem (Elliott et al., 1995)) Suppose $(\Omega, \mathcal{F}, P)$ is a probability space and $\mathcal{G} \subseteq \mathcal{F}$ is a sub-$\sigma$-field. Suppose $\bar{P}$ is another probability measure absolutely continuous with respect to $P$ and with Radon-Nikodym derivative $\frac{d\bar{P}}{dP} = \Lambda$. Then if $\phi$ is any $\bar{P}$-integrable random variable*

$$E^{\bar{P}}[\phi \mid \mathcal{G}] = \psi \quad \text{where} \quad \psi = \frac{E^P[\Lambda \phi \mid \mathcal{G}]}{E^P[\Lambda \mid \mathcal{G}]} \quad \text{if } E^P[\Lambda \mid \mathcal{G}] > 0$$

*and $\psi = 0$ otherwise.*

**Theorem 23.** *(Syed & Wang (2023)) Let $(Z_1, Z_2)$ be a 2-stage stochastic process and there exists $p \geq 1$, such that $E[|Z_2|^p] < \infty$. Conditioning on $Z_1$, sample i.i.d. $Z_2(1), \ldots, Z_2(n)$. Then there exists a constant $B_p$ depending only on $p$ such that*

$$E\left[ \left| \frac{1}{n} \sum_{i=1}^{n} Z_2(i) - E[Z_2 \mid Z_1] \right|^p \right] \leq \begin{cases} B_p \frac{E[|Z_2|^p]}{n^{p/2}} & p > 2, \\ B_p \frac{E[|Z_2|^p]}{n^{p-1}} & 1 \leq p \leq 2. \end{cases}$$

**Theorem 24.** *(Danskin Theorem (Bonnans & Shapiro, 2013)) Let $\Theta \subset \mathbb{R}^d$ be a nonempty compact set and $B$ be a Banach space. Suppose the mapping $G : B \times \Theta \to \mathbb{R}$ satisfies that $G(\mu, \theta)$ and $D_\mu(\mu, \theta)$ are continuous on $O_{\mu_0} \times \Theta$, where $O_{\mu_0} \subset B$ is a neighborhood around $\mu_0$. Let $\phi : B \to \mathbb{R}$ be the inf-functional*

$$\phi(\mu) = \inf_{\theta \in \Theta} G(\mu, \theta)$$

*and $\bar{\Theta}(\mu) = \arg\max_{\theta \in \Theta} G(\mu, \theta)$. Then, the functional $\phi$ is directionally differentiable at $\mu_0$ and*

$$G'_{\mu_0}(d) = \inf_{\theta \in \bar{\Theta}(\mu_0)} D_\mu(\mu_0, \theta) d.$$

**Theorem 25.** *(Delta Theorem (Shapiro et al., 2009)) Let $B_1$ and $B_2$ be Banach spaces, equipped with their Borel $\sigma$-algebras, $Y_N$ be a sequence of random elements of $B_1$, $G : B_1 \to B_2$ be a mapping, and $\tau_N$ be a sequence of positive numbers tending to infinity as $N \to \infty$. Suppose that the space $B_1$ is separable, the mapping $G$ is Hadamard directionally differentiable at a point $\mu \in B_1$, and the sequence $X_N = \tau_N(Y_N - \mu)$ converges in distribution to a random element $Y$ of $B_1$. Then,*

$$\tau_N\big(G(Y_N) - G(\mu)\big) \Rightarrow G'_\mu(Y),$$

*and*

$$\tau_N\big(G(Y_N) - G(\mu)\big) = G'_\mu(X_N) + o_p(1).$$

**Theorem 26.** *((Shapiro et al., 2009)) Let $B_1$ and $B_2$ be two Banach spaces, $G : B_1 \to B_2$ and $\mu \in B_1$. Then the following statements are true:*

- *(1) If $G$ is Hadamard directionally differentiable at $\mu$, then the directional derivative $G'_\mu$ is continuous.*

- *(2) If $G$ is Lipschitz continuous in a neighborhood of $\mu$ and directionally differentiable at $\mu$, then $G$ is Hadamard directionally differentiable at $\mu$.*

**Theorem 27.** *(Sion's minmax Theorem (Sion, 1958)) Let $M$ be any convex topological space and $O$ is a compact and convex space, $h : M \times O \to \mathbb{R}$ is a function such that*

- *(1) For any fixed $x \in M$, $h(x, .)$ is lower semi-continuous and quasi-convex.*

- *(2) For any fixed $y \in O$, $h(., y)$ is upper semi-continuous and quasi-concave.*

*Then*

$$\sup_{x \in M} \inf_{y \in O} h(x, y) = \inf_{y \in O} \sup_{x \in M} h(x, y).$$

# B  REVIEW OF BAYESIAN AND MARTINGALE METHODS

## B.1  MARTINGALE METHOD REVIEW

In the classical literature (Merton, 1971; Cox & Huang, 1991; Karatzas et al., 1991), $B$ is not a random variable but is a fixed real number. This problem is called Merton's problem in financial literature. In this case, we assume the controls are all $\mathcal{F}$-adapted. There are two ways to solve Problem (3) in this case: dynamic programming and martingale method.

The dynamic method is to guess a Hamilton–Jacobi–Bellman equation (HJB) equation (which is typically a nonlinear partial differential equation in the portfolio optimization problems) that the optimal control may satisfy, and then we use a verification method to show that solution (with some regularity) of this HJB equation indeed gets the optimal control. The final step is to either solve the HJB equation in closed-form or numerically. The dynamic programming method works well in the time-consistent case, but not well in the time-inconsistent case (See, for example, a discussion in He & Zhou (2016)). Thus, in this review section, we focus on another method. We remark that the traditional distributionally robust control are done via a dynamic programming approach, which we will revisit in Appendix I.3.2.

On the other hand, the martingale and duality method, which we will call martingale method later, is a more probabilistic approach and works well for portfolio optimization problems in both time consistent and inconsistent cases (notice that this is only for the financial problems which allows

replication of the portfolios (Björk, 2009)). Thus, we focus on the martingale method in this section now.

Recall that we want to solve

$$V(x_0) = \sup_{\pi \in \mathcal{A}(x_0)} E^P [u(X_T)],$$

where $\mathcal{A}(x_0)$ is the collection of all $\mathcal{F}$-progressively measurable stochastic processes $\pi = \{\pi_t\}_{t \in [0,T]}$ such that $\int_0^T \|\pi_t\|_2^2 dt < \infty$ and Equation (2) with $B$ being a fixed constant admits a unique strong solution. The hardness of this problem is that the optimal control is indeed a stochastic process, which is infinite-dimensional. A change of measure argument will convert this problem into one-dimensional (i.e. with respect to a single random variable rather than a stochastic process).

Consider the set of all portfolios that can be generated starting with an initial capital of $x$. By design, our final wealth $X_T$ lies within this set, making it a suitable search domain for optimization with respect to $X_T$. Since we only have one risky asset, then the market is complete, and according to the fundamental theorem of asset pricing (essentially Girsanov's theorem), (Karatzas & Shreve, 1998) there exists a unique probability measure $Q$ that is equivalent to $P$ such that under $Q$, the discounted processes $e^{-rt}S_t$ and $e^{-rt}X_t$ are martingales. Thus the original problem becomes

$$V(x_0) = \sup_{X_T \in \mathcal{A}(L^2)} E^P [u(X_T)],$$

where $\mathcal{A}(L^2)$ is the collection of $L^2$ random variables such that $e^{-rt}X_t$ is a martingale, which is equivalent to

$$E^Q \left[ e^{-rT} X_T \right] = x_0 = E^P \left[ \frac{dQ}{dP} e^{-rT} X_T \right] = E^P [\rho_T X_T],$$

where $\rho_t = \exp\left( - \left( r + \frac{\theta^2}{2} - \theta W_t \right) \right)$ and $\theta = \frac{B-r}{\sigma}$ is called the market price of risk. (For the computation of $\frac{dQ}{dP}$, see Karatzas et al. (1991)). Thus, writing down the Lagrangian duality of the new problem and using the point-wise optimization technique (Liang & Liu, 2019), we derive the optimal terminal wealth $X_T^*$.

Next we use the martingale property to derive the optimal wealth process by computing

$$X_t^* = \frac{1}{\rho_t} E^P [\rho_T X_T^* | \mathcal{F}_t].$$

Finally, from the martingale representation theorem (Øksendal, 2003) (since $\rho_t X_t$ is a martingale under $P$), there exists an $\mathcal{F}$-adapted process $\{\phi_t\}_{t \in [0,T]}$ such that

$$\rho_t X_t^* = x_0 + \int_0^t \phi_s dW_s,$$

where $W_t$ is the $P$-Brownian motion driving the dynamics of the risky asset. Applying Itô's lemma to $X_t^*$, we can match the diffusion term $\pi_t \sigma dW_t$ in the wealth equation to the diffusion term $\phi_t dW_t$. This gives

$$\pi_t^* = \frac{\phi_t}{\sigma}.$$

Thus, the optimal control $\pi_t^*$ is directly expressed in terms of the martingale representation process $\phi_t$, which is determined by the terminal condition $X_T^*$.

There are two takeaways from this method. Firstly, as long as we can construct the appropriate change of measure and the duality theory (in the complete market), then solving for the optimal control is equivalent to solving for the optimal terminal wealth. Secondly, the explicit computation based on the martingale property and Itô's lemma depends highly on the Gaussian (Brownian motion) assumptions, thus this may not generalize to other cases (without specific distributions).

### B.2 BAYESIAN TECHNIQUE REVIEW AND PROOF OF THEOREM 28

In practice, to use Merton's model to guide the investment, constants $B$ and $\sigma$ need to be estimated from the market data, then the model can be fitted with these estimators. However, in practice, even

though the estimation of $\sigma$ is relatively easy, the estimation of $B$ as a constant is hard (Mehra & Prescott, 1985). Thus, it is natural to consider the Bayesian technique to randomized $B$ as a random variable that is independent of the Brownian motion $W$, and the prior distribution of $B$ can be chosen via experts' advice. This is why this stochastic diffusion control problem is called Bayesian. In this case, martingale method combined with Bayesian and filtering techniques can be used to derive the closed-form solution, where for the dynamic programming approach we still need to solve a second-order nonlinear partial differential equation (Bismuth et al., 2019; Rieder & Bäuerle, 2005).

Karatzas & Zhao (1998) builds the solvability of Problem (3), but the prior distribution is given to the random market price of risk $\Theta = \frac{B-r}{\sigma}$, which is equivalent to our setting theoretically, but the exact closed form solution will be slightly different. In this section, we adapt the theory in Karatzas & Zhao (1998) to give the optimal solution of Problem (3) in Theorem 28.

**Theorem 28.** *The optimal value function of Problem (3) is given by* $V(x_0) = \frac{(x_0 e^{rT})^\alpha}{\alpha} \left( \int_\mathbb{R} (F(T, z))^{\frac{1}{1-\alpha}} \varphi_T(z) dz \right)^{1-\alpha}$, *and the optimal fraction invested in the stock for each time $t \geq 0$ is given by*

$$\frac{\pi_t^*}{X_t^*} = \frac{\int_\mathbb{R} \nabla F(T, z + Y_t)(F(T, z + Y_t))^{\frac{\alpha}{1-\alpha}} \varphi_{T-t}(z) dz}{(1-\alpha)\sigma \int_\mathbb{R} (F(T, z + Y_t))^{\frac{1}{1-\alpha}} \varphi_{T-t}(z) dz},$$

*where $\varphi_T$ is the density function of $\mathcal{N}(0, T)$, $Y_t = \frac{B-r}{\sigma}t + W_t$, and $F(t, y) := F_\mu(t, y) = \int_\mathbb{R} L_t(z, y) d\mu(z)$ with $L_t(z, y) = 1$ if $t = 0$ and $L_t(z, y) = \exp\left(\frac{z-r}{\sigma}y - \frac{1}{2}\left(\frac{z-r}{\sigma}\right)^2 t\right)$ if $t > 0$. Moreover, the filtration generated by the process $\{Y_t\}_{t \in [0, T]}$ is the same as $\mathcal{F}^S$.*

*Proof.* To begin with, from Girsanov's theorem (Karatzas & Shreve, 1991),

$$\frac{1}{Z_t} = L_t^{-1}(B, Y_t) = \exp\left(-\frac{B-r}{\sigma}Y_t + \frac{1}{2}\left(\frac{B-r}{\sigma}\right)^2 t\right)$$

is a $\mathcal{G}$-martingale under $P$ (hence also $\mathcal{F}^S$-martingale under $P$). If we define a probability measure $\tilde{P}$ by $\frac{d\tilde{P}}{dP} = \frac{1}{Z_T}$, then under $\tilde{P}$, $B$ still has the distribution $\mu$ and is independent of the Brownian motion $Y$.

Next, from the tower property, for fixed $t \in [0, T]$,

$$\hat{Z}_t := \tilde{E}\left[Z_T | \mathcal{F}_t^S\right] = \tilde{E}\left[\tilde{E}\left[Z_T | \mathcal{G}_t\right] | \mathcal{F}_t^S\right] = F(t, Y_t).$$

Moreover, from the abstract Bayes' rule (Elliott et al., 1995), the conditional expectation of $B$ given the stock price is

$$\hat{B}_t := E^P\left[B | \mathcal{F}_t^S\right] = G(t, Y_t),$$

where $G(t, y) = \left(\frac{\nabla F}{F}\right)(t, y)$.

Next, we introduce the so-called innovations process of filtering theory

$$N_t = Y_t - \int_0^t G(s, Y_s) ds,$$

and from Itô's lemma with $\hat{\Lambda}_t := \frac{1}{\hat{Z}_t}$, $(\langle ., . \rangle_t$ denotes the quadratic variation)

$$d\left(\hat{\Lambda}_t \cdot e^{-rt} X_t\right) = \hat{\Lambda}_t d\left(e^{-rt} X_t\right) + e^{-rt} X_t d\hat{\Lambda}_t + d\langle e^{-rt} X, \hat{\Lambda}\rangle_t$$

$$= e^{-rt}\left[\hat{\Lambda}_t \pi_t \sigma dY_t - \hat{\Lambda}_t X_t \hat{B}_t dN_t - \hat{\Lambda}_t \pi_t \sigma \hat{B}_t dt\right]$$

$$= e^{-rt}\hat{\Lambda}_t\left[\sigma \pi_t - X_t \hat{B}_t\right] dN_t.$$

Therefore, the process $f$ is an $(\mathcal{F}^S, P)$ local martingale, where $f(t) = e^{-rt}\hat{\Lambda}_t X_t$, thus we can use the martingale method as usual. If we define

$$I(x) = (u')^{-1}(x),$$

then it is not hard to show the optimal terminal wealth $X_T^*$ is given by $X_T^* = I\left(\frac{\mathcal{K}(x_0)e^{-rT}}{F(T,Y(T))}\right)$, (Karatzas & Zhao, 1998)where $\mathcal{K}(x_0)$ is a well-defined constant. Then Example 3.5 in Karatzas & Zhao (1998) gives the formulas of Theorem 28.

Finally, it suffices to show that there exists a measurable function $f$ such that $Y_t = f(t, S_t)$. We denote $S_0 = s_0 > 0$, then for $t \in [0, T]$,

$$S_t = s_0 \exp\left(\left(B - \frac{\sigma^2}{2}\right)t + \sigma W_t\right).$$

Therefore,

$$\log S_t = \log s_0 + \left(B - \frac{\sigma^2}{2}\right)t + \sigma W_t,$$

which implies that

$$W_t = \frac{\log\left(\frac{S_t}{s_0}\right) - \left(B - \frac{\sigma^2}{2}\right)t}{\sigma}.$$

Hence

$$Y_t = \frac{\ln\left(\frac{S_t}{s_0}\right)}{\sigma} + \left(\frac{\sigma}{2} - \frac{r}{\sigma}\right)t, \tag{13}$$

which finishes the proof. $\qquad\square$

In high dimensional case, the dynamic of risky assets becomes:

$$dS_i(t) = S_i(t)\left[b_i dt + \sum_{j=1}^d \sigma_{ij} dW_j(t)\right], \quad 0 \le t \le T,$$

where $b = (b_1, \ldots, b_d)^\top$ is the expected instantaneous rate of return of the asset, the invertible matrix $\sigma = \{\sigma_{ij}\}_{1 \le i,j \le d}$ is the instantaneous standard deviation of returns, and $0 < T < \infty$ is the terminal time. Then the optimal value function becomes

$$V(x_0) = \frac{\left(x_0 e^{rT}\right)^\alpha}{\alpha}\left(\int_{\mathbb{R}^d}(F(T,z))^{\frac{1}{1-\alpha}}\varphi_T(z)dz\right)^{1-\alpha},$$

where $F(t,z) := F_\mu(t,z) = \int_{\mathbb{R}^d} L_t(x,z)d\mu(x)$ with $L_t(x,z) = 1$ if $t = 0$ and

$$L_t(x,z) = \exp\left(\left(\sigma^{-1}(x - r\mathbf{1})\right)^\top z - \frac{1}{2}\left\|\sigma^{-1}(x - r\mathbf{1})\right\|_2^2 t\right)$$

And the optimal fraction is

$$\frac{\pi^*(t)}{X^*(t)} = \left(\sigma^\top\right)^{-1}\frac{\int_{\mathbb{R}^d}\nabla F(T, z + Y(t))(F(T, z + Y(t)))^{\frac{\alpha}{1-\alpha}}\varphi_{T-t}(z)dz}{(1-\alpha)\int_{\mathbb{R}^d}(F(T, z + Y(t)))^{\frac{1}{1-\alpha}}\varphi_{T-t}(z)dz},$$

## C PROOF OF RESULTS IN SECTION 2

### C.1 PROOF OF THEOREM 1

*Proof.* • (1) It is easy to see by checking the definitions, Equation (4) defines an equivalence relation, then the set of all equivalence classes under the quotient topology defines $\mathcal{U}_\delta$.

• (2) Fix $Q \in \mathcal{U}_\delta$, then there are 4 cases to check.

– (a) $\frac{dQ}{dP} = f(B)$, where $f : \mathbb{R} \to \mathbb{R}$ is bounded and (Borel) measurable. Then from definition of the uncertainty set, for any $A \in \mathcal{B}(\mathbb{R})$, there exists $\nu \ll \mu$ such that

$$Q(B \in A) = E^Q\left[\mathbf{1}_{\{B \in A\}}\right] = E^P\left[\frac{dQ}{dP}\mathbf{1}_{\{B \in A\}}\right] = E^P\left[f(B)\mathbf{1}_{\{B \in A\}}\right]$$

$$= \nu(A) = \int_A \frac{d\nu}{d\mu}d\mu = E^P\left[\frac{d\nu}{d\mu}(B)\mathbf{1}_{\{B \in A\}}\right],$$

which implies that $\frac{dQ}{dP} = \frac{d\nu}{d\mu}(B)$ $P$-almost surely from the uniqueness of Radon-Nikodym derivative.

- (b) $\frac{dQ}{dP} = f(W)$, where $f : \Omega \to \mathbb{R}$ is bounded and measurable. In this case, it is convenient to assume $\Omega = C([0, T]; \mathbb{R})$. Thus the standard Brownion motion $W : \Omega \to \Omega$ can be viewed as a function-valued random element. Therefore, for $A \in \mathcal{B}(\Omega)$,

$$Q(W \in A) = E^Q \left[\mathbf{1}_{\{W \in A\}}\right] = E^P \left[\frac{dQ}{dP}\mathbf{1}_{\{W \in A\}}\right]$$

$$= E^P \left[f(W)\mathbf{1}_{\{W \in A\}}\right] \neq P(W \in A)$$

unless $f = 1$, which is equivalent to the case when $\frac{dQ}{dP} = \frac{d\nu}{d\mu}(B)$ with $\nu = \mu$.

- (c) $\frac{dQ}{dP} = f(B, W)$, where $f : \mathbb{R} \times \Omega \to \mathbb{R}$ is bounded and jointly measurable. From part (b) and independence between $B$ and $W$, it is easy to construct a contradiction.

- (d) $\frac{dQ}{dP} = f(Y)$, where $f : \Omega \to \mathbb{R}$ is bounded and measurable, and $Y$ is a stochastic process that is independent of both $B$ and $W$. From the definition of the equivalence classes, this case is equivalent to the case when $\frac{dQ}{dP} = \frac{d\nu}{d\mu}(B)$ with $\nu = \mu$.

Moreover,

$$D_\phi(Q \parallel P) = \int \phi\left(\frac{dQ}{dP}\right) dP = \int \phi\left(\frac{d\nu}{d\mu}(B)\right) dP = \int \phi\left(\frac{d\nu}{d\mu}\right) d\mu = D_\phi(\nu \parallel \mu).$$

$\square$

# D PROOF OF RESULTS IN SECTION 3

## D.1 PROOF OF THEOREM 2

*Proof.* The proof of this utilizes the law invariance theory developed in Shapiro (2017) for $\phi$ divergence. Recall that we begin with the complete probability space $(\Omega, \mathcal{F}, P)$. Let $\hat{\mathcal{F}} = \sigma(B) \subset \mathcal{F}$, and define $\hat{P} = P|_{\hat{\mathcal{F}}}$, then the triple $(\Omega, \hat{\mathcal{F}}, \hat{P})$ is a probability space (may not be complete) such that for any $\hat{\mathcal{F}}$-measurable random variables $Z$,

$$E^P[Z] = E^{\hat{P}}[Z]. \tag{14}$$

Now, for a fixed $Q \in \mathcal{U}_\delta$, we define the restriction of $Q$ as $\hat{Q} = Q|_{\hat{\mathcal{F}}}$. We define the space $L^1\left(\Omega, \hat{\mathcal{F}}, \hat{P}\right) \subset L^1(\Omega, \mathcal{F}, P)$ as a subspace given that $u(X_T) \in L^p(\Omega, \mathcal{F}, P)$. We define an equivalence relation $\sim_\phi$ with respect to a convex function $\phi$ (which is easy to check) between two functions with mean 1 $X, Y \in L^1\left(\Omega, \hat{\mathcal{F}}, \hat{P}\right)$ by

$$X \sim_\phi Y \text{ if and only if } \int_\Omega \phi(X) d\hat{P} = \int_\Omega \phi(Y) d\hat{P}.$$

Following the notations in Shapiro (2017), we define a quotient space of $L^1\left(\Omega, \hat{\mathcal{F}}, \hat{P}\right)$ with respect to $\sim_\phi$ by

$$\hat{\mathcal{A}} = \left\{\left[\hat{X}\right], \text{ where } \hat{X} = \frac{d\hat{Q}}{d\hat{P}}, Q \in \mathcal{U}_\delta\right\}.$$

Let $\hat{\mathcal{U}}_\delta$ be the collection of restrictions of all $Q \in \mathcal{U}_\delta$ on $\hat{\mathcal{F}}$. For any $Q \in \mathcal{U}_\delta$, there is a unique $\hat{Q} \in \hat{\mathcal{U}}_\delta$ such that $\hat{Q} = Q|_{\hat{\mathcal{F}}}$ from definition. On the other hand, for a fixed $\hat{Q} \in \hat{\mathcal{U}}_\delta$, there is also a unique $Q \in \mathcal{U}_\delta$ such that $\hat{Q} = Q|_{\hat{\mathcal{F}}}$. To see this, suppose $Q_1, Q_2 \in \mathcal{U}_\delta$ and $\hat{Q} = Q_1|_{\hat{\mathcal{F}}} = Q_2|_{\hat{\mathcal{F}}}$, which implies that $Q_1 \sim Q_2$, hence uniqueness is shown.

We define two $\hat{\mathcal{F}}$-measurable random variables $X$ and $Y$ to be distributionally equivalent under the measure $\hat{P}$ if for any $A \in \mathcal{B}(\mathbb{R})$, $\hat{P}(X \in A) = \hat{P}(Y \in A)$. Next, we notice that for any $X_1$ and $X_2$ that are two $\hat{\mathcal{F}}$-measurable random variables such that $X_1 \in \hat{\mathcal{A}}$ and $X_1$ and $X_2$ are distributionally equivalent, then $X_2 \in \hat{\mathcal{A}}$.

For a fixed probability measure $Q \in \mathcal{U}_\delta$, we know that

$$E^Q[u(X_T)] = E^Q\left[E^{Q^B}[u(X_T)]\right],$$

where $Q^B$ is the regular conditional probability under $Q$ (the current measure) conditioned on the random variable $B$. The random variable $E^{Q^B}[u(X_T)]$ is $\hat{\mathcal{F}}$-measurable.

Next, we define the functional corresponding to the law invariant set $\hat{\mathcal{A}}$, then from the law invariance theory in Shapiro (2017), this functional is also law invariant thus implies the strong duality. We firstly define a function space $D = \{\alpha E^{P^B}[u(X_T)], \alpha \in \mathbb{R}\}$, which is an one-dimensional linear subspace of $L^p\left(\Omega, \hat{\mathcal{F}}, \hat{P}\right)$ given that $u(X_T) \in L^p(\Omega, \mathcal{F}, P)$. Note that here $P^B$ represents the regular conditional distribution given $B$ under $P$, and its notation will be changed to $Q^B$ if the underlying probability measure is changed to $Q$. From the one-to-one correspondence between $\hat{\mathcal{U}}_\delta$ and $\hat{\mathcal{A}}$, the functional $\rho : D \to \mathbb{R}$ defined by

$$\rho(Z) = \inf_{\hat{Q} \in \hat{\mathcal{U}}_\delta} E^{\hat{Q}}[Z] = \inf_{\hat{Q} \in \hat{\mathcal{U}}_\delta} E^{\hat{Q}}\left[E^{Q^B}[u(X_T)]\right]$$

is law invariant with respect to $\left(\Omega, \hat{\mathcal{F}}, \hat{P}\right)$.

Since the topological and convexity structures are the preserved, then with a similar argument as in Section 3.2 in Shapiro (2017), the Lagrangian of the problem $\inf_{\hat{Q} \in \hat{\mathcal{U}}_\delta} E^{\hat{Q}}\left[E^{Q^B}[u(X_T)]\right]$ is given by

$$\mathcal{L}_Z\left(\hat{X}, \lambda, \beta\right) = \int_\Omega Z\hat{X} dP + \lambda\left(\int_\Omega \phi\left(\hat{X}\right) dP - \delta\right) + \beta\left(1 - \int_\Omega \hat{X} dP\right)$$

$$= \beta - \lambda\delta + \int_\Omega\left(Z\hat{X} + \lambda\phi\left(\hat{X}\right) - \beta\hat{X}\right) dP,$$

and the Lagrangian dual problem is

$$\sup_{\lambda \geq 0, \mu \in \mathbb{R}} \inf_{\hat{X} \geq 0} \mathcal{L}_Z\left(\hat{X}, \lambda, \beta\right).$$

Since the space $L^p\left(\Omega, \hat{\mathcal{F}}, \hat{P}\right)$ is decomposable, then as in Shapiro (2017),

$$\inf_{\hat{X} \geq 0} \mathcal{L}_Z\left(\hat{X}, \lambda, \beta\right) = \beta - \lambda\delta + \inf_{\hat{X} \geq 0}\left\{\int_\Omega\left(Z\hat{X} + \lambda\phi\left(\hat{X}\right) - \beta\hat{X}\right) d\hat{P}\right\}$$

$$= \beta - \lambda\delta + \int_\Omega \inf_{x \geq 0}(Zx + \lambda\phi(x) - \beta x) d\hat{P}$$

$$= \beta - \lambda\delta + \int_\Omega -(\lambda\phi)^*(\beta - Z) d\hat{P}$$

$$= \beta - \lambda\delta + E^{\hat{P}}\left[-(\lambda\phi)^*(\beta - E^{P^B}[u(X_T)])\right]$$

$$= \beta - \lambda\delta + E^{\hat{P}}\left[\Phi_{\lambda,\beta}\left(E^{P^B}[u(X_T)]\right)\right].$$

Therefore,

$$\inf_{Q \in \mathcal{U}_\delta} E^Q \left[ u(X_T) \right] = \inf_{Q \in \mathcal{U}_\delta} E^Q \left[ E^{Q^B} \left[ u(X_T) \right] \right]$$

$$= \inf_{\hat{Q} \in \hat{\mathcal{U}}_\delta} E^{\hat{Q}} \left[ E^{Q^B} \left[ u(X_T) \right] \right]$$

$$= \sup_{\lambda \geq 0, \beta \in \mathbb{R}} \left\{ \beta - \lambda\delta + E^{\hat{P}} \left[ \Phi_{\lambda,\beta} \left( E^{P^B} \left[ u(X_T) \right] \right) \right] \right\}$$

$$= \sup_{\lambda \geq 0, \beta \in \mathbb{R}} \left\{ \beta - \lambda\delta + E^P \left[ \Phi_{\lambda,\beta} \left( E^{P^B} \left[ u(X_T) \right] \right) \right] \right\}$$

$$= \sup_{\lambda \geq 0, \beta \in \mathbb{R}} \left\{ \beta - \lambda\delta + \int_{\mathbb{R}} \Phi_{\lambda,\beta} \left( E^{P^b} \left[ u(X_T) \right] \right) d\mu(b) \right\},$$

which finishes the proof. $\qquad \square$

*Remark* 29. If we plug in $u(X_T)$ as an $\mathcal{F}$-measurable random variable, then the strong duality would look like

$$\inf_{Q \in \mathcal{U}_\delta} E^Q \left[ u(X_T) \right] = \sup_{\lambda > 0, \beta \in \mathbb{R}} \left\{ \beta - \lambda\delta + \lambda E^P \left[ G_{\lambda,\beta}(X_T) \right] \right\},$$

where for fixed $\lambda > 0, \beta \in \mathbb{R}$ (assuming multipliers nonzero),

$$G_{\lambda,\beta}(x) := (-\phi^*) \left( \frac{\beta - u(x)}{\lambda} \right).$$

The strong duality looks better than the one stated in Theorem 2 since it is more tractable. However, this is wrong since $\mathcal{U}_\delta$ is not law invariant with respect to $\mathcal{F}$. As in the language of Shapiro (2017), suppose we have a Radon-Nikodym derivative $X = f(B)$, where $f$ is a measurable function and $B \in \mathcal{N}(0, 1)$, define $Y = f(W_1)$, then $X$ and $Y$ are distributionally equivalent but $Y$ cannot induce a probability measure that belongs to $\mathcal{U}_\delta$ (by Theorem 1).

### D.2 PROOF OF THEOREM 3

*Proof.* Since $\phi(x) = x \log x - x + 1$, then $\Phi_{\lambda,\beta}(x) = \lambda \left( 1 - \exp \left( \frac{\beta - x}{\lambda} \right) \right)$ if $\lambda > 0$. Therefore, if optimal $\lambda^* \neq 0$, then

$$\inf_{Q \in \mathcal{U}_{\text{KL},\delta}} E^Q \left[ u(X_T) \right] = \sup_{\lambda \geq 0, \beta \in \mathbb{R}} \left\{ \beta - \lambda\delta + \int_{\mathbb{R}} \lambda \left( 1 - \exp \left( \frac{\beta - E^{P^b}[u(X_T)]}{\lambda} \right) \right) d\mu(b) \right\}.$$

If we take derivative with respect to $\beta$, then we get the optimal $\beta^*$

$$\beta^* = -\lambda \log \left( \int_{\mathbb{R}} \exp \left( \frac{-E^{P^b}[u(X_T)]}{\lambda} \right) d\mu(b) \right).$$

After plugging in, we have

$$\inf_{Q \in \mathcal{U}_{\text{KL},\delta}} E^Q \left[ u(X_T) \right] = \sup_{\lambda \geq 0} \left\{ -\lambda\delta - \lambda \log \left( \int_{\mathbb{R}} \exp \left( \frac{-E^{P^b}[u(X_T)]}{\lambda} \right) d\mu(b) \right) \right\}.$$

The case when $\lambda = 0$ is from discussion of case 1 after Assumption 1 in Hu & Hong (2013). $\qquad \square$

### D.3 EXTENSION TO CRESSIE-READ DIVERGENCE

We give a theorem for the Cressie-Read divergence with the uncertainty set is denoted as $\mathcal{U}_{k,\delta}$ for $\phi_k(x) = \frac{x^k - kx + k - 1}{k(k-1)}$ with $k \in (1, \infty)$. We further define $k_* = \frac{k}{k-1}$ and $c_k(\delta) = (1 + k(k-1)\delta)^{\frac{1}{k}}$.

**Theorem 30.**

$$\inf_{Q \in \mathcal{U}_{k,\delta}} E^Q \left[ u(X_T) \right] = \sup_{\beta \in \mathbb{R}} \left\{ \beta - c_k(\delta) \left( \int_{\mathbb{R}} \left( \beta - E^{P^b} [u(X_T)] \right)_+^{k_*} d\mu(b) \right)^{\frac{1}{k_*}} \right\}.$$

*Proof.* From Duchi & Namkoong (2021), we know that

$$\phi_k^*(x) = \frac{1}{k}\left((k-1)x+1\right)_+^{k_*} - \frac{1}{k}.$$

Therefore, by plugging in this to Theorem 2, we have

$$\inf_{Q \in \mathcal{U}_{k,\delta}} E^Q\left[u(X_T)\right]$$

$$= \sup_{\lambda \geq 0, \beta \in \mathbb{R}} \left\{\beta - \lambda\delta + \lambda E^P\left[\Phi_{\lambda,\beta}\left(E^{P^B}\left[u(X_T)\right]\right)\right]\right\}$$

$$= \sup_{\lambda \geq 0, \beta \in \mathbb{R}} \left\{\beta - \lambda\left(\delta - \frac{1}{k}\right) - \lambda^{1-k_*}\frac{(k-1)^{k_*}}{k}E^P\left[\left(\beta - E^{P^B}\left[u(X_T)\right] + \frac{\lambda}{k-1}\right)_+^{k_*}\right]\right\}$$

$$= \sup_{\lambda \geq 0, \tilde{\beta} \in \mathbb{R}} \left\{\tilde{\beta} - \lambda\left(\delta + \frac{1}{k(k-1)}\right) - \lambda^{1-k_*}\frac{(k-1)^{k_*}}{k}E^P\left[\left(\tilde{\beta} - E^{P^B}\left[u(X_T)\right]\right)_+^{k_*}\right]\right\},$$

where we define $\tilde{\beta} = \beta + \frac{\lambda}{k-1}$ thus the last equality holds. Noting that $\frac{k_*-1}{k_*} = \frac{1}{k}$ and taking derivatives with respect to $\lambda$ to minimize the preceding expression, we have

$$\lambda = (k-1)\left(\delta k(k-1)+1\right)^{-\frac{1}{k_*}}\left(E^P\left[\left(\tilde{\beta} - E^{P^B}\left[u(X_T)\right]\right)_+^{k_*}\right]\right)^{\frac{1}{k_*}}.$$

By substituting back, we have

$$\inf_{Q \in \mathcal{U}_{k,\delta}} E^Q\left[u(X_T)\right]$$

$$= \sup_{\tilde{\beta} \in \mathbb{R}} \left\{\tilde{\beta} - \left(\delta k(k-1)+1\right)^{\frac{1}{k}}\left(E^P\left[\left(\tilde{\beta} - E^{P^B}\left[u(X_T)\right]\right)_+^{k_*}\right]\right)^{\frac{1}{k_*}}\right\}$$

$$= \sup_{\tilde{\beta} \in \mathbb{R}} \left\{\tilde{\beta} - c_k(\delta)\left(\int_{\mathbb{R}}\left(\tilde{\beta} - E^{P^b}\left[u(X_T)\right]\right)_+^{k_*}d\mu(b)\right)^{\frac{1}{k_*}}\right\},$$

which finishes the proof. $\qquad\square$

### D.4 Existence and Uniqueness of the Dual Optimizer

**Theorem 31.** *Assume Assumption 6 and 7 hold, then for a fixed $\pi \in \mathcal{A}(x_0)$, there exist positive and finite $\bar{\lambda}$ and $\underline{\lambda}$ such that any optimal $\lambda^*(\pi) \in [\underline{\lambda}, \bar{\lambda}]$.*

*Proof.* From Jensen's inequality, we have

$$\log\left(\int_{\mathbb{R}}\exp\left(\frac{-Z^b}{\lambda}\right)d\mu(b)\right) \geq \int_{\mathbb{R}}\log\left(\exp\left(\frac{-Z^b}{\lambda}\right)\right)d\mu(b) = \frac{-1}{\lambda}E^P[Z^B],$$

thus

$$-\lambda\log\left(\int_{\mathbb{R}}\exp\left(\frac{-Z^b}{\lambda}\right)d\mu(b)\right) - \lambda\delta \leq E^P[Z^B] - \lambda\delta \leq M - \lambda\delta.$$

On the other hand,

$$\inf_{Q \in \mathcal{U}_{\text{KL},\delta}} E^Q\left[u(X_T)\right] \geq \operatorname{ess\,inf} u(X_T) \geq 0$$

since $X_T \geq 0$ and $u$ is strictly increasing, then $M - \lambda\delta \geq 0$ gives the upper bound $\lambda^*(\pi) \leq \bar{\lambda} := \frac{M}{\delta}$.

For the lower bound, it suffices to show there exists $\lambda > 0$ such that

$$-\lambda\log\left(\int_{\mathbb{R}}\exp\left(\frac{-E^{P^b}[u(X_T)]}{\lambda}\right)d\mu(b)\right) - \lambda\delta > \operatorname{ess\,inf} E^{P^B}[u(X_T)] \geq u(m),$$

which is equivalent to

$$E\left[\exp\left(\frac{-E^{P^b}[u(X_T)] + u(m)}{\lambda}\right)\right] < e^{-\delta}.$$

Since as $\lambda \to 0$, LHS $\to 0$ (as a left limit), thus as long as we pick $\lambda > 0$ small enough, then the inequality is achieved, thus the optimal $\lambda^* > 0$, thus the optimal $\lambda^*$ is attained inside the interval $[\underline{\lambda}, \bar{\lambda}]$. □

*Remark* 32. We remark that for the proof of upper bound of the dual optimizer, the moment condition can be relaxed to $\int_{\mathbb{R}} V_b(x_0) d\mu(b) < \infty$. The bound $\int_{\mathbb{R}} \big(V_b(x_0)\big)^{3(1+w)} d\mu(b) < \infty$ is essential in the construction of the rMLMC estimator.

**Theorem 33.** *If Assumption 6 and 8 satisfied, then for a fixed $\pi \in \mathcal{A}(x_0)$, there exist positive and finite $\bar{\lambda}$ and $\underline{\lambda}$ such that any optimal $\lambda^*(\pi) \in [\underline{\lambda}, \bar{\lambda}]$.*

*Proof.* The proof of the upper bound is the same as proof of Theorem 31. For the lower bound, even though the same proof as Theorem 31 can be done, we adapt the proof of Lemma A12 from Si et al. (2023) to illustrate the usefulness of the stronger Assumption 8.

Essentially, the proof of density and mass function cases are the same, so without loss of generality, we use $f$ to denote the continuous density of $E^{P^B}[u(X_T)]$ on a compact set $K$, thus $\bar{b} = \sup_{x \in K} f(x) < \infty$ exists. Further, we define for a fixed $\pi \in \mathcal{A}(x_0)$, $g(\lambda) = g_\pi(\lambda) = -\lambda \log\left(\int_{\mathbb{R}} \exp\left(\frac{-Z^b}{\lambda}\right) d\mu(b)\right) - \lambda\delta$. Therefore, since

$$\lim_{\lambda \to 0} g(\lambda) = 0,$$

it suffices to show that

$$\liminf_{\lambda \to 0} \frac{dg(\lambda)}{d\lambda} > 0.$$

Indeed,

$$\frac{dg(\lambda)}{d\lambda} = \frac{\int_{\mathbb{R}} \frac{-Z^b}{\lambda} \exp\left(\frac{-Z^b}{\lambda}\right) d\mu(b)}{\int_{\mathbb{R}} \exp\left(\frac{-Z^b}{\lambda}\right) d\mu(b)} - \delta - \log\left(\int_{\mathbb{R}} \exp\left(\frac{-Z^b}{\lambda}\right) d\mu(b)\right).$$

We notice that

$$\lim_{\lambda \to 0} -\log\left(\int_{\mathbb{R}} \exp\left(\frac{-Z^b}{\lambda}\right) d\mu(b)\right) = \infty$$

and

$$\liminf_{\lambda \to 0} \frac{\int_{\mathbb{R}} \frac{-Z^b}{\lambda} \exp\left(\frac{-Z^b}{\lambda}\right) d\mu(b)}{\int_{\mathbb{R}} \exp\left(\frac{-Z^b}{\lambda}\right) d\mu(b)} = \liminf_{\lambda \to 0} \frac{\int_{\mathbb{R}} \frac{-Z^b}{\lambda} \exp\left(\frac{-Z^b}{\lambda}\right) d\mu(b)/\lambda}{\int_{\mathbb{R}} \exp\left(\frac{-Z^b}{\lambda}\right) d\mu(b)/\lambda} \geq -\frac{\bar{b}}{b}$$

since

$$\int_0^\infty \frac{y}{\lambda} e^{-\frac{y}{\lambda}} dy = \int_0^\infty e^{-\frac{y}{\lambda}} dy = \lambda.$$

As a result,

$$\liminf_{\lambda \to 0} \frac{dg(\lambda)}{d\lambda} = \infty.$$

Since the objective function is strictly concave in $\lambda$ (Theorem 39), then the uniqueness is shown, thus finishes the proof. □

*Remark* 34. Similarly, mild assumptions on the moments (Duchi & Namkoong, 2021) gives the uniqueness and strictly concavity of the objective function in the Cressie-Read divergence case.

# E    SUPPLEMENTARY FOR SECTION 4

We first provide two lemmas that will be used in the proof of Theorem 11.

**Lemma 35.** *Suppose Assumption 10 holds and $\mu$ is compactly supported with a continuous density, then for each fixed $\pi \in \mathcal{A}(x_0)$, there exists a constant $M > 0$ such that with probability one,*

$$0 \leq Z^B = E^{P^B}[u(X_T)] \leq M.$$

*That is, the random variable $Z^B$ has a compact support.*

*Proof.* Since function $b \mapsto Z^b$ is continuously differentiable and monotone on its support, thus the random variable $Z^B$ has a continuous density. Since the support is compact, then with probability one, $Z^B$ has attainable maximum and minimum. $\square$

*Remark* 36. Lemma 35 helps establish the finite variance of the estimator $\mathcal{E}_{\pi,\delta,\lambda,\beta}$. The same result will be automatically true if Assumption 8 holds.

### E.1 Proof of Theorem 11

*Proof.* With no confusion in the notation, we use $\mathcal{E}^b = \mathcal{E}^b_{\lambda,\beta}$ and $\mathcal{E} = \mathcal{E}_{\pi,\delta,\lambda,\beta}$ for simplicity.

Since $\Phi_{\lambda,\beta}$ satisfies the linear growth condition, then for a fixed $b \in \mathbb{R}$, there exists $c_1 > 0$ such that

$$\left| \Phi_{\lambda,\beta}\left(\frac{S_n^b}{n}\right) \right|^2 \leq c_1 \left(1 + \left\| \frac{S_n^b}{n} \right\|_2^2 \right).$$

Hence, Assumption 6 implies that

$$E^{P^b}\left[ \left| \Phi_{\lambda,\beta}\left(\frac{S_n^b}{n}\right) \right|^2 \right] \leq c_1 \left(1 + \frac{\text{Var}^b\left(\hat{Z}_1^b\right)}{n} + E^{P^b}\left[\hat{Z}_1^b\right] \right) < \infty, \tag{15}$$

where $\text{Var}^b$ represents the corresponding expectation is taken with respect to the measure $P^b$. Thus $\Phi_{\lambda,\beta}\left(\frac{S_n^b}{n}\right)$ is uniformly integrable with respect to $P^b$. Since for each $n \geq n_0$ (samples are i.i.d.),

$$E^P\left[\Delta_n^{b_1}\right] = E^P\left[\Phi_{\lambda,\beta}\left(\frac{S_{2^{n+1}}^{b_1}}{2^{n+1}}\right)\right] - E^P\left[\Phi_{\lambda,\beta}\left(\frac{S_{2^n}^{b_1}}{2^n}\right)\right],$$

then from the dominated convergence theorem,

$$\begin{aligned}
E^{P^{b_1}}[\mathcal{E}] = E^P\left[\mathcal{E}^{b_1}\right] \\
= E^P\left[\frac{\Delta_{N^{b_1}}^{b_1}}{p(N^{b_1})}\right] + E^P\left[\Phi_{\lambda,\beta}\left(\frac{S_{2^{n_0}}^{b_1}}{2^{n_0}}\right)\right] \\
= \sum_{n=n_0}^{\infty} E^P\left[\Delta_n^{b_1}\right] + E^P\left[\Phi_{\lambda,\beta}\left(\frac{S_{2^{n_0}}^{b_1}}{2^{n_0}}\right)\right] \sum_{n=n_0+1}^{\infty} E^P\left[\Delta_n^{b_1}\right] \\
= \lim_{m\to\infty} E^P\left[\Phi_{\lambda,\beta}\left(\frac{S_{2^m}^{b_1}}{2^m}\right)\right] = E^P\left[\lim_{m\to\infty}\Phi_{\lambda,\beta}\left(\frac{S_{2^m}^{b_1}}{2^m}\right)\right] = \Phi_{\lambda,\beta}\left(Z^{b_1}\right).
\end{aligned}$$

Finally, from tower property, we have

$$E^P[\mathcal{E}] = E^P\left[E^{P^B}[\mathcal{E}]\right] = \int_{\mathbb{R}} E^{P^b}[\mathcal{E}]\,d\mu(b) = \int_{\mathbb{R}} \Phi_{\lambda,\beta}\left(Z^b\right) d\mu(b),$$

which shows the unbiasedness.

Lemma 35 and Assumption 4 imply that it suffices to restrict the study of the function $\Phi((\lambda,\beta), Z^b) = \Phi_{\lambda,\beta}\left(Z^b\right)$ on a compact product $K_1 \times K_2$, thus Assumption 5 implies that there exists a constant $K > 0$ such that

$$\sup_{((\lambda,\beta),Z^b)\in K_1\times K_2} \left\| D^2\Phi_{\lambda,\beta} \right\| < K. \tag{16}$$

In order to show $\text{Var}\left(\mathcal{E}^{b_1}\right) < \infty$, from Equation (15), it suffices to show $E^P\left[\left(\frac{\Delta_N^B}{p(N)}\right)^2\right] < \infty$.

From Taylor's expansion, we have for fixed $n \in \mathbb{N}$, with probability one, there exists $\xi_{n+1}^B$ between $Z^B$ and $\frac{S_{2^{n+1}}^B}{2^{n+1}}$, $\xi_n^{O,B}$ between $Z^B$ and $\frac{S_{2^n}^{O,B}}{2^n}$, and $\xi_n^{E,B}$ between $Z^B$ and $\frac{S_{2^n}^{E,B}}{2^n}$ such that

$$\Phi_{\lambda,\beta}\left(\frac{S_{2^{n+1}}^B}{2^{n+1}}\right) = \Phi_{\lambda,\beta}\left(Z^B\right) + D\Phi_{\lambda,\beta}\left(Z^B\right)\left(\frac{S_{2^{n+1}}^B}{2^{n+1}} - Z^B\right) + \frac{1}{2}D^2\Phi_{\lambda,\beta}\left(\xi_{n+1}^B\right)\left(\frac{S_{2^{n+1}}^B}{2^{n+1}} - Z^B\right)^2,$$

$$\Phi_{\lambda,\beta}\left(\frac{S^{O,B}_{2^n}}{2^n}\right) = \Phi_{\lambda,\beta}\left(Z^B\right) + D\Phi_{\lambda,\beta}\left(Z^B\right)\left(\frac{S^{O,B}_{2^n}}{2^n} - Z^B\right) + \frac{1}{2}D^2\Phi_{\lambda,\beta}\left(\xi^{O,B}_n\right)\left(\frac{S^{O,B}_{2^n}}{2^n} - Z^B\right)^2,$$

and

$$\Phi_{\lambda,\beta}\left(\frac{S^{E,B}_{2^n}}{2^n}\right) = \Phi_{\lambda,\beta}\left(Z^B\right) + D\Phi_{\lambda,\beta}\left(Z^B\right)\left(\frac{S^{E,B}_{2^n}}{2^n} - Z^B\right) + \frac{1}{2}D^2\Phi_{\lambda,\beta}\left(\xi^{E,B}_n\right)\left(\frac{S^{E,B}_{2^n}}{2^n} - Z^B\right)^2.$$

Therefore,

$$\Delta^B_n = \Phi_{\lambda,\beta}\left(\frac{S^B_{2^{n+1}}}{2^{n+1}}\right) - \frac{1}{2}\left(\Phi_{\lambda,\beta}\left(\frac{S^{O,B}_{2^n}}{2^n}\right) + \Phi_{\lambda,\beta}\left(\frac{S^{E,B}_{2^n}}{2^n}\right)\right)$$

$$= D\Phi_{\lambda,\beta}\left(Z^B\right)\left(\frac{S^B_{2^{n+1}}}{2^{n+1}} - Z^B\right) + \frac{1}{2}D^2\Phi_{\lambda,\beta}\left(\xi^B_{n+1}\right)\left(\frac{S^B_{2^{n+1}}}{2^{n+1}} - Z^B\right)^2$$

$$- \frac{1}{2}\left[D\Phi_{\lambda,\beta}\left(Z^B\right)\left(\frac{S^{O,B}_{2^n}}{2^n} - Z^B\right) + \frac{1}{2}D^2\Phi_{\lambda,\beta}\left(\xi^{O,B}_n\right)\left(\frac{S^{O,B}_{2^n}}{2^n} - Z^B\right)^2\right]$$

$$- \frac{1}{2}\left[D\Phi_{\lambda,\beta}\left(Z^B\right)\left(\frac{S^{E,B}_{2^n}}{2^n} - Z^B\right) + \frac{1}{2}D^2\Phi_{\lambda,\beta}\left(\xi^{E,B}_n\right)\left(\frac{S^{E,B}_{2^n}}{2^n} - Z^B\right)^2\right].$$

Since $\hat{Z}^B$ is an unbiased estimate of $Z^B$, then from the boundedness of second derivative (Equation (16)),

$$E^P\left[\left(\Delta^B_n\right)^2\right] \leq \frac{1}{4}E^P\left[\left\|D^2\Phi_{\lambda,\beta}\left(\xi^B_{n+1}\right)\left(\frac{S^B_{2^{n+1}}}{2^{n+1}} - Z^B\right)^2\right\|^2_2\right]$$

$$+ \frac{1}{8}E^P\left[\left\|D^2\Phi_{\lambda,\beta}\left(\xi^{O,B}_n\right)\left(\frac{S^{O,B}_{2^n}}{2^n} - Z^B\right)^2\right\|^2_2\right] + \frac{1}{8}E^P\left[\left\|D^2\Phi_{\lambda,\beta}\left(\xi^{E,B}_n\right)\left(\frac{S^{E,B}_{2^n}}{2^n} - Z^B\right)^2\right\|^2_2\right]$$

$$\leq \frac{K}{4}E^P\left[\left\|\left(\frac{S^B_{2^{n+1}}}{2^{n+1}} - Z^B\right)^2\right\|^2_2\right] + \frac{K}{8}E^P\left[\left\|\left(\frac{S^{O,B}_{2^n}}{2^n} - Z^B\right)^2\right\|^2_2\right] + \frac{K}{8}E^P\left[\left\|\left(\frac{S^{E,B}_{2^n}}{2^n} - Z^B\right)^2\right\|^2_2\right].$$

From Jensen's inequality and Lemma 23, there exists a constant $K_2$ such that

$$E^P\left[\left\|\left(\frac{S^B_{2^{n+1}}}{2^{n+1}} - Z^B\right)^2\right\|^2_2\right] \leq E^P\left[\left\|\frac{S^B_{2^{n+1}}}{2^{n+1}} - Z^B\right\|^4_2\right] \leq \frac{K_2}{2^{2(n+1)}}E^P\left[\left\|Z^B\right\|^4_2\right] \lesssim \mathcal{O}\left(2^{-2n}\right),$$

where $f(n) \lesssim \mathcal{O}\left(2^{-2n}\right)$ means there is a constant $C > 0$ such that $f(n) \leq C2^{-2n}$.

Therefore, there exists a constant $C > 0$ such that

$$E^P\left[\left(\frac{\Delta^B_N}{p(N)}\right)^2\right] = \sum_{n=0}^{\infty}\frac{E^P\left[\left(\Delta^B_n\right)^2\right]}{p(n)} \leq C\sum_{n=0}^{\infty}\frac{2^{-2n}}{p(n)} < \infty,$$

from the discussion of in Section 3 in Blanchet & Glynn (2015). Therefore, $\mathrm{Var}\left(\mathcal{E}_{\pi,\delta,\lambda,\beta}\right)$ is also finite.

$\square$

## E.2 First and Second Order Conditions

In this section, we give the first and second order conditions for the general $\phi$-divergence case with $\Phi_{\lambda,\beta}$ strictly concave and strictly increasing in $(\lambda, \beta)^T$ for large $n$. We remark that for a fixed

$n \in \mathbb{N}$, the value function is not strictly concave in $(\lambda, \beta)^T$ since the definition of rMLMC estimator introduces negative linear combination of strictly concave functions. Since the estimator is not strictly concave and is random, it may be better to use the (stochastic) gradient descent method to escape from irregular points (e.g. saddle points).

**Theorem 37.** *Define $F(\lambda, \beta; \pi, \delta) = \beta - \lambda\delta + \mathcal{E}$ for fixed $\delta > 0$ and policy $\pi$, then $F$ is strictly concave in $(\lambda, \beta)$ as $n \to \infty$ under Assumption 5 and its first and second (mixed) partial derivatives admit the expressions*

$$\frac{\partial F}{\partial \lambda} := \frac{\partial F(\lambda, \beta; \pi, \delta)}{\partial \lambda} = \frac{1}{n} \sum_{i=1}^{n} \frac{\Delta_{N^{b_i}, \lambda}^{b_i}}{p(N^{b_i})} - \delta + \frac{\partial \Phi_{\lambda, \beta}}{\partial \lambda} \left( \frac{S_{2^{n_0}}^{b_i}}{2^{n_0}} \right), \tag{17}$$

$$\frac{\partial F}{\partial \beta} := \frac{\partial F(\lambda, \beta; \pi, \delta)}{\partial \beta} = \frac{1}{n} \sum_{i=1}^{n} \frac{\Delta_{N^{b_i}, \beta}^{b_i}}{p(N^{b_i})} + 1 + \frac{\partial \Phi_{\lambda, \beta}}{\partial \beta} \left( \frac{S_{2^{n_0}}^{b_i}}{2^{n_0}} \right), \tag{18}$$

$$\frac{\partial^2 F}{\partial \lambda \partial \beta} := \frac{\partial^2 F(\lambda, \beta; \pi, \delta)}{\partial \lambda \partial \beta} = \frac{1}{n} \sum_{i=1}^{n} \frac{\Delta_{N^{b_i}, \lambda, \beta}^{b_i}}{p(N^{b_i})} + \frac{\partial^2 \Phi_{\lambda, \beta}}{\partial \lambda \partial \beta} \left( \frac{S_{2^{n_0}}^{b_i}}{2^{n_0}} \right), \tag{19}$$

$$\frac{\partial^2 F}{\partial \lambda^2} := \frac{\partial^2 F(\lambda, \beta; \pi, \delta)}{\partial \lambda^2} = \frac{1}{n} \sum_{i=1}^{n} \frac{\Delta_{N^{b_i}, \lambda^2}^{b_i}}{p(N^{b_i})} + \frac{\partial^2 \Phi_{\lambda, \beta}}{\partial \lambda^2} \left( \frac{S_{2^{n_0}}^{b_i}}{2^{n_0}} \right), \tag{20}$$

*and*

$$\frac{\partial^2 F}{\partial \beta^2} := \frac{\partial^2 F(\lambda, \beta; \pi, \delta)}{\partial \beta^2} = \frac{1}{n} \sum_{i=1}^{n} \frac{\Delta_{N^{b_i}, \beta^2}^{b_i}}{p(N^{b_i})} + \frac{\partial^2 \Phi_{\lambda, \beta}}{\partial \beta^2} \left( \frac{S_{2^{n_0}}^{b_i}}{2^{n_0}} \right), \tag{21}$$

*where for fixed $b \in \mathbb{R}$, $n \in \mathbb{N}$, and $y, y' \in \{\lambda, \beta\}$,*

$$\Delta_{n,y}^{b} = \frac{\partial}{\partial y} \Phi_{\lambda, \beta} \left( \frac{S_{2^{n+1}}^{b}}{2^{n+1}} \right)$$
$$- \frac{1}{2} \frac{\partial}{\partial y} \left( \Phi_{\lambda, \beta} \left( \frac{S_{2^n}^{O,b}}{2^n} \right) + \Phi_{\lambda, \beta} \left( \frac{S_{2^n}^{E,b}}{2^n} \right) \right),$$

$$\Delta_{n,y^2}^{b} = \frac{\partial^2}{\partial y^2} \Phi_{\lambda, \beta} \left( \frac{S_{2^{n+1}}^{b}}{2^{n+1}} \right)$$
$$- \frac{1}{2} \frac{\partial^2}{\partial y^2} \left( \Phi_{\lambda, \beta} \left( \frac{S_{2^n}^{O,b}}{2^n} \right) + \Phi_{\lambda, \beta} \left( \frac{S_{2^n}^{E,b}}{2^n} \right) \right),$$

*and*

$$\Delta_{n,y,y'}^{b} = \frac{\partial^2}{\partial y \partial y'} \Phi_{\lambda, \beta} \left( \frac{S_{2^{n+1}}^{b}}{2^{n+1}} \right)$$
$$- \frac{1}{2} \frac{\partial^2}{\partial y \partial y'} \left( \Phi_{\lambda, \beta} \left( \frac{S_{2^n}^{O,b}}{2^n} \right) + \Phi_{\lambda, \beta} \left( \frac{S_{2^n}^{E,b}}{2^n} \right) \right).$$

*Proof.* Since $\Phi_{\lambda, \beta}$ is strictly concave in $(\lambda, \beta)^T$ and $\mathcal{E}$ is an unbiased and consistent estimator, then so is $F$ as $n \to \infty$. The computation of derivatives is elementary. $\square$

Next, we give a specific example of the KL divergence case. Recall that in this case,

$$Q_{\text{DRBCKL}}(\pi) = \sup_{\lambda > 0} \left\{ -\lambda\delta - \lambda \log \left( \int_{\mathbb{R}} \exp \left( \frac{-Z^b}{\lambda} \right) d\mu(b) \right) \right\}, $$

which has a form that is more complicated than the general case: in the KL case there are two nonlinear transformations $\log$ and $\exp$ in the nested expectation, thus to get an unbiased estimator of $Q_{\text{DRBCKL}}(\pi)$ we need to apply the rMLMC method recursively (Syed & Wang, 2023).

Another way is to keep the form when there are two multipliers $\lambda$ and $\beta$ rather than using the form $Q_{\text{DRBCKL}}(\pi) = \sup_{\lambda > 0} \left\{ -\lambda\delta - \lambda \log \left( \int_{\mathbb{R}} \exp \left( \frac{-Z^b}{\lambda} \right) d\mu(b) \right) \right\}$. This can avoid the multi-layer nested expectation and nonlinear transformations if the $\Phi_{\lambda,\beta}$ corresponding to the KL divergence is strictly concave. However, this is always false.

**Theorem 38.** *When $\phi(x) = x \log x - x + 1$, then $\Phi_{\lambda,\beta}$ is concave but it is never strictly concave in $(\lambda, \beta)^T$ (jointly).*

*Proof.* If $\phi(x) = x \log x - x + 1$, then $\Phi_{\lambda,\beta}(x) = \lambda \left( 1 - \exp \left( \frac{\beta - x}{\lambda} \right) \right)$ if $\lambda > 0$. Therefore, if optimal $\lambda^* \neq 0$, then

$$\frac{\partial}{\partial \lambda} \Phi_{\lambda,\beta}(x) = 1 + \exp \left( \frac{\beta - x}{\lambda} \right) \left( \frac{\beta - x}{\lambda} - 1 \right),$$

$$\frac{\partial}{\partial \beta} \Phi_{\lambda,\beta}(x) = -\exp \left( \frac{\beta - x}{\lambda} \right),$$

$$\frac{\partial^2}{\partial \lambda \partial \beta} \Phi_{\lambda,\beta}(x) = \frac{\beta - x}{\lambda^2} \exp \left( \frac{\beta - x}{\lambda} \right),$$

$$\frac{\partial^2}{\partial \lambda^2} \Phi_{\lambda,\beta}(x) = -\frac{(\beta - x)^2}{\lambda^3} \exp \left( \frac{\beta - x}{\lambda} \right),$$

and

$$\frac{\partial^2}{\partial \beta^2} \Phi_{\lambda,\beta}(x) = -\frac{1}{\lambda} \exp \left( \frac{\beta - x}{\lambda} \right).$$

Therefore,

$$|H(\Phi_{\lambda,\beta})| = \left| \begin{bmatrix} \frac{\partial^2 \Phi_{\lambda,\beta}}{\partial \lambda^2} & \frac{\partial^2 \Phi_{\lambda,\beta}}{\partial \lambda \partial \beta} \\ \frac{\partial^2 \Phi_{\lambda,\beta}}{\partial \lambda \partial \beta} & \frac{\partial^2 \Phi_{\lambda,\beta}}{\partial \beta^2} \end{bmatrix} \right|$$

$$= \exp \left( \frac{2(\beta - x)}{\lambda} \right) (\beta - x)^2 \left( \frac{1}{\lambda^3 \cdot \lambda} - \frac{1}{\lambda^2 \cdot \lambda^2} \right) = 0,$$

thus $\Phi_{\lambda,\beta}$ is never strictly concave in $(\lambda, \beta)^T$. $\qquad\square$

A solution to this is to focus on the form when there is only one multiplier term. We can do the recursive rMLMC method to get an unbiased final estimate, but we can also do the plug-in method (which is biased) to get the same central limit theorem in the KL case as Theorem 13. Doing this not only simplify the code and the numerical analysis, but also keep the same rate of convergence. To be more specific, let $\Phi_\lambda(x) = \exp \left( \frac{-x}{\lambda} \right)$ for $x > 0$ and $\lambda > 0$, the rMLMC estimator first samples $b \sim \mu$, then independently samples a random $N^b = \tilde{N}^b + n_0$, where $n_0$ is a fixed non-negative integer and $\tilde{N}^b \sim \text{Geo}(R)$ with $R \in \left( \frac{1}{2}, \frac{3}{4} \right)$, finally generates $2^{N^b+1}$ unbiased estimates $\left\{ \hat{Z}_i^b \right\}_{1 \leq i \leq 2^{N^b}}$ of $Z^b = E^{P^b}[u(X_T)]$, and the random estimator of $\int_{\mathbb{R}} \Phi_\lambda \left( Z^b \right) d\mu(b)$ is given by $\mathcal{E}_{\text{KL}}^b = \mathcal{E}_{\lambda,\text{KL}}^b = \frac{\Delta_{N^b}^b}{p(N^b)} + \Phi_\lambda \left( \frac{S_{2^{n_0}}^b}{2^{n_0}} \right)$, where $p(.)$ is the probability mass function of $N^b$ and $S_l^b = \sum_{i=1}^{l} \hat{Z}_i^b$. For $N^b \geq n_0$, we define

$$\Delta_{N^b}^b = \Phi_\lambda \left( \frac{S_{2^{N^b+1}}^b}{2^{N^b+1}} \right) - \frac{1}{2} \left( \Phi_\lambda \left( \frac{S_{2^{N^b}}^{O,b}}{2^{N^b}} \right) + \Phi_\lambda \left( \frac{S_{2^{N^b}}^{E,b}}{2^{N^b}} \right) \right),$$

where $S_l^{O,b} = \sum_{i=1}^{l} \hat{Z}_i^{O,b}$ and $S_l^{E,b} = \sum_{i=1}^{l} \hat{Z}_i^{E,b}$ with $\left\{ \hat{Z}_i^{E,b} \right\}_{1 \leq i \leq 2^{N^b+1}}$ and $\left\{ \hat{Z}_i^{O,b} \right\}_{1 \leq i \leq 2^{N^b+1}}$ denotes the estimates indexed by even and odd values, respectively. The estimator $\bar{\mathcal{E}}_{\text{KLpre}}$ for $\int_{\mathbb{R}} \exp \left( \frac{-Z^b}{\lambda} \right) d\mu(b)$ is (for fixed $\pi$ and $\delta$)

$$\mathcal{E}_{\text{KLpre}} = \mathcal{E}_{\pi,\delta,\lambda,\text{KL}} = \frac{1}{n} \sum_{i=1}^{n} \mathcal{E}_{\text{KL}}^{b_i}, \tag{22}$$

given $n$ i.i.d. samples $\{b_i\}_{1 \le i \le n}$ generated from $\mu$. We then use the plug-in estimator $\mathcal{E}_{\text{KL}} = -\lambda\delta - \lambda \log(\mathcal{E}_{\text{KLpre}})$ as a biased but consistent estimator for $-\lambda\delta - \lambda \log\left(\int_{\mathbb{R}} \exp\left(\frac{-Z^b}{\lambda}\right) d\mu(b)\right)$.

Similarly as the general case, we define $\hat{Q}_{\text{DRBCKL}}(\pi) := \sup_{\lambda > 0}\{\mathcal{E}_{\text{KL}}\}$.

The first order condition is given by Theorem 39 and the steps are given by Algorithm 1. Because of the complexity of the derivative information, we will use (stochastic) gradient method in this case. We also provide the theoretical guarantee in Theorem 40. The proofs are given in Appendix E.4.

**Theorem 39.** *The map* $\lambda \mapsto -\lambda\delta - \lambda \log\left(\int_{\mathbb{R}} \exp\left(\frac{-Z^b}{\lambda}\right) d\mu(b)\right)$ *is strictly concave,* $\lim_{n \to \infty} P\left(\mathcal{E}_{KL} \text{ is strictly concave}\right) = 1$, *and the derivative of the map* $\lambda \mapsto \mathcal{E}_{KL}$ *admits the expression*

$$\frac{\partial}{\partial \lambda}\mathcal{E}_{KL} = -\delta - \log\left(\mathcal{E}_{KLpre}\right) \tag{23}$$

$$-\frac{1}{\mathcal{E}_{KLpre}}\frac{1}{n}\sum_{i=1}^{n}\left(A\left(\frac{S_{2^{n_0}}^{b_i}}{2^{n_0}\lambda}\right) + \frac{1}{p(N^{b_i})}\left[A\left(\frac{S_{2^{N^{b_i}+1}}^{b_i}}{2^{N^{b_i}+1}\lambda}\right) - \frac{1}{2}\left(A\left(\frac{S_{2^{N^{b_i}}}^{O,b_i}}{2^{N^{b_i}}\lambda}\right) + A\left(\frac{S_{2^{N^{b_i}}}^{E,b_i}}{2^{N^{b_i}}\lambda}\right)\right)\right]\right),$$

*where* $A(x) = xe^{-x}$ *for* $x > 0$.

**Theorem 40.** *Suppose Assumption 6, 7, 10, and 12 hold,* $\mu$ *is compactly supported with a continuous density. For fixed* $\pi \in \mathcal{A}(x_0)$, *let* $n$ *denote the number of i.i.d. samples of* $\{\mathcal{E}_{KL}^{b_i}\}_{1 \le i \le n}$, *then*

$$\sqrt{n}\left(\hat{Q}_{DRBCKL}(\pi) - Q_{DRBCKL}(\pi)\right) \Rightarrow \mathcal{N}\left(0, \frac{(\lambda^*)^2 \, Var\left(\mathcal{E}_{\lambda^*,KL}^{b_1}\right)}{\left[\int_{\mathbb{R}} \exp\left(\frac{-Z^b}{\lambda^*}\right) d\mu(b)\right]^2}\right),$$

*where* $\lambda^*$ *is defined in Section 3.*

---

**Algorithm 1** rMLMC DRBC Policy Evaluation Step in the KL Divergence Case

---

**Input:** A simulator $\mathcal{S}$, rMLMC parameter $R \in (\frac{1}{2}, \frac{3}{4})$, prior distribution $\mu$, parameter $n_0$, initialization of $\lambda$ and $k = 0$, policy $\pi$, step-size sequence $\{\alpha_k : k \in \mathbb{Z}_{\ge 0}\}$.
**Output:** Estimator of rMLMC DRBC policy value $\hat{Q}_{\text{DRBCKL}}(\pi)$.
**repeat**

- Draw $n$ i.i.d. samples $\{b_i\}_{1 \le i \le n}$ from $\mu$. For each $i = 1, 2, \ldots, n$, sample $\tilde{N}^{b_i} \sim \text{Geo}(R)$ independently, and compute $N^{b_i} = \tilde{N}^{b_i} + n_0$, then give $b_i$ to $\mathcal{S}$ and generate $2^{N^{b_i}+1}$ i.i.d. samples of $\hat{Z}_i^b$.
- Compute $\mathcal{E}_{\text{KLpre}}$, $\mathcal{E}_{\text{KL}}$, and GF $:= \frac{\partial}{\partial \lambda}\mathcal{E}_{\text{KL}}$ (Theorem 39).
- Update $\lambda = \lambda + \alpha_k \text{GF}$ and update $k = k + 1$.

**until** $\lambda$ converges.
**Return** $\hat{Q}_{\text{DRBCKL}}(\pi) = \mathcal{E}_{\text{KL}}$ (Equation (22)).

---

### E.3 PROOF OF THEOREM 13

*Proof.* The proof is adapted from the proof of Theorem 1 from Si et al. (2023). Since the estimator (7) is unbiased and has finite variance under the assumptions, then from (the classical) central limit theorem,

$$\sqrt{n}\left(\mathcal{E}_{\pi,\delta,\lambda,\beta} - \int_{\mathbb{R}} \Phi_{\lambda,\beta}\left(Z^b\right) d\mu(b)\right) \Rightarrow \mathcal{N}\left(0, \text{Var}\left(\mathcal{E}_{\lambda,\beta}^{b_1}\right)\right). \tag{24}$$

From Assumption 4, there exists $lb, ub \in \mathbb{R}$ such that $\left\|(\lambda^*, \beta^*)^T\right\|_2^2 \in K = [lb, ub]$. Without loss of generality, we abuse the notation and denote the support of optimizers as $K$. Since the function $(\lambda, \beta)^T \mapsto \mathcal{E}_{\pi,\delta,\lambda,\beta}$ over $K$ is Lipschitz continuous (continuously differentiable on a compact set), then

$$\sqrt{n}\left(f(.) - g(.)\right) \Rightarrow L(.) \tag{25}$$

uniformly in the Banach space $C(K; \mathbb{R})$ of continuous functions $\varphi : K \to \mathbb{R}$ equipped with the sup norm, where $f((\lambda, \beta)) = \mathcal{E}_{\pi, \delta, \lambda, \beta}$, $g((\lambda, \beta)) = \int_{\mathbb{R}} \Phi_{\lambda, \beta}(Z^b) d\mu(b)$, and $L((\lambda, \beta)) \sim \mathcal{N}\left(0, \text{Var}\left(\mathcal{E}_{\lambda, \beta}^{b_1}\right)\right)$. $L$ is a random element taking values in $C(K; \mathbb{R})$.

Define the functionals

$$G\left(\varphi, (\lambda, \beta)^T\right) = -\beta + \lambda\delta - \varphi(\lambda, \beta)$$

and

$$V(\varphi) = \inf_{(\lambda, \beta)^T \in K} G\left(\varphi, (\lambda, \beta)\right),$$

then by Danskin theorem, $V$ is directionally differentiable at any $\mu \in C(K; \mathbb{R})$ with $\mu > 0$ and for any $\nu \in C(K; \mathbb{R})$,

$$V'_\mu(\nu) = \inf_{(\lambda, \beta)^T \in \bar{X}(\mu)} -\nu(\lambda, \beta),$$

where $\bar{X}(\mu) = \arg\min_{(\lambda, \beta)^T \in K} \{-\beta + \lambda\delta - \mu(\lambda, \beta)\}$ and $V'_\mu(\nu)$ is the directional derivative of $V$ at $\mu$ in the direction of $\nu$. In addition, $V$ is Hadamard directionally differentiable at $g(.)$ since

$$g((\lambda, \beta)) = \int_{\mathbb{R}} \Phi_{\lambda, \beta}\left(E^{P^b}[u(X_T)]\right) d\mu(b) \geq \int_{\mathbb{R}} \Phi_{\lambda, \beta}\left(u(m)\right) d\mu(b) > 0 \tag{26}$$

and $V(\varphi)$ is Lipschitz continuous if $\varphi$ is bounded below away from zero.

Therefore, by the Delta Theorem, we have

$$\sqrt{n}\left(V(f(.)) - V(g(.))\right) \Rightarrow V'_{g(.)}(L).$$

Since the map $(\lambda, \beta)^T \mapsto -\beta + \lambda\delta - \int_{\mathbb{R}} \Phi_{\lambda, \beta}(Z^b) d\mu(b)$ is strictly convex, then we have the uniqueness of the optimizer and

$$V'_{g(.)}(L) = -L(\lambda^*, \beta^*) \sim \mathcal{N}\left(0, \text{Var}\left(\mathcal{E}_{\lambda^*, \beta^*}^{b_1}\right)\right).$$

Recall that

$$Q_{\text{DRBC}}(\pi) = -\inf_{\substack{\lambda > 0 \\ \beta \in \mathbb{R}}} \left\{-\beta + \lambda\delta - \int_{\mathbb{R}} \Phi_{\lambda, \beta}(Z^b) d\mu(b)\right\} = -V(g(.))$$

and

$$\hat{Q}_{\text{DRBC}}(\pi) = -\inf_{\substack{\lambda > 0 \\ \beta \in \mathbb{R}}} \left\{-\beta + \lambda\delta - \mathcal{E}_{\pi, \delta, \lambda, \beta}\right\}.$$

Notice that from Assumption 12 we have

$$\lim_{n \to \infty} P\left(\hat{Q}_{\text{DRBC}}(\pi) \neq -V(f(.))\right) \to 0. \tag{27}$$

Then from Slutsky's theorem,

$$\sqrt{n}\left(\hat{Q}_{\text{DRBC}}(\pi) - Q_{\text{DRBC}}(\pi)\right) = \sqrt{n}\left(\hat{Q}_{\text{DRBC}}(\pi) + V(f(\lambda, \beta))\right) + \sqrt{n}\left(V(f(\lambda, \beta)) - V(g(\lambda, \beta))\right)$$

$$\Rightarrow 0 + \mathcal{N}\left(0, \text{Var}\left(\mathcal{E}_{\lambda^*, \beta^*}^{b_1}\right)\right) \sim \mathcal{N}\left(0, \text{Var}\left(\mathcal{E}_{\lambda^*, \beta^*}^{b_1}\right)\right),$$

which finishes the proof. $\qquad\square$

### E.4 PROOF OF RESULTS FOR THE KL DIVERGENCE CASE

#### E.4.1 PROOF OF THEOREM 39

*Proof.* Since $\mathcal{E}_{\text{KL}}$ is a consistent estimator of $-\lambda\delta - \lambda\log\left(\int_{\mathbb{R}} \exp\left(\frac{-Z^b}{\lambda}\right) d\mu(b)\right)$, it suffices to show the strict concavity of $g(\lambda) = -\lambda\log\left(\int_{\mathbb{R}} \exp\left(\frac{-Z^b}{\lambda}\right) d\mu(b)\right)$. To do this, we compute the first and second order derivatives:

$$\frac{\partial}{\partial\lambda} g(\lambda) = -\log\left(\int_{\mathbb{R}} e^{-Z^b/\lambda} d\mu(b)\right) - \frac{\int_{\mathbb{R}} Z^b e^{-Z^b/\lambda} d\mu(b)}{\lambda \int_{\mathbb{R}} e^{-Z^b/\lambda} d\mu(b)}$$

and

$$\frac{\partial^2}{\partial \lambda^2} g(\lambda) = -\frac{\int_{\mathbb{R}} (Z^b)^2 e^{-Z^b/\lambda} \, d\mu(b)}{\lambda^3 \int_{\mathbb{R}} e^{-Z^b/\lambda} \, d\mu(b)} + \frac{\left( \int_{\mathbb{R}} Z^b e^{-Z^b/\lambda} \, d\mu(b) \right)^2}{\lambda^3 \left( \int_{\mathbb{R}} e^{-Z^b/\lambda} \, d\mu(b) \right)^2}$$

$$= \frac{1}{\lambda^3 \left( \int_{\mathbb{R}} e^{-Z^b/\lambda} \, d\mu(b) \right)^2} \left( \left( \int_{\mathbb{R}} Z^b e^{-Z^b/\lambda} \, d\mu(b) \right)^2 - \int_{\mathbb{R}} (Z^b)^2 e^{-Z^b/\lambda} \, d\mu(b) \int_{\mathbb{R}} e^{-Z^b/\lambda} \, d\mu(b) \right)$$

$$\leq 0,$$

where from Cauchy Schwartz inequality, the equality holds if and only if $Z^B$ is constant almost surely, which is not true in our case. Therefore, strict concavity is shown. Then the computations of the derivatives are elementary.

$\square$

### E.4.2 PROOF OF THEOREM 40

*Proof.* The proof is similar to the proof of Theorem 13; the only difference is that we will have more explicit forms of the limiting Gaussian variance. Since the function $\Phi_\lambda(x) = \exp\left(\frac{-x}{\lambda}\right)$ satisfies Assumption 5, together with other assumptions, it is easy to see that $\mathcal{E}_{\text{KLpre}}$ is an unbiased estimator for $\int_{\mathbb{R}} \exp\left(\frac{-Z^b}{\lambda}\right) d\mu(b)$ and has a finite variance. Therefore, from central limit theorem,

$$\sqrt{n} \left( \mathcal{E}_{\text{KLpre}} - \int_{\mathbb{R}} \exp\left(\frac{-Z^b}{\lambda}\right) d\mu(b) \right) \Rightarrow \mathcal{N}\left( 0, \text{Var}\left( \mathcal{E}_{\lambda,\text{KL}}^{b_1} \right) \right). \tag{28}$$

Recall that from Theorem 31, the optimal $\lambda^* \in K$, where $K = [\underline{\lambda}, \bar{\lambda}] \subset \mathbb{R}$ is compact. From the Lipschitz continuity of $\lambda \mapsto \mathcal{E}_{\text{KLpre}}$ over $K$, we have

$$\sqrt{n} \left( f(.) - g(.) \right) \Rightarrow L(.) \tag{29}$$

uniformly in the Banach space $C(K; \mathbb{R})$ of continuous functions $\varphi : K \to \mathbb{R}$ equipped with the sup norm, where $f(\lambda) = \mathcal{E}_{\text{KLpre}}$, $g(\lambda) = \int_{\mathbb{R}} \exp\left(\frac{-Z^b}{\lambda}\right) d\mu(b)$, and $L(\lambda) \sim \mathcal{N}\left( 0, \text{Var}\left( \mathcal{E}_{\lambda,\text{KL}}^{b_1} \right) \right)$. $L$ is a random element taking values in $C(K; \mathbb{R})$. We next define the functionals

$$G(\varphi, \lambda) = \lambda \log(\varphi(\lambda)) + \lambda \delta$$

and

$$V(\varphi) = \inf_{\lambda \in K} G(\varphi, \lambda),$$

then by Danskin theorem, $V$ is directionally differentiable at any $\mu \in C(K; \mathbb{R})$ with $\mu > 0$ and for any $\nu \in C(K; \mathbb{R})$,

$$V'_\mu(\nu) = \inf_{\lambda \in \bar{X}(\mu)} \lambda \left( \frac{1}{\mu(\lambda)} \right) \nu(\lambda),$$

where $\bar{X}(\mu) = \arg\min_{\lambda \in K} \{ \lambda \log(\mu(\lambda)) + \lambda \delta \}$ and $V'_\mu(\nu)$ is the directional derivative of $V$ at $\mu$ in the direction of $\nu$. In addition, $V$ is Hadamard directionally differentiable at $g(.)$ since (Assumption 6)

$$g(\lambda) = \int_{\mathbb{R}} \exp\left(\frac{-Z^b}{\lambda}\right) d\mu(b) > 0 \tag{30}$$

and $V(\varphi)$ is Lipschitz continuous if $\varphi$ is bounded below away from zero.

Therefore, by the Delta Theorem, we have

$$\sqrt{n} \left( V(f(.)) - V(g(.)) \right) \Rightarrow V'_{g(.)}(L).$$

Since the map $\lambda \mapsto \int_{\mathbb{R}} \exp\left(\frac{-Z^b}{\lambda}\right) d\mu(b)$ is strictly convex, then we have the uniqueness of the optimizer and

$$V'_{g(.)}(L) = \lambda^* \left(\frac{1}{\int_{\mathbb{R}} \exp\left(\frac{-Z^b}{\lambda^*}\right) d\mu(b)}\right) L(\lambda^*)$$

$$\sim \left(\frac{\lambda^*}{\int_{\mathbb{R}} \exp\left(\frac{-Z^b}{\lambda^*}\right) d\mu(b)}\right) \mathcal{N}\left(0, \mathrm{Var}\left(\mathcal{E}^{b_1}_{\lambda^*,\mathrm{KL}}\right)\right)$$

$$\sim \mathcal{N}\left(0, \frac{(\lambda^*)^2 \, \mathrm{Var}\left(\mathcal{E}^{b_1}_{\lambda^*,\mathrm{KL}}\right)}{\left[\int_{\mathbb{R}} \exp\left(\frac{-Z^b}{\lambda^*}\right) d\mu(b)\right]^2}\right).$$

Recall that

$$Q_{\mathrm{DRBCKL}}(\pi) = -\inf_{\lambda>0}\left\{\lambda\delta + \lambda \int_{\mathbb{R}} \exp\left(\frac{-Z^b}{\lambda^*}\right) d\mu(b)\right\} = -V(g(.))$$

and

$$\hat{Q}_{\mathrm{DRBCKL}}(\pi) = -\inf_{\lambda>0}\left\{\mathcal{E}_{\mathrm{KL}}\right\}.$$

Notice that from Assumption 12 we have

$$\lim_{n\to\infty} P\left(\hat{Q}_{\mathrm{DRBCKL}}(\pi) \neq -V(f(.))\right) \to 0. \tag{31}$$

Then from Slutsky's theorem,

$$\sqrt{n}\left(\hat{Q}_{\mathrm{DRBCKL}}(\pi) - Q_{\mathrm{DRBCKL}}(\pi)\right) = \sqrt{n}\left(\hat{Q}_{\mathrm{DRBCKL}}(\pi) + V(f(\lambda))\right) + \sqrt{n}\left(V(f(\lambda)) - V(g(\lambda))\right)$$

$$\Rightarrow 0 + \mathcal{N}\left(0, \frac{(\lambda^*)^2 \, \mathrm{Var}\left(\mathcal{E}^{b_1}_{\lambda^*,\mathrm{KL}}\right)}{\left[\int_{\mathbb{R}} \exp\left(\frac{-Z^b}{\lambda^*}\right) d\mu(b)\right]^2}\right) \sim \mathcal{N}\left(0, \frac{(\lambda^*)^2 \, \mathrm{Var}\left(\mathcal{E}^{b_1}_{\lambda^*,\mathrm{KL}}\right)}{\left[\int_{\mathbb{R}} \exp\left(\frac{-Z^b}{\lambda^*}\right) d\mu(b)\right]^2}\right),$$

which finishes the proof. $\square$

# F  PROOF OF RESULTS IN SECTION 5

## F.1  PROOF OF LEMMA 14

*Proof.* The proof is the same as the proof of Theorem 1. $\square$

## F.2  PROOF OF THEOREM 15

*Proof.* If we denote $\gamma = \frac{1}{\lambda}$, then from a dual representation of the entropic risk measure (Föllmer & Schied, 2016),

$$\sup_{\pi\in\mathcal{A}(x_0)}\left\{-\lambda\log\left(\int_{\mathbb{R}} \exp\left(\frac{-Z^b}{\lambda}\right) d\mu(b)\right)\right\}$$

$$= \sup_{\pi\in\mathcal{A}(x_0)}\left\{-\frac{1}{\gamma}\log\left(\int_{\mathbb{R}} \exp\left(-\gamma E^{P^b}[u(X_T)]\right) d\mu(b)\right)\right\}$$

$$= \sup_{\pi\in\mathcal{A}(x_0)}\left\{-\sup_{\tilde{Q}\in\mathcal{AC}}\left\{E^{\tilde{Q}}\left[-E^{\tilde{Q}^b}[u(X_T)]\right] - \frac{1}{\gamma}D_{\mathrm{KL}}\left(\tilde{Q}\parallel P\right)\right\}\right\}$$

$$= \sup_{\pi\in\mathcal{A}(x_0)}\left\{\inf_{\tilde{Q}\in\mathcal{AC}}\left\{E^{\tilde{Q}}\left[E^{\tilde{Q}^b}[u(X_T)]\right] + \frac{1}{\gamma}D_{\mathrm{KL}}\left(\tilde{Q}\parallel P\right)\right\}\right\}$$

$$= \sup_{\pi\in\mathcal{A}(x_0)}\inf_{\tilde{Q}\in\mathcal{AC}}\left\{E^{\tilde{Q}}[u(X_T)] + \lambda D_{\mathrm{KL}}\left(\tilde{Q}\parallel P\right)\right\}.$$

$\square$

### F.3 PROOF OF THEOREM 16

*Proof.* In order to use Sion's minmax theorem, we need the compactness of $\mathcal{AC}$, which is not true in general. However, since $m = \operatorname{essinf} X_T$ exists and $P(X_T = m) > 0$, we denote the set where $X_T = m$ as $E$ and define a measure such that for any measurable set $A$,

$$Q(A) = \frac{P(A \cap E)}{P(E)},$$

then $Q \in \mathcal{AC}$ and for any $\tilde{Q} \in \mathcal{AC}$,

$$E^{\tilde{Q}}[u(X_T)] \geq u(m) = E^Q[u(X_T)],$$

thus $Q$ is an optimizer of $\inf_{\tilde{Q} \in \mathcal{AC}} E^{\tilde{Q}}[u(X_T)]$. Moreover,

$$\frac{dQ}{dP} = \frac{\mathbf{1}_E}{P(E)},$$

thus for a fixed integer $p > 1$, $\left\| \frac{dQ}{dP} \right\|_p \leq \frac{1}{P(E)}$, thus the optimal $Q$ for $\inf_{\tilde{Q} \in \mathcal{AC}} E^{\tilde{Q}}[u(X_T)]$ is attained inside a compact set $K_1$ in the topology of weak convergence by Prokhorov's theorem. Note that the proof of the above argument does not use the finite support assumption.

On the other hand, since $\mu$ is finitely supported, and for any $\tilde{Q} \in \mathcal{AC}$, there exists $\nu \ll \mu$ such that $\frac{d\tilde{Q}}{dP} = \frac{d\nu}{d\mu}(B)$, then $\nu$ is also finitely supported and $\tilde{Q} \in \mathcal{P}(K_0)$, where $K_0$ is compact. Thus we can also restrict the choice of $Q$ to a weakly compact set $K_2$ and define $K = K_1 \cap K_2$, thus $K$ is compact. As a conclusion, $\inf_{\tilde{Q} \in \mathcal{AC}} E^{\tilde{Q}}[u(X_T)] = \inf_{\tilde{Q} \in K} E^{\tilde{Q}}[u(X_T)]$.

In addition, we notice that the mapping $\tilde{Q} \mapsto E^{\tilde{Q}}[u(X_T)] + \lambda D_{\mathrm{KL}}\left(\tilde{Q} \parallel P\right)$ is convex and weakly continuous since the first part is linear and the second part is convex, and the mapping $\pi \mapsto E^{\tilde{Q}}[u(X_T)] + \lambda D_{\mathrm{KL}}\left(\tilde{Q} \parallel P\right)$ is concave, and the constraint sets are both convex, then from Sion's minmax theorem,

$$\sup_{\pi \in \mathcal{A}(x_0)} \inf_{\tilde{Q} \in \mathcal{AC}} \left\{ E^{\tilde{Q}}\left[u(X_T^\pi)\right] + \lambda D_{\mathrm{KL}}\left(\tilde{Q} \parallel P\right) \right\}$$

$$= \sup_{\pi \in \mathcal{A}(x_0)} \inf_{\tilde{Q} \in K} \left\{ E^{\tilde{Q}}\left[u(X_T^\pi)\right] + \lambda D_{\mathrm{KL}}\left(\tilde{Q} \parallel P\right) \right\}$$

$$= \inf_{\tilde{Q} \in K} \sup_{\pi \in \mathcal{A}(x_0)} \left\{ E^{\tilde{Q}}\left[u(X_T^\pi)\right] + \lambda D_{\mathrm{KL}}\left(\tilde{Q} \parallel P\right) \right\}$$

$$= \inf_{\tilde{Q} \in \mathcal{AC}} \sup_{\pi \in \mathcal{A}(x_0)} \left\{ E^{\tilde{Q}}\left[u(X_T^\pi)\right] + \lambda D_{\mathrm{KL}}\left(\tilde{Q} \parallel P\right) \right\},$$

where the last equality holds since if we assume the existence of optimal $\pi^*$ for the inner supremum problem, then for a fixed control, the optimal $\tilde{Q}^*$ is taken in a weakly compact set. Finally, since $\mu$ is finitely supported, then the optimizers are attained from this compactness, which finishes the proof. $\qquad\square$

**Theorem 41.** *Problem (9) is convex on the set $\mathcal{AC}$ for any prior distribution $\mu$.*

*Proof.* Since for fixed $P$, $D_{\mathrm{KL}}(Q \parallel P)$ is convex in $Q$, then it is enough to show that $\tilde{V}$ is convex, which follows from the fact that convexity are preserved under linear transform and it is also preserved under composition between a convex function and a convex and nondecreasing function. $\quad\square$

### F.4 ALTERNATE ADMISSIBLE CONTROLS

In this section, we give the definition of the admissible controls that will help the tractability of the general case.

To illustrate our definition of the admissible set, we go back to Problem (3) with the strictly concave utility function $u$. (For a full review, see Appendix B.2) It is shown that the optimal terminal

wealth $X_T^*$ is given by $X_T^* = I\left(\frac{\mathcal{K}(x_0)e^{-rT}}{F(T,Y(T))}\right)$ (Karatzas & Zhao, 1998). For a density function $\lambda : \mathbb{R} \to \mathbb{R}$, we define the inner product $\left\langle E^{P^\cdot}\left[u(X_T^*)\right], \lambda\right\rangle := \int_{\mathbb{R}} E^{P^b}\left[u(X_T^*)\right]\lambda(b)d\mu(b) = \int_{\mathbb{R}} E^{P^b}\left[(u \circ I)\left(\frac{\mathcal{K}(x_0)e^{-rT}}{F(T,Y(T))}\right)\right]\lambda(b)d\mu(b)$. We want to see whether there exists a function $h : \mathbb{R} \to \mathbb{R}$ and a functional $\rho : L^1 \to \mathbb{R}$ such that

$$\int_{\mathbb{R}} E^{P^b}\left[u(X_T^*)\right]\lambda(b)d\mu(b) = h(x_0)\rho(\lambda) \tag{32}$$

and

$$h(1) = u(e^{rT}). \tag{33}$$

**Theorem 42.** *For the following cases (1) and (2), the conditions (32) and (33) are met, where for case (3), the conditions are not met:*

- *(1) $u(x) = \frac{1}{\alpha}x^\alpha$, where $\alpha < 1$ and $\alpha \neq 0$.*

- *(2) $u(x) = \frac{-1}{\gamma}e^{-\gamma x}$, where $\gamma > 0$.*

- *(3) $u(x) = \log(x)$.*

*Proof.* • (1) Here,
$$I(y) = y^{\frac{1}{1-\alpha}},$$

thus
$$(u \circ I)(y) = \frac{1}{\alpha}y^{\frac{\alpha}{1-\alpha}},$$

which implies that

$$\int_{\mathbb{R}} E^{P^b}\left[u(X_T^*)\right]\lambda(b)d\mu(b) = \int_{\mathbb{R}} E^{P^b}\left[(u \circ I)\left(\frac{\mathcal{K}(x_0)e^{-rT}}{F(T,Y(T))}\right)\right]\lambda(b)d\mu(b)$$

$$= \int_{\mathbb{R}} E^{P^b}\left[\frac{1}{\alpha}\left(\frac{\mathcal{K}(x_0)e^{-rT}}{F(T,Y(T))}\right)^{\frac{\alpha}{1-\alpha}}\right]\lambda(b)d\mu(b)$$

$$= \frac{1}{\alpha}\left(\mathcal{K}(x_0)e^{-rT}\right)^{\frac{\alpha}{1-\alpha}}\int_{\mathbb{R}} E^{P^b}\left[(F(T,Y(T)))^{\frac{\alpha}{\alpha-1}}\right]\lambda(b)d\mu(b)$$

$$= h(x_0)\rho(\lambda),$$

where
$$h(x) = \frac{1}{\alpha}\left(\mathcal{K}(x)e^{-rT}\right)^{\frac{\alpha}{1-\alpha}}$$

and
$$\rho(\lambda) = \int_{\mathbb{R}} E^{P^b}\left[(F(T,Y(T)))^{\frac{\alpha}{\alpha-1}}\right]\lambda(b)d\mu(b).$$

- (2) Here,
$$I(y) = \frac{-1}{\gamma}\log(y),$$

thus
$$(u \circ I)(y) = \frac{-1}{\gamma}y,$$

which implies that

$$\int_{\mathbb{R}} E^{P^b}\left[u(X_T^*)\right]\lambda(b)d\mu(b) = \int_{\mathbb{R}} E^{P^b}\left[(u \circ I)\left(\frac{\mathcal{K}(x_0)e^{-rT}}{F(T,Y(T))}\right)\right]\lambda(b)d\mu(b)$$

$$= \int_{\mathbb{R}} E^{P^b}\left[\frac{-1}{\gamma}\left(\frac{\mathcal{K}(x_0)e^{-rT}}{F(T,Y(T))}\right)\right]\lambda(b)d\mu(b)$$

$$= h(x_0)\rho(\lambda),$$

where

$$h(x) = \frac{-1}{\gamma} \mathcal{K}(x_0) e^{-rT}$$

and

$$\rho(\lambda) = \int_{\mathbb{R}} E^{P^b} \left[ \frac{1}{F(T, Y(T))} \right] \lambda(b) d\mu(b).$$

- (3) Here,

$$I(y) = \frac{1}{y} \text{ and } \mathcal{K}(x_0) = \frac{1}{x_0},$$

thus

$$(u \circ I)(y) = -\log(y),$$

which implies that

$$\begin{aligned}
\int_{\mathbb{R}} E^{P^b} \left[ u(X_T^*) \right] \lambda(b) d\mu(b) &= \int_{\mathbb{R}} E^{P^b} \left[ (u \circ I) \left( \frac{\mathcal{K}(x_0) e^{-rT}}{F(T, Y(T))} \right) \right] \lambda(b) d\mu(b) \\
&= \int_{\mathbb{R}} E^{P^b} \left[ -\log \left( \frac{\mathcal{K}(x_0) e^{-rT}}{F(T, Y(T))} \right) \right] \lambda(b) d\mu(b) \\
&= x_0 + rT + \int_{\mathbb{R}} E^{P^b} \left[ \log \left( F(T, Y(T)) \right) \right] \lambda(b) d\mu(b),
\end{aligned}$$

which shows that the decomposition is impossible.

$\square$

*Remark* 43. In general, if $u \circ I$ satisfies

$$u \circ I \left( \frac{a}{b} \right) = f(a) g(b),$$

then the decomposition condition holds. There are other examples satisfying this condition.

Now we define a subset $\tilde{\mathcal{A}}(x_0) \subset \mathcal{A}(x_0)$ such that for all $\pi \in \tilde{\mathcal{A}}(x_0)$, there exists corresponding function $h : \mathbb{R} \to \mathbb{R}$ and a functional $\rho : L^1 \to \mathbb{R}$ such that for the controlled terminal wealth $X_T^\pi$, $\int_{\mathbb{R}} E^{P^b} \left[ u(X_T^\pi) \right] \lambda(b) d\mu(b) = h(x_0) \rho(\lambda)$ and $h(1) = u(e^{rT})$. Thus for some utility functions, Problem (3) is equivalent to $\sup_{\pi \in \tilde{\mathcal{A}}(x_0)} E^P \left[ u(X_T) \right]$. Besides, if we define a function $\tilde{u}(x) = u(x) + C$, where $C$ is a constant that does not depend on $x$, then the problems $\sup_{\pi \in \tilde{\mathcal{A}}(x_0)} E^P \left[ u(X_T) \right]$ and $\sup_{\pi \in \tilde{\mathcal{A}}(x_0)} E^P \left[ \tilde{u}(X_T) \right]$ should have the same optimal solution (not the same value function), at least in the definition of the classical problem. However this is not true if we follow our current definition of admissible controls since the decomposition cannot be satisfied by these two problems at the same time.

To rescue this, we firstly define an equivalence relation for two real-valued function $f$ and $g$

$$f \sim g \iff \text{ there exists a constant } C \text{ such that } f(x) = g(x) + C.$$

We call $\tilde{\mathcal{A}}(x_0)$ the collection of all alternate admissible controls $\pi$, which is a subset of $\mathcal{A}(x_0)$ such that there exists a function $v \sim u$ ($u$ is the utility function in the objective function) such that there exist corresponding function $h : \mathbb{R} \to \mathbb{R}$ and a functional $\rho : L^1 \to \mathbb{R}$ such that for the controlled terminal wealth $X_T^\pi$, $\int_{\mathbb{R}} E^{P^b} \left[ v(X_T^\pi) \right] \lambda(b) d\mu(b) = h(x_0) \rho(\lambda)$ and $h(1) = v(e^{rT})$.

In the DRBC formulation, since the deviation from the prior distribution and its corresponding underlying probability space is small, it is reasonable to continue to search for optimal solutions in the space of $\tilde{\mathcal{A}}(x_0)$ for those utilities. Thus we call $\tilde{\mathcal{A}}(x_0)$ the collection of all alternate admissible controls and Problem (10) becomes

$$\sup_{\pi \in \tilde{\mathcal{A}}(x_0)} \int_{\mathbb{R}} \Phi_{\lambda, \beta} \left( E^{P^b} \left[ u(X_T) \right] \right) d\mu(b). \tag{34}$$

## F.5 DISCUSSION ON CLOSED-FORM COMPUTATIONS

In this section, we review how the closed-form computation for a smooth ambiguity problem is derived in some special cases, and how does this method fail in our case.

Recall that the DRBC problem becomes

$$
\sup_{\pi \in \tilde{\mathcal{A}}(x_0)} \inf_{Q \in \mathcal{U}_\delta} E^Q \left[ u(X_T) \right] = \sup_{\pi \in \tilde{\mathcal{A}}(x_0)} \sup_{\substack{\lambda > 0 \\ \beta \in \mathbb{R}}} \left\{ \beta - \lambda \delta + \int_{\mathbb{R}} \Phi_{\lambda, \beta} \left( E^{P^b} [u(X_T^\pi)] \right) d\mu(b) \right\}.
$$

For simplicity, suppose there exists a unique pair $(\pi^*, \lambda^*, \beta^*)$ such that

$$
\sup_{\pi \in \tilde{\mathcal{A}}(x_0)} \inf_{Q \in \mathcal{U}_\delta} E^Q \left[ u(X_T) \right] = \beta^* - \lambda^* \delta + \int_{\mathbb{R}} \Phi_{\lambda^*, \beta^*} \left( E^{P^b} [u(X_T^{\pi^*})] \right) d\mu(b),
$$

thus it suffices to solve the smooth ambiguity problem with strictly concave and strictly increasing function $\Phi_{\lambda^*, \beta^*}$:

$$
\sup_{\pi \in \tilde{\mathcal{A}}(x_0)} \left\{ \int_{\mathbb{R}} \Phi_{\lambda^*, \beta^*} \left( E^{P^b} [u(X_T^\pi)] \right) d\mu(b) \right\}. \tag{35}
$$

We remark that this unique pair assumption is hard to check because of the jointly non-convexity in terms of $\pi$ and $(\lambda, \beta)^T$. For notational convenience, we use $(\lambda, \beta)^T$ in the rest of this section instead of $(\lambda^*, \beta^*)^T$.

Therefore, the duality theory in Guan et al. (2022) can be applied (for a brief review, see Section 5.2) and we derive the conditions to derive the optimal terminal wealth: From Corollary 3.8 in Guan et al. (2022), if $X_T$ is a terminal wealth such that there exists a constant $\kappa > 0$ with

$$
\begin{cases} u'(X_T) = \dfrac{\kappa \eta_T \int_{\mathbb{R}} \Phi'_{\lambda, \beta} \left( E^{Q^b} [u(X_T)] \right) d\mu(b)}{\int_{\mathbb{R}} \Phi'_{\lambda, \beta} \left( E^{Q^b} [u(X_T)] \right) \eta_T^b \, d\mu(b)} \\ E^{Q^r} [X_T] = x_0 e^{rT}, \end{cases} \tag{36}
$$

then $X_T$ is the optimal terminal wealth corresponding to Problem (35). In Guan et al. (2022), a closed-form solution is provided when the nonlinear transformation function (Problem (35) with $\Phi_{\lambda, \beta}$ changed to $\phi$) is

$$
\phi(x) = \begin{cases} \frac{x^\gamma}{\gamma}, & \text{if } x \geq 0, \\ \frac{-(-x)^\gamma}{\gamma}, & \text{if } x < 0, \end{cases} \quad \text{where } \gamma \in (0, 1) \text{ is a fixed constant.}
$$

With the nonlinear function $\phi$ and the utility function $u(x) = \frac{1}{\alpha} x^\alpha$ with $\alpha \in (0, 1)$, we guess that the optimal terminal wealth $X_T^*$ has the form ($p$, $q$, and $c$ are parameters to solve later)

$$
X_T^* = \exp \left( \frac{1}{\alpha} \left( \frac{p}{2T} \hat{W}_T^2 + q \hat{W}_T + c \right) \right).
$$

Hence,

$$
E^{Q^b} [u(X_T^*)] = E^{\hat{Q}} \left[ \frac{1}{\alpha} X_T^{* \alpha} \eta_T^b \right]
$$

$$
= \frac{1}{\alpha \sqrt{1-p}} \exp \left( \frac{pT}{2(1-p)} \nu_b^2 - \frac{qT}{1-p} \nu_b + \frac{q^2 T}{2(1-p)} + c \right).
$$

Therefore, if we assume $\mu \sim \mathcal{N}(\mu_0, \sigma_0^2)$, then

$$
\int_{\mathbb{R}} \phi' \left( E^{Q^b} [u(X_T^*)] \right) \eta_T^b d\mu(b) \tag{37}
$$

$$
\propto \int_{\mathbb{R}} \exp \left( \frac{(\gamma - 1)}{2} \frac{pT}{1-p} \nu_b^2 - (\gamma - 1) \frac{qT}{1-p} \nu_b - \frac{T}{2} \nu_b^2 - \hat{W}_T \nu_b - \frac{\sigma_0^2 T}{2} \nu_b^2 \right) d\nu_b
$$

$$
\propto \exp \left( \frac{1}{2T \left( \frac{1-\gamma p}{1-p} + \sigma_0^2 \right)} \left( \hat{W}_T^2 + \frac{2(\gamma - 1) T q}{1-p} \hat{W}_T \right) \right).
$$

Substituting Equation (37) into Equation (36), we obtain the following equations

$$
\begin{cases}
(\gamma + \sigma_0^2)p^2 - \left(\sigma_0^2 + \frac{1}{1-\alpha}\right)p + \frac{\alpha}{1-\alpha} = 0, \\
\frac{(1-\alpha)(1-\gamma p)}{\alpha(1-p)}q = \nu, \\
\exp\left(\frac{\left(\nu - \frac{q}{\alpha}\right)^2}{2\left(1 - \frac{p}{\alpha}\right)T} - \frac{1}{2}\nu^2 T + \frac{c}{\alpha}\right) = x_0 e^{rT},
\end{cases}
$$

which admits a solution of pair $(p, q, c)$ if $p < 1$.

The essential part of the above computation is that when we plug in the composition of power and exponential functions, the result is still exponential and this fits well with the computation of Gaussian integrals. However, in our case, the nonlinear transformation is defined by for fixed $\lambda \geq 0, \beta \in \mathbb{R}$,

$$
\Phi_{\lambda,\beta}(x) := -(\lambda\phi)^*(\beta - x).
$$

Thus, even though we use the $\phi$-divergence which induces a power-type $\Phi_{\lambda,\beta}$ (e.g. the Cressie-Read divergence), the above computation cannot be adapted unless the utility function $\tilde{u} = u + C(\lambda, \beta)$ (we call it an artificial utility) contains the information of the optimal $\lambda$ and $\beta$, which is highly non-practical and it seems that this is only true in theory.

However, the validity of this artificial utility is not guaranteed. To begin with, note that we cannot simply regard $\lambda$ and $\beta$ as constants and thus $u$ and $\tilde{u}$ are not equivalent. Thus, even though $u$ is a good utility function with the alternative admissible controls (Theorem 42), we do not know whether $\tilde{u}$ is also a good one. Thus, the method may not apply in this case.

### F.6 PROPERTIES OF $Q^b$

**Theorem 44.** *Under $Q^b$ for any $b \in \mathbb{R}$, the distribution of $B$ is still $\mu$.*

*Proof.* Let $A \in \mathcal{F}$, then from independence of $B$ and $W$, we have

$$
\mu_b(A) = Q^b(B \in A) = E^{Q^b}\left[\mathbf{1}_{\{B \in A\}}\right]
$$

$$
= E^P\left[\frac{dQ^b}{dP}\mathbf{1}_{\{B \in A\}}\right] = E^P\left[\exp\left(-\frac{B - b}{\sigma}W_T - \frac{(B - b)^2}{2\sigma^2}T\right)\mathbf{1}_{\{B \in A\}}\right]
$$

$$
= \int_A \int_{\mathbb{R}} \exp\left(-\frac{x - b}{\sigma}y - \frac{(x - b)^2}{2\sigma^2}T\right)\mu(x)f_{W_T}(y)dydx
$$

$$
= \int_A \int_{\mathbb{R}} \exp\left(-\frac{x - b}{\sigma}y - \frac{(x - b)^2}{2\sigma^2}T\right)\mu(x)\frac{1}{\sqrt{T}}\varphi\left(\frac{y}{\sqrt{T}}\right)(y)dydx.
$$

Therefore,

$$
\mu_b(x) = \mu(x)\int_{\mathbb{R}}\frac{1}{\sqrt{T}}\exp\left(-\frac{y(x - b)}{\sigma} - \frac{(x - b)^2}{2\sigma^2}T\right)\varphi\left(\frac{y}{\sqrt{T}}\right)dy
$$

$$
= \frac{\mu(x)}{\sqrt{2\pi T}}\int_{\mathbb{R}}\exp\left(\frac{y(b - x)}{\sigma} - \frac{(x - b)^2}{2\sigma^2}T - \frac{y^2}{2T}\right)dy
$$

$$
= \frac{\mu(x)}{\sqrt{2\pi T}}\int_{\mathbb{R}}\exp\left(-\frac{1}{2T}\left(y^2 - \frac{2Ty(b - x)}{\sigma} + \frac{(x - b)^2}{\sigma^2}T^2\right)\right)dy
$$

$$
= \frac{\mu(x)}{\sqrt{2\pi T}}\int_{\mathbb{R}}\exp\left(-\frac{1}{2T}\left(y - \frac{b - x}{\sigma}T\right)^2\right)dy
$$

$$
= \mu(x).
$$

$\square$

### F.7 PROOF OF THEOREM 17

*Proof.* From Corollary 3.8 in Guan et al. (2022), if $X_T$ is a terminal wealth such that there exists a constant $\kappa > 0$ with

$$\begin{cases} u'(X_T) = \dfrac{\kappa \eta_T \int_{\mathbb{R}} \Phi'_{\lambda,\beta} \left( E^{Q^b} \left[ u(X_T) \right] \right) d\mu(b)}{\int_{\mathbb{R}} \Phi'_{\lambda,\beta} \left( E^{Q^b} \left[ u(X_T) \right] \right) \eta_T^b \, d\mu(b)} \\ E^{Q^r} \left[ X_T \right] = x_0 e^{rT}, \end{cases} \tag{38}$$

then $X_T$ is the optimal terminal wealth corresponding to Problem (10). Since $u$ and $\Phi_{\lambda,\beta}$ are both strictly increasing and strictly concave, then $u' > 0$ and $\Phi'_{\lambda,\beta} > 0$, hence the first equation is equivalent to (we abuse the notation by using $\kappa$ for both the positive multiplier and its logarithm)

$$\log \left( u'(X_T) \right) = \kappa + \log(\eta_T) + \log \left( \int_{\mathbb{R}} \Phi'_{\lambda,\beta} \left( E^{Q^b} \left[ u(X_T) \right] \right) d\mu(b) \right)$$

$$- \log \left( \int_{\mathbb{R}} \Phi'_{\lambda,\beta} \left( E^{Q^b} \left[ u(X_T) \right] \right) \eta_T^b \, d\mu(b) \right),$$

where $\log(\eta_T) = \frac{r - b_0}{\sigma} \left( W_T + \frac{B - b_0}{\sigma} T \right) - \frac{(b_0 - r)^2}{2\sigma^2} T$ and $\log \left( \Phi'_{\lambda,\beta} \left( E^{Q^b} \left[ u(X_T) \right] \right) d\mu(b) \right)$ does not involve an exponential term so it will not explode. Therefore it suffices to simplify $\log \left( \int_{\mathbb{R}} \Phi'_{\lambda,\beta} \left( E^{Q^b} \left[ u(X_T) \right] \right) \eta_T^b \, d\mu(b) \right)$.

Suppose $A : \Omega \to \mathbb{R}$ is a random variable that is independent from $B$ and $W$ with distribution $\mu$, and if we denote the $\sigma$-algebra generated by $B$ and $W_T$ as $\mathcal{G}$, then

$$\int_{\mathbb{R}} \Phi'_{\lambda,\beta} \left( E^{Q^b} \left[ u(X_T) \right] \right) \eta_T^b \, d\mu(b)$$

$$= \int_{\mathbb{R}} \Phi'_{\lambda,\beta} \left( E^{Q^a} \left[ u(X_T) \right] \right) \exp \left( -\frac{b_0 - a}{\sigma} \left( W_T + \frac{B - b_0}{\sigma} T \right) - \frac{(b_0 - a)^2}{2\sigma^2} T \right) d\mu(a)$$

$$= E^P \left[ \Phi'_{\lambda,\beta} \left( E^{Q^A} \left[ u(X_T) \right] \right) \exp \left( -\frac{b_0 - A}{\sigma} \left( W_T + \frac{B - b_0}{\sigma} T \right) - \frac{(b_0 - A)^2}{2\sigma^2} T \right) \bigg| \mathcal{G} \right]$$

$$= E^Q \left[ \Phi'_{\lambda,\beta} \left( E^{Q^A} \left[ u(X_T) \right] \right) \bigg| \mathcal{G} \right] E^P \left[ \exp \left( -\frac{b_0 - A}{\sigma} \left( W_T + \frac{B - b_0}{\sigma} T \right) - \frac{(b_0 - A)^2}{2\sigma^2} T \right) \bigg| \mathcal{G} \right],$$

where the last equality is from abstract Bayes' rule (Elliott et al., 1995) with the Radon-Nikodym derivative

$$\frac{dQ}{dP} = \frac{\exp \left( -\frac{b_0 - A}{\sigma} \left( W_T + \frac{B - b_0}{\sigma} T \right) - \frac{(b_0 - A)^2}{2\sigma^2} T \right)}{E^P \left[ \exp \left( -\frac{b_0 - A}{\sigma} \left( W_T + \frac{B - b_0}{\sigma} T \right) - \frac{(b_0 - A)^2}{2\sigma^2} T \right) \bigg| \mathcal{G} \right]}.$$

For the first term, we need to know the distribution of $A$ under $Q$ conditioned on $\mathcal{G}$. That is, for any Borel measurable set $E \subset \mathbb{R}$, from the abstract Bayes' rule, we want to compute

$$Q \left( A \in E \mid \mathcal{G} \right) = E^Q \left[ \mathbf{1}_{\{A \in E\}} \mid \mathcal{G} \right] = \frac{E^P \left[ \frac{dQ}{dP} \mathbf{1}_{\{A \in E\}} \bigg| \mathcal{G} \right]}{E^P \left[ \frac{dQ}{dP} \bigg| \mathcal{G} \right]}$$

$$= \frac{E^P \left[ \exp \left( -\frac{b_0 - A}{\sigma} D - \frac{(b_0 - A)^2}{2\sigma^2} T \right) \mathbf{1}_{\{A \in E\}} \bigg| \mathcal{G} \right]}{E^P \left[ \exp \left( -\frac{b_0 - A}{\sigma} D - \frac{(b_0 - A)^2}{2\sigma^2} T \right) \bigg| \mathcal{G} \right]}.$$

Moreover, if $D = W_T + \frac{B - b_0}{\sigma} T$, then

$$\text{Numerator} = \int_E \exp \left( -\frac{b_0 - a}{\sigma} D - \frac{(b_0 - a)^2}{2\sigma^2} T \right) \cdot \frac{1}{\sqrt{2\pi\sigma_0^2}} \exp \left( -\frac{(a - \mu_0)^2}{2\sigma_0^2} \right) da,$$

$$\text{Denominator} = \int_{-\infty}^{\infty} \exp \left( -\frac{b_0 - a}{\sigma} D - \frac{(b_0 - a)^2}{2\sigma^2} T \right) \cdot \frac{1}{\sqrt{2\pi\sigma_0^2}} \exp \left( -\frac{(a - \mu_0)^2}{2\sigma_0^2} \right) da.$$

If we define $K_1 = \dfrac{T}{2\sigma^2} + \dfrac{1}{2\sigma_0^2}$, $K_2 = \dfrac{Tb_0}{\sigma^2} + \dfrac{D}{\sigma} + \dfrac{\mu_0}{\sigma_0^2}$, $K_3 = \dfrac{Tb_0^2}{2\sigma^2} + \dfrac{b_0 D}{\sigma} + \dfrac{\mu_0^2}{2\sigma_0^2}$, and $C = \dfrac{K_2^2}{4K_1} - K_3$, then

$$Q\left(A \in E \mid \mathcal{G}\right) = \frac{\exp(C) \displaystyle\int_E \exp\left(-K_1\left(a - \frac{K_2}{2K_1}\right)^2\right) da}{\exp(C) \int_{-\infty}^{\infty} \exp\left(-K_1\left(a - \frac{K_2}{2K_1}\right)^2\right) da}$$

$$= \frac{\exp(C) \displaystyle\int_E \exp\left(-K_1\left(a - \frac{K_2}{2K_1}\right)^2\right) da}{\exp(C)\sqrt{\dfrac{\pi}{K_1}}}$$

$$= \int_E \frac{1}{\sqrt{\pi/K_1}} \exp\left(-K_1\left(a - \frac{K_2}{2K_1}\right)^2\right) da$$

$$= \int_E \frac{1}{\sqrt{2\pi\sigma_Q^2}} \exp\left(-\frac{(a - \mu_Q)^2}{2\sigma_Q^2}\right) da,$$

where $\mu_Q = \dfrac{K_2}{2K_1} = \sigma_Q^2\left(\dfrac{Tb_0}{\sigma^2} + \dfrac{D}{\sigma} + \dfrac{\mu_0}{\sigma_0^2}\right) = \sigma_Q^2\left(\dfrac{W_T}{\sigma} + \dfrac{BT}{\sigma^2} + \dfrac{\mu_0}{\sigma_0^2}\right)$ and $\sigma_Q^2 = \dfrac{1}{2K_1} = \left(\dfrac{T}{\sigma^2} + \dfrac{1}{\sigma_0^2}\right)^{-1}$. This computation also gives the second term

$$E^P\left[\exp\left(-\frac{b_0 - A}{\sigma}D - \frac{(b_0 - A)^2}{2\sigma^2}T\right) \,\middle|\, \mathcal{G}\right] = \exp(C)\frac{\sigma_Q}{\sigma_0}.$$

Thus, under $Q$ conditioned on $\mathcal{G}$, $A$ follows a Gaussian distribution $\mathcal{N}\left(\mu_Q, \sigma_Q^2\right) \sim \mu_A$. Therefore,

$$\log \text{ term} = \log\left(\int_{\mathbb{R}} \Phi'_{\lambda,\beta}\left(E^{Q^b}[u(X_T)]\right) \eta_T^b \, d\mu(b)\right)$$

$$= \log\left(\exp(C)\frac{\sigma_Q}{\sigma_0}\right) + \log\left(\int_{\mathbb{R}} \Phi'_{\lambda,\beta}\left(E^{Q^a}[u(X_T)]\right) d\mu_A(a)\right)$$

$$= C + \log(\sigma_Q) - \log(\sigma_0) + \log\left(\int_{\mathbb{R}} \Phi'_{\lambda,\beta}\left(E^{Q^a}[u(X_T)]\right) d\mu_A(a)\right).$$

Hence, if $X_T$ is a terminal wealth such that there exists a constant $\kappa > 0$ with $E^{Q^r}[X_T] = x_0 e^{rT}$ and

$$\log\left(u'(X_T)\right) = \kappa + \frac{r - b_0}{\sigma}\left(W_T + \frac{B - b_0}{\sigma}T\right) - \frac{(b_0 - r)^2}{2\sigma^2}T$$

$$+ \log\left(\int_{\mathbb{R}} \Phi'_{\lambda,\beta}\left(E^{Q^b}[u(X_T)]\right) d\mu(b)\right)$$

$$- \log\left(\int_{\mathbb{R}} \Phi'_{\lambda,\beta}\left(E^{Q^b}[u(X_T)]\right) \eta_T^b \, d\mu(b)\right)$$

$$= \log(\kappa) + \frac{r - b_0}{\sigma}\left(W_T + \frac{B - b_0}{\sigma}T\right) - \frac{(b_0 - r)^2}{2\sigma^2}T$$

$$+ \log\left(\int_{\mathbb{R}} \Phi'_{\lambda,\beta}\left(E^{Q^b}[u(X_T)]\right) d\mu(b)\right)$$

$$- \log\left(\int_{\mathbb{R}} \Phi'_{\lambda,\beta}\left(E^{Q^a}[u(X_T)]\right) d\mu_A(a)\right) - C - \log(\sigma_Q) + \log(\sigma_0),$$

then $X_T$ is the optimal terminal wealth corresponding to Problem (10). If we define

$$L_\kappa(X_T) = \kappa - \log\left(u'(X_T)\right) + \log\left(\int_{\mathbb{R}} \Phi'_{\lambda,\beta}\left(E^{Q^b}\left[u(X_T)\right]\right) d\mu(b)\right)$$

$$- \log\left(\int_{\mathbb{R}} \Phi'_{\lambda,\beta}\left(E^{Q^a}[u(X_T)]\right) d\mu_A(a)\right)$$

$$+ K_3 - \frac{K_2^2}{4K_1} - \log\left(\sigma_Q\right) + \log\left(\sigma_0\right) + \frac{r - b_0}{\sigma}\left(W_T + \frac{B - b_0}{\sigma}T\right) - \frac{(b_0 - r)^2}{2\sigma^2}T,$$

then $L_\kappa(X_T) = 0$. $\qquad\square$

# G ALGORITHMS

## G.1 DEFERRED POLICY EVALUATION ALGORITHM

---
**Algorithm 2** rMLMC DRBC Policy Evaluation Step

---
**Input:** A simulator $\mathcal{S}$, rMLMC parameter $R \in (\frac{1}{2}, \frac{3}{4})$, prior distribution $\mu$, parameter $n_0, R$, function $\phi$, initializations of $(\lambda, \beta)^T$, policy $\pi$.
**Output:** Estimator of DRBC policy value $\hat{Q}_{\text{DRBC}}(\pi)$.
**repeat**

- Draw $n$ i.i.d. samples $\{b_i\}_{1 \le i \le n}$ from $\mu$. For each $i = 1, 2, \ldots, n$, sample $\tilde{N}^{b_i} \sim \text{Geo}(R)$ independently, and compute $N^{b_i} = \tilde{N}^{b_i} + n_0$, then give $b_i$ to $\mathcal{S}$ and generate $2^{N^{b_i}+1}$ i.i.d. samples of $\hat{Z}_i^b$.

- Compute $\nabla F$ and $H(F)$ by Theorem 37. Update $(\lambda, \beta)^T = (\lambda, \beta)^T - H(F)^{-1}\nabla F$.

**until** $(\lambda, \beta)^T$ converges.
**Return** $\hat{Q}_{\text{DRBC}}(\pi) = \beta - \lambda\delta + \mathcal{E}_{\pi,\delta,\lambda,\beta}$ (Equation (7)).

---

## G.2 DEFERRED ALGORITHM FOR FINITE PRIOR EXAMPLE

---
**Algorithm 3** DRBC KL Problem Solver

---
**Input:** A simulator $\mathcal{S}$, simulation observations $S = \{S_t\}_{t\in[0,T]}$, real observations $\tilde{S} = \{\tilde{S}_t\}_{t\in[0,T]}$ rMLMC parameter $R \in (\frac{1}{2}, \frac{3}{4})$ and $n_0$, prior $\boldsymbol{p}$, initializations of $\lambda$, $\pi$, and $k = 0$, step-size sequence $\{\alpha_k : k \in \mathbb{Z}_{\geq 0}\}$.
**Output:** DRBC optimal policy $\hat{\pi}_{\text{DRBCKL}}$.
**repeat**

- Draw $n$ i.i.d. samples $\{b_i\}_{1 \le i \le n}$ from $\mu$. For each $i = 1, 2, \ldots, n$, sample $\tilde{N}^{b_i} \sim \text{Geo}(R)$ independently, and compute $N^{b_i} = \tilde{N}^{b_i} + n_0$, then give $b_i$ to $\mathcal{S}$ and generate $2^{N^{b_i}+1}$ i.i.d. samples of $\hat{Z}_i^b$.

- Compute GF $:= \frac{\partial}{\partial\lambda}\mathcal{E}_{\text{KL}}$ (Theorem 39). Update $\lambda = \lambda + \alpha_k\text{GF}$ and update $k = k + 1$.

- Update $\pi$ by Algorithm 4 with $\lambda$ and $S$.

**until** $\lambda$ converges.
**Return** $\hat{\pi}_{\text{DRBCKL}} = $ output of Algorithm 4 with $\lambda$ and $\tilde{S}$.

---

## G.3 LEARNING THE OPTIMAL ALTERNATIVE MEASURE FOR THE FINITE PRIOR EXAMPLE

Recall that the problem that we want to solve is

$$\inf_{\substack{\boldsymbol{q}=(q_1,\ldots,q_d) \\ \sum_{i=1}^d q_i=1,\text{and } q_i \geq 0}} V(\boldsymbol{q}) = \inf_{\substack{\boldsymbol{q}=(q_1,\ldots,q_d) \\ \sum_{i=1}^d q_i=1,\text{and } q_i \geq 0}} \tilde{V}(\boldsymbol{q}) + \lambda \sum_{i=1}^d q_i \log\frac{q_i}{p_i},$$

where $\tilde{V}(\boldsymbol{p}) := \frac{(x_0 e^{rT})^\alpha}{\alpha} \left( \int_{\mathbb{R}} \left( \tilde{F}_{\boldsymbol{p}}(T,z) \right)^{\frac{1}{1-\alpha}} \varphi_T(z) dz \right)^{1-\alpha}$ and $\tilde{F}_{\boldsymbol{p}}(t,z) = \sum_{i=1}^d p_i L_t(b_i, z)$.

Note that the value function can be seen as an expectation with respect to the Gaussian distribution $\mathcal{N}(0,T)$, thus it can be computed via simulation methods (plain Monte Carlo).

On the other hand, the gradient information of $\tilde{V}(\boldsymbol{p})$ is hard to compute. Thus, we will focus on the zero-order methods (Nesterov & Spokoiny, 2017; Shamir, 2017; Duchi et al., 2015). We will mimic first-order optimization strategies by replacing the gradient of the objective function with an approximation built through finite differences (LeVeque, 2007): for a fixed $i = 1, 2, \ldots, d$, let $h > 0$ small, then

$$\frac{\partial \tilde{V}(\boldsymbol{p})}{\partial p_i} \approx \frac{1}{2h} \left[ \tilde{V}(\boldsymbol{p} + h\boldsymbol{e}_i) - \tilde{V}(\boldsymbol{p} - h\boldsymbol{e}_i) \right],$$

where $\boldsymbol{e}_i$ is the unit vector in the $i$th coordinate direction. Note that here we use the central difference method, which has the $\mathcal{O}(h^2)$ rate of convergence, instead of the backward or forward difference method, which both have the $\mathcal{O}(h)$ rate of convergence. The exact steps to solve Problem (9) are summarized in Algorithm 4.

---

**Algorithm 4** rMLMC DRBC KL Policy Learning Step

**Input:** Prior distribution $\boldsymbol{p}$, step-size sequence $\{\alpha_k\}_{k \in \mathbb{Z}_{\geq 0}}$, parameter $h > 0$ and $\lambda > 0$, observations $\{S_t\}_{t \in [0,T]}$ (see remark in Section 5.1 for which process to plug in), initializations of $\boldsymbol{q}$ and $k = 0$.
**Output:** DRBC optimal policy $\{\pi^*_{\lambda,t}\}_{t \in [0,T]}$.
**repeat**

- For each $i = 1, 2, \ldots, d$, approximate $\tilde{V}(\boldsymbol{q} + h\boldsymbol{e}_i)$ and $\tilde{V}(\boldsymbol{q} - h\boldsymbol{e}_i))$ by simulations.

- For each $i = 1, 2, \ldots, d$, compute $\text{GF}_i = \frac{1}{2h} \left[ \tilde{V}(\boldsymbol{q} + h\boldsymbol{e}_i) - \tilde{V}(\boldsymbol{q} - h\boldsymbol{e}_i) \right] + \lambda \left( \log \left( \frac{q_i}{p_i} \right) + 1 \right)$.

- Update $\boldsymbol{q} = \boldsymbol{q} - \alpha_k \text{GF}$, update $\boldsymbol{q} = \text{softmax}(\boldsymbol{q})$, and update $k = k + 1$.

**until** $\boldsymbol{q}$ converges to $\boldsymbol{q}^*$.
**Return** $\pi^*_{\lambda,t} = \frac{\int_{\mathbb{R}} \nabla F_{\boldsymbol{q}^*}(T, z+Y_t) \left( F_{\boldsymbol{q}^*}(T, z+Y_t) \right)^{\frac{\alpha}{1-\alpha}} \varphi_{T-t}(z) dz}{(1-\alpha)\sigma \int_{\mathbb{R}} \left( F_{\boldsymbol{q}^*}(T, z+Y_t) \right)^{\frac{1}{1-\alpha}} \varphi_{T-t}(z) dz}$, for each $t \in [0,T]$, where $Y_t$ is given by Equation (13).

---

### G.4 IMPLEMENTATION DETAILS FOR THE KL DIVERGENCE CASE (ALGORITHM 3)

Here we elaborate on the details of simulating a sample of the optimal terminal wealth $X_T$. Recall that the controlled SDE is given by

$$dX_t = (X_t - \pi_t)r \, dt + \pi_t \left( B dt + \sigma dW_t \right), \tag{39}$$

where $\pi_t$ is the amount of money invested in the stock at time $t \in [0,T]$. Theorem 28 provides the optimal fraction of total wealth invested in stock at time $t$. If we model $\pi_t$ as the fraction of total wealth invested in stock at time $t$, then the controlled SDE becomes

$$dX_t = X_t \left( rdt + \pi_t \left( B - r \right) dt + \sigma dW_t \right). \tag{40}$$

These two formulations are equivalent since they give the same optimal value function (thus the same optimal terminal wealth) and the optimal fraction for the formulation (39) indeed gives the optimal control for the formulation (40). Thus, in Algorithm 3, when we sample $X_T$ for the KL divergence case, we plug in the optimal fraction for Equation (40) and use either Euler's method or rMLMC method (Rhee & Glynn, 2015).

### G.5 DEEP LEARNING METHOD TO LEARN OPTIMAL TERMINAL WEALTH FOR THE GENERAL CASE

In this section, we discuss and give the Algorithm 5 to compute the loss function $\mathcal{L}(\boldsymbol{\theta})$ with fixed neural network parameter (get high quality estimates (e.g. unbiased, low variance, and fast comput-

ing speed)), and then auto-differentiation (back propagation neural network (Rumelhart et al., 1986; LeCun et al., 2015)) can be used to do the optimizations (Algorithm 6).

Before discussing the loss function (11), we remark that Equation (38) also motivates a loss function for $b_1 \in \mathbb{R}$,

$$\tilde{\mathcal{L}}(\boldsymbol{\theta}) = E^{Q^{b_1}}\left[\left\|\left|u'(h_\theta(W_T, B)) \int_{\mathbb{R}} \Phi'_{\lambda,\beta}\left(E^{Q^a}\left[u(X_T)\right]\right) *\right.\right.\right. \tag{41}$$

$$\exp\left(-\frac{b_0 - a}{\sigma}\left(W_T + \frac{B - b_0}{\sigma}T\right) - \frac{(b_0 - a)^2}{2\sigma^2}T\right) d\mu(a)$$

$$-\kappa \exp\left(\frac{r - b_0}{\sigma}\left(W_T + \frac{B - b_0}{\sigma}T\right) - \frac{(b_0 - r)^2}{2\sigma^2}T\right) \int_{\mathbb{R}} \Phi'_{\lambda,\beta}\left(E^{Q^b}\left[u(h_\theta(W_T, B))\right]\right) d\mu(b)\bigg\|_2^2\bigg]$$

$$+ \left(E^{Q^r}[h_\theta(W_T, B)] - x_0 e^{rT}\right)^2.$$

In theory, loss (11) and (41) are equivalent. However, in terms of numerical computation, (41) is bad since when we initializing neural network parameters and do the optimization steps, we cannot control the flow and with high probability, the terms in (41) becomes $\infty$ since $\frac{1}{\sigma^2}$ is small. Moreover, the scalar learnable parameter in (41) is restricted to be positive so we need to do projected gradient descent, while in (11) the scalar learnable parameter can be any real number. Therefore, in order to achieve the numerical stability and convenience for implementation, we choose loss (11).

Recall that from Theorem 17, if we replace $\Phi_{\lambda,\beta}$ by $\phi$, then the conditions that the parametrized optimal terminal wealth $h_\theta(W_T, B)$ needs to satisfy are

$$E^{Q^r}[h_\theta(W_T, B)] = x_0 e^{rT}$$

and

$$\log\left(u'(h_\theta(W_T, B))\right) = \log\left(\int_{\mathbb{R}} \phi'\left(E^{Q^b}\left[u(h_\theta(W_T, B))\right]\right) d\mu(b)\right) \text{ (constant log term)} \tag{42}$$

$$- \log\left(\int_{\mathbb{R}} \phi'\left(E^{Q^a}\left[u(h_\theta(W_T, B))\right]\right) d\mu_A(a)\right) \text{ (random log term)} \tag{43}$$

$$+ \kappa + K_3 - \frac{K_2^2}{4K_1} - \log\left(\sigma_Q\right) + \log\left(\sigma_0\right) \tag{44}$$

$$+ \frac{r - b_0}{\sigma}\left(W_T + \frac{B - b_0}{\sigma}T\right) - \frac{(b_0 - r)^2}{2\sigma^2}T, \tag{45}$$

where the second condition is an equation of random variables, thus for each $b_1$ in the support of $\mu$, we need to sample $W_T$ and $B$ under the probability measure $Q^{b_1}$. From Appendix F.6, under $Q^{b_1}$, the distribution of $B$ is unchanged. Moreover, since under $Q^{b_1}$, $W^{b_1}$ is a standard Brownian motion and $W_t^{b_1} = W_t + \frac{B - b_1}{\sigma}t$, then we get samples of $W_T$ by first sampling $N \sim \mathcal{N}(0, T)$, $B \sim \mu$, and then compute $W_T = N - \frac{B - b_1}{\sigma}T$. The reason to do this can be found in Section G.6.

Once we sample from $W_T$ and $B$ under $Q^{b_1}$, we need to compute the nested expectations (42) and (43) since (45) is easy to compute. Essentially, computations of (42) and (43) are the same, except the outermost distribution in (42) is deterministic, where the outermost distribution in (43) depends on the sampling of $W_T$ and $B$ under $Q^{b_1}$. We can view the whole loss function as a nested expectation with layers in $x \mapsto x^2$ and $x \mapsto \log(x)$ and then apply the method in Syed & Wang (2023), but then the regularity conditions are hard to check. Thus, we only consider the rMLMC method for the nonlinear transform $\phi'$. For the rest of the estimator, we use the plug-in method.

## G.6 Deep Learning Method to Learn Optimal Policy for the General Case

Suppose we have numerically computed the approximation of the optimal terminal wealth $X_T^* \approx h_{\theta^*}(W_T, B)$, then for the policy evaluation step (Section 4), the simulation is simpler than the KL case (since we can directly sample unbiased terminal wealth directly without simulating an SDE).

---

**Algorithm 5** Loss Estimator with rMLMC

---

**Input:** Functions $\phi$ and $u$, scalar parameters $(x_0, b_0, T, \mu_0, \sigma_0, r, \sigma)$, fixed parameters $\kappa$ and $\theta$, data sets $\{W_k^{(1)}\}_{1 \leq k \leq n^{(1)}}$, $\{B_k^{(1)}\}_{1 \leq k \leq n^{(1)}}$, $\{W_{i,j}^{(3)}\}_{1 \leq j \leq n^{(2)}, 1 \leq i \leq 2^{N+1}}$, $\{B_{i,j}^{(3)}\}_{1 \leq j \leq n^{(2)}, 1 \leq i \leq 2^{N+1}}$, $\{W_{i,j,k}^{(3)}\}_{1 \leq j \leq n^{(2)}, 1 \leq i \leq 2^{N+1}, 1 \leq k \leq n^{(1)}}$, and $\{B_{i,j,k}^{(3)}\}_{1 \leq j \leq n^{(2)}, 1 \leq i \leq 2^{N+1}, 1 \leq k \leq n^{(1)}}$, sample $N$.

**Output:** Estimate of $\mathcal{L}(\boldsymbol{\theta}) = \mathcal{L}(\theta, \kappa)$.

- **Compute the constant log term: For each** $j = 1, \ldots, n^{(2)}$

  - For each $i = 1, \ldots, 2^{N+1}$, define $Y_{i,j} = \left( W_{i,j}^{(3)}, B_{i,j}^{(3)} \right)^T$ and compute $X_{i,j} = u\left(h_\theta\left(Y_{i,j}\right)\right)$.

  - Split the sequence of $X_{i,j}$ into odd and even indices for $i$. Compute $S_{j,l} = \sum_{i=1}^{l} X_{i,j}$, $S_{j,l}^O = \sum_{i=1}^{k} X_{i,j}^O$, and $S_{j,l}^E = \sum_{i=1}^{l} X_{i,j}^E$.

  - Compute $\Delta_N = \phi'\left(\frac{S_{j,2^{N+1}}}{2^{N+1}}\right) - \frac{1}{2}\left(\phi'\left(\frac{S_{j,2^N}^O}{2^N}\right) + \phi'\left(\frac{S_{j,2^N}^E}{2^N}\right)\right)$.

  - Compute $Z_j = \frac{\Delta_N}{p(N)} + \phi'\left(\frac{S_{j,2^{n_0}}}{2^{n_0}}\right)$, where $p(N)$ is the probability mass function of $N$.

  **end for** Compute $\hat{I}_c = \log\left(\frac{1}{n^{(2)}} \sum_{j=1}^{n^{(2)}} Z_j\right)$.

- Compute $D = W_k^{(1)} + \frac{B_k^{(1)} - b_0}{\sigma}T$, $K_1 = \frac{T}{2\sigma^2} + \frac{1}{2\sigma_0^2}$, $K_2 = \frac{Tb_0}{\sigma^2} + \frac{D}{\sigma} + \frac{\mu_0}{\sigma_0^2}$, and $K_3 = \frac{Tb_0^2}{2\sigma^2} + \frac{b_0 D}{\sigma} + \frac{\mu_0^2}{2\sigma_0^2}$.

- Compute $\hat{I}_{r_1} = \frac{r - b_0}{\sigma}\left(W_k^{(1)} + \frac{B_k^{(1)} - b_0}{\sigma}T\right) - \frac{(b_0 - r)^2}{2\sigma^2}T + K_3 - \frac{K_2^2}{4K_1} - \log\left(\sigma_Q\right) + \log\left(\sigma_0\right)$.

- **Compute the random log term: For each** $k = 1, \ldots, n^{(1)}$

  - **For each** $j = 1, \ldots, n^{(2)}$

    * For each $i = 1, \ldots, 2^{N+1}$, define $Y_{i,j,k} = \left(W_{i,j,k}^{(3)}, B_{i,j,k}^{(3)}\right)^T$ and compute $X_{i,j,k} = u\left(h_\theta\left(Y_{i,j,k}\right)\right)$.

    * Split the sequence of $X_{i,j,k}$ into odd and even indices for $i$. Compute $S_{j,l,k} = \sum_{i=1}^{l} X_{i,j,k}$, $S_{j,l,k}^O = \sum_{i=1}^{k} X_{i,j,k}^O$, and $S_{j,l,k}^E = \sum_{i=1}^{l} X_{i,j,k}^E$.

    * Compute $\Delta_{N,k} = \phi'\left(\frac{S_{j,2^{N+1},k}}{2^{N+1}}\right) - \frac{1}{2}\left(\phi'\left(\frac{S_{j,2^N,k}^O}{2^N}\right) + \phi'\left(\frac{S_{j,2^N,k}^E}{2^N}\right)\right)$ and $Z_{j,k} = \frac{\Delta_{N,k}}{p(N)} + \phi'\left(\frac{S_{j,2^{n_0},k}}{2^{n_0}}\right)$.

    **end for** Compute $\hat{I}_{r_2} = \log\left(\frac{1}{n^{(2)}} \sum_{j=1}^{n^{(2)}} Z_{j,k}\right)$. **end for**

- Compute $\mathcal{L}_2 = \left(\frac{1}{n^{(1)}} \sum_{k=1}^{n^{(1)}} h_\theta\left(W_k^{(1)}, B_k^{(1)}\right) - x_0 e^{rT}\right)^2$.

**Return:** $\mathcal{L}(\boldsymbol{\theta}) = \frac{1}{n^{(1)}} \sum_{k=1}^{n^{(1)}} \left(\kappa - \log\left(u'\left(h_\theta\left(W_k^{(1)}, B_k^{(1)}\right)\right)\right) + \hat{I}_{r_1} - \hat{I}_{r_2} + \hat{I}_c\right)^2 + \mathcal{L}_2$.

---

---

**Algorithm 6** rMLMC DR Policy Learning Step for the General Case

---

**Input:** Functions $\phi$ and $u$, scalar parameters $(x_0, b_0, T, \mu_0, \sigma_0, r, \sigma)$, step-size sequence $\{\alpha_k\}_{k \in \mathbb{Z}_{\geq 0}}$, initilizations $\boldsymbol{\theta} = (\theta, \kappa)^T$ and $k = 0$.

**Output:** DR optimal terminal wealth $X_T$.

**Samples for constant log term:** sample $\tilde{N} \sim \text{Geo}(R)$ and compute $N = \tilde{N} + n_0$; sample $n^{(2)}$ i.i.d. values $\{B_j^{(2)}\}_{1 \leq j \leq n^{(2)}}$ from $\mathcal{N}(\mu_0, \sigma_0^2)$. **For each** $j = 1, \ldots, n^{(2)}$

- Sample $2^{N+1}$ i.i.d. samples $\{B_{i,j}^{(3)}\}_{1 \leq i \leq 2^{N+1}}$ from $\mathcal{N}(\mu_0, \sigma_0^2)$.

- Sample $2^{N+1}$ i.i.d. samples $\{N_{i,j}^{(3)}\}_{1 \leq i \leq 2^{N+1}}$ from $\mathcal{N}(0, T)$.

- For each $i = 1, \ldots, 2^{N+1}$, compute $W_{i,j}^{(3)} = N_{i,j}^{(3)} - \frac{B_{i,j}^{(3)} - B_j^{(2)}}{\sigma} T$.

Sample $n^{(1)}$ i.i.d. values $\{B_k^{(1)}\}_{1 \leq k \leq n^{(1)}}$ from $\mathcal{N}(\mu_0, \sigma_0^2)$ and $n^{(1)}$ i.i.d. values $\{N_k^{(1)}\}_{1 \leq k \leq n^{(1)}}$ from $\mathcal{N}(0, T)$. For each $k = 1, \ldots, n^{(1)}$, compute $W_k^{(1)} = N_k^{(1)} - \frac{B_k^{(1)} - r}{\sigma} T$. Compute $\sigma_Q^2 = \left( \frac{T}{\sigma^2} + \frac{1}{\sigma_0^2} \right)^{-1}$.

**Samples for random log term: For each** $k = 1, \ldots, n^{(1)}$

- Compute $\mu_Q = \sigma_Q^2 \left( \frac{W_k^{(1)}}{\sigma} + \frac{B_k^{(1)} T}{\sigma^2} + \frac{\mu_0}{\sigma_0^2} \right)$. Sample $n^{(2)}$ i.i.d. values $\{B_{j,k}^{(2)}\}_{1 \leq j \leq n^{(2)}}$ from $\mathcal{N}(\mu_Q, \sigma_Q^2)$. **For each** $j = 1, \ldots, n^{(2)}$

  - Sample $2^{N+1}$ i.i.d. samples $\{B_{i,j,k}^{(3)}\}_{1 \leq i \leq 2^{N+1}}$ from $\mathcal{N}(\mu_0, \sigma_0^2)$.

  - Sample $2^{N+1}$ i.i.d. samples $\{N_{i,j,k}^{(3)}\}_{1 \leq i \leq 2^{N+1}}$ from $\mathcal{N}(0, T)$.

  - For each $i = 1, \ldots, 2^{N+1}$, compute $W_{i,j,k}^{(3)} = N_{i,j,k}^{(3)} - \frac{B_{i,j,k}^{(3)} - B_{j,k}^{(2)}}{\sigma} T$.

**repeat**

- Compute $\mathcal{L}(\boldsymbol{\theta})$ by Algorithm 5 and the above samples.

- Update $\boldsymbol{\theta} = \boldsymbol{\theta} - \alpha_k \nabla_{\boldsymbol{\theta}} \mathcal{L}(\boldsymbol{\theta})$, where the gradient is computed by back propagation; update $k = k + 1$.

**until** $\boldsymbol{\theta} = (\theta, \kappa)^T$ converges to $\boldsymbol{\theta}^* = (\theta^*, \kappa^*)^T$.

**Return** $X_T = h_{\theta^*}(W_T, B)$.

---

Table 3: Policy evaluation for different sample sizes of $B$.

| $\delta$ | $n = 10^2$ | $n = 10^3$ | $n = 10^4$ |
|---|---|---|---|
| 0.01 | $3.7642 \pm 0.0399$ | $3.7663 \pm 0.0124$ | $3.7654 \pm 0.0040$ |
| 0.05 | $3.6951 \pm 0.0413$ | $3.6971 \pm 0.0130$ | $3.6964 \pm 0.0040$ |
| 0.10 | $3.6425 \pm 0.0421$ | $3.6449 \pm 0.0135$ | $3.6446 \pm 0.0041$ |

However, in order to use the rMLMC estimator for $Q_{\text{DRBC}}$, we still need to draw $\{b_1, b_2, \ldots, b_n\}$ from the distribution of $\mu$ and generate unbiased estimator of

$$E^{P^{b_i}}[u(X_T)] = E^{Q^{b_i}}[u(X_T)] = E^{Q^{b_i}}[u(h_{\theta^*}(W_T, B))].$$

Thus, it is better to design a loss function for each $b_i$ rather than choose a "uniformly" best $Q^{b^*}$ for the loss function.

Suppose the multipliers of the alternative optimization algorithm converge finally, then we get realizations of $\hat{V}(x_0)$ in the worst case. Since the optimal value function of the distributionally robust problem is just plugging the worst case probability $\mu^*$ into the original form of the solution (Theorem 28), thus we can solve the optimization problem $\inf_\theta \mathcal{L}(\theta)$ to derive $\mu^*$ if we parametrize the density by a neural network:

$$\mathcal{L}(\theta) = E^P\left[\left\|\hat{V}(x_0) - V(x_0, f_\theta)\right\|_2^2\right],$$

where $\theta$ represents the parameter of the neural network $f_\theta$ for the approximation of $\rho_{\mu^*}$ and $V(x_0, \mu) = \frac{(x_0 e^{rT})^\alpha}{\alpha}\left(\int_{\mathbb{R}} (F_\mu(T, z))^{\frac{1}{1-\alpha}} \varphi_T(z) dz\right)^{1-\alpha}$ for a fixed distribution $\mu \in \mathcal{P}(\mathbb{R})$. To simplify the trackability issue, we may parametrize $\rho_{\mu^*}$ as exponential family or Gaussian mixtures (Goodfellow et al., 2016). Finally, to get the DRBC optimal control with real observations of stock prices, we plug in Theorem 28 with the learned worst-case probability and the observations.

## H    ADDITIONAL EXPERIMENTS

### H.1    RATE OF CONVERGENCE FOR POLICY EVALUATION

In this section, we investigate the convergence rate in the KL case established by Theorem 13 (specifically, Theorem 40) for values of $\delta$ equal to 0.01, 0.05, and 0.1. For each fixed $\delta$, we sample and compare three different numbers of independent and identically distributed copies of $B$: $n = 10^2$, $10^3$, and $10^4$. Table 3 presents the means and standard deviations of the estimator with a fixed policy $\pi$, computed from 100 independent experiments. The numerical results demonstrate that the estimator $\hat{Q}_{\text{DRBCKL}}(\pi)$ converges, and both the scaling rates of the standard deviation and the difference $\left|\hat{Q}_{\text{DRBCKL}}(\pi) - Q_{\text{DRBCKL}}(\pi)\right|$ are consistent with the $\mathcal{O}_p\left(n^{-1/2}\right)$ rate predicted by theory.

To make sure the validity of our comparison, we first run the policy learning step to get the $\pi$ initialization. We run policy learning step with prior values $[0.01, 0.46, 0.30, 0.21, 0.27]$ and probability mass function of prior random variable $B$ $[0.05, 0.35, 0.35, 0.15, 0.1]$. We equally divide $[0, T]$ into 1000 intervals, and initialize $\lambda = 100$. When calculating the gradient of $\tilde{V}$, we set $h = 10^{-6}$ and learning rate $\alpha_k = 10^{-5}$ a same number across all loops $k$. We set the convergence condition to be the sum of squared errors of $\boldsymbol{q}_k - \boldsymbol{q}_{k-1}$ less than $10^{-5}$. After we get the converged $\boldsymbol{q}^*$, we get $\pi$ from Equation (13), as in Algorithm 4.

Then we run Algorithm 1. We set $n_0 = 3$ and $\alpha_k = 0.01$. Other hyperparameters are the same as Section I.1.2 and prior is the same as above. Here we set the convergence condition to be $\frac{\partial}{\partial \lambda}\mathcal{E}_{\text{KL}} < 0.3\delta$. For using inner samples to estimate $E^{P^b}[u(X_T)]$, we simulate Equation (40) with $B = b$ 100 times to get $X_T$ and then get the average. Finally we use the converged $\lambda^*$ to calculate $\mathcal{E}_{\text{KL}}$ with different sizes of $n$ 100 times to get the results in Table 3. We document that using 4 Intel Skylake 6148, 20-core, 2.4GHz, 150W processors, the whole process including all three $n$ takes about 40 hours.

Table 4: Setting 1: comparison of average Sharpe ratios and expected utilities for Bayesian with correct prior (BCP), Bayesian with incorrect prior (BIP), and DRBC ($\delta = 10^{-3}/10^{-2}$).

| METHOD | SHARPE RATIO | EXPECTED UTILITY |
|---|---|---|
| BIP | 0.68057 | 3.41674 |
| BCP | 1.05472 | 4.37966 |
| DRBC ($10^{-3}$) | 0.93987 | 3.89788 |
| DRBC ($10^{-2}$) | 0.93989 | 3.89793 |

Table 5: Setting 2: comparison of average Sharpe ratios and expected utilities for Bayesian with correct degenerate prior (BCPD); DRC and DRBC ($\delta = 10^{-3}/10^{-2}$).

| METHOD | SHARPE RATIO | EXPECTED UTILITY |
|---|---|---|
| BCPD | 2.44595 | 6.07575 |
| DRC($10^{-3}$) | 1.04900 | 3.46948 |
| DRC($10^{-2}$) | 0.91977 | 3.38634 |
| DRBC ($10^{-3}$) | 1.48139 | 4.43138 |
| DRBC ($10^{-2}$) | 1.48136 | 4.43145 |

## H.2 THE FINITE PRIOR CASE WITH KL UNCERTAINTY SET

In this section, we use Sharpe ratio and value function (expected utility) for Problem (3) as evaluation metrics to compare DRBC method with different baselines in different settings. In Setting 1, market paths are generated from Equation (1) multiple times with a groundtruth distribution of drift. We choose an incorrect prior for both Bayesian and DRBC methods. We compare the performance of the Bayesian approach with the incorrect prior (BIP), the correct prior (BCP) (using grondtruth) to DRBC method and report the results in Table 4. The results indicate the effectiveness of DRBC over prior misspecification.

Setting 2 gives comparisons between DRBC and DRC methods under another market setting where drift $B$ in Equation (1) degenerates to a single point. Again, we choose an incorrect prior for both DRBC and DRC to compute the optimal policies, and report evaluation metrics of them together with the BCP in this case (BCPD) in Table 5. The results clearly show that DRBC reduces the overpessimism and is relatively stable in terms of $\delta$ compared with DRC.

## H.3 HIGH DIMENSIONAL SYNTHETIC EXPERIMENTS WITH KL UNCERTAINTY SET

In this section, we scale the dimension of SDE up from one to one hundred to show the performance of our method with Sharpe Ratio and also show the necessity practicality of our method since it is designed beyond low dimensional cases. We apply Theorem 27 and modify algorithm 4 to high dimensional formulas to get optimal fractions. The results confirm DRBC can reduce over-pessimism, and certify our method in high dimensional settings. Details of implementation of DRBC and DRC are in section I.4.

Table 6: Comparison of average Sharpe ratios for Bayesian with correct degenerate prior (BCPD); DRC and DRBC for 100 assets.

| METHOD | SHARPE RATIO |
|---|---|
| BCPD | 0.954 |
| DRC | 0.397 |
| DRBC | 0.591 |

## I   EXPERIMENT DETAILS

### I.1   DETAILS OF EXPERIMENTS IN SECTION 6.1

#### I.1.1   CHOICE OF PERFORMANCE MEASURES

From Guan et al. (2022), if

$$\Phi(x) = \begin{cases} \frac{x^\gamma}{\gamma}, & \text{if } x \geq 0, \\ \frac{-(-x)^\gamma}{\gamma}, & \text{if } x < 0, \end{cases} \quad \text{where } \gamma \in (0, 1) \text{ is a fixed constant,}$$

then the optimal terminal wealth has the closed-form

$$X_T^* = \exp\left( \frac{1}{\alpha} \left( \frac{p}{2T} \left( W_T + \frac{B - b_0}{\sigma} \right)^2 + q \left( W_T + \frac{B - b_0}{\sigma} \right) + c \right) \right), \tag{46}$$

where

$$\begin{cases} p = \dfrac{\frac{1}{1-\alpha} + \sigma_0^2 - \sqrt{\sigma_0^4 + \frac{2-4\alpha}{1-\alpha}\sigma_0^2 + \frac{1}{(1-\alpha)^2} - \frac{4\alpha}{1-\alpha}\gamma}}{2(\sigma_0^2 + \gamma)}, \\[4mm] q = \dfrac{\alpha(1-p)}{(1-\alpha)(1-\gamma p)}\nu, \\[4mm] c = \alpha\left[ \log(x_0) + rT + \left( \nu^2 - \dfrac{\left(\nu - \frac{q}{\alpha}\right)^2}{1 - \frac{p}{\alpha}} \right) \dfrac{T}{2} \right]. \end{cases}$$

From Guan et al. (2022), for a fixed $b \in \mathbb{R}$ (more precisely in the support of the prior)

$$E^{Q^b}[u(X_T^*)] = \frac{1}{\alpha\sqrt{1-p}} \exp\left( \frac{pT}{2(1-p)}\nu_b^2 - \frac{qT}{1-p}\nu_b + \frac{q^2T}{2(1-p)} + c \right)$$

$$= \frac{1}{\alpha\sqrt{1-p}} \exp\left( \frac{pT}{2(1-p)} \frac{(b-b_0)^2}{\sigma^2} + \frac{qT}{1-p} \frac{b-b_0}{\sigma} + \frac{q^2T}{2(1-p)} + c \right).$$

Following the discussion in Appendix G.6, in the alternate optimization steps for the DRBC algorithm, the essential property is how accurate that this optimal terminal wealth can be used to compute $E^{Q^{b_1}}[u(X_T)]$ for a fixed $b_1 \in \mathbb{R}$, thus we choose it to compare with the closed form solution. The performance measures that we taken into considerations are their equivalence properties are used in the alternative optimization procedures.

#### I.1.2   IMPLEMENTATION DETAILS

We start with the hyperparameter settings for the synthetic data, then introduce our neural network settings. The network is trained on 1 Nvidia A100 GPU for about one GPU hours.

Sample size is set to be 2000. Second level sample size for rMLMC method to be 100, and geometric distribution parameter $R = 0.65$. To make the synthetic data close to real financial market observations, we let $\sigma = 0.4$, $\sigma_0 = 2$, $r = 0.05/0.1$, $b_0 = 0.1$, $T = 1$, $\mu_0 = 0.1$, $b = 0.1/0.3$, and $x_0 = 1$. For the function $\Phi'$, to get numerical stability, we use a truncation to approximate at 0.01 and 2, when $x$ is smaller than 0.001, $\Phi'$ gives a constant; when $x$ is larger than 2, $\Phi'$ is $x^{-\frac{1}{2}}$; between 0.001 and 2, $\Phi'$ is the linear interpolation of above two functions.

We use a modified multi-layer perceptron (MLP) $h_\theta$ to estimate $X_T^*$. The MLP has four layers in total, first layer takes 2-dim input $(W_T, B)$ and maps to 128 hidden nodes; second layer maps 128 nodes to 256 hidden nodes; third layer maps 256 nodes to 256 nodes, and final layer maps 256 nodes to one output. We use LeakyReLU (Maas et al., 2013) as activation function with parameter 0.01. Since our parameter settings mimic real financial data, we need to modify the output of the MLP to satisfy non-negative constraint and match the real terminal wealth distribution easier. We impose a partial linear structure on top of the MLP with constant 1 and learnable parameter $b$. The constant is from financial practices that under optimal portfolio strategy, investor earns excess return. $b$ is set

here for an easier learning process and better gradient flow. We also set $\kappa$ a learnable parameter and use gradient descent alternatively for MLP parameters, $b$ and $\kappa$ in the training pipeline.

For learning rate schedule, we adopt a Warmup-Stable-Decay (Wen et al., 2024) learning rate schedule which achieves great success in large language models. We use 1000 epochs in total, with 10 epochs to linearly warmup the learning rate to 0.001, then steadily train 400 epochs, and finally decay to 0.0003 for the rest 590 epochs. For $b$, learning rate is fixed at $10^{-4}$. For $\kappa$, learning rate linearly decays from 0.01 to $10^{-4}$, then stays until training is done. We use Adam (Kingma & Ba, 2015) as the optimizer and saved models can be found at our repository. Training losses for different $b$ and $r$ values are shown in Figure 1. In all three hyperparameter settings, our network shows stable loss curves and achieve good performances comparing to theory results, as stated in Section 6.1.

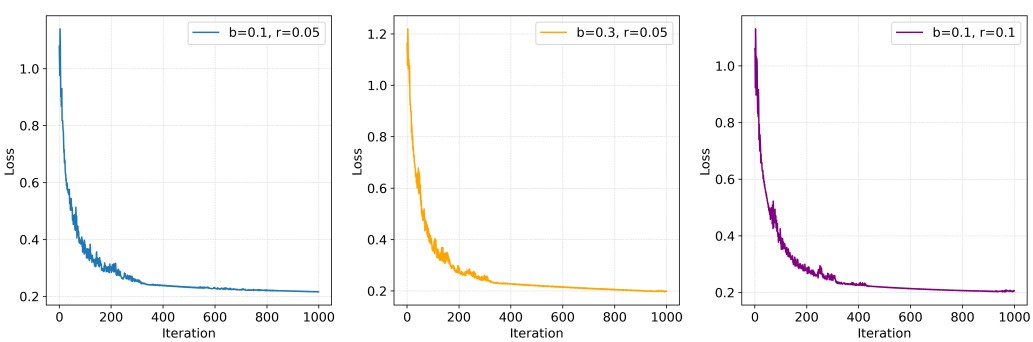

Figure 1: Training losses for different $b$ and $r$ values

We denote the trained optimal parameter as $\theta^*$. We evaluate the model by randomly generating new input pairs $(W_T, B)$ of 2000 samples with different seeds to get $E^{Q^b}[u(h_{\theta^*}(W_T, B))]$. And we run such experiment 100 times to get the mean and standard deviations shown in Table 1.

### I.2 DETAILS OF EXPERIMENTS IN SECTION 6.2

The daily $S\&P$ 500 constituents data from 2015-01-01 to 2024-12-31 is from Wharton Research Data Services. For the ease of data cleaning and to avoid stock inclusion and exclusion to the index, we only keep stocks that are $S\&P$ 500 constituents in the whole time window, resulting in 326 stocks left. Interest rate data is from Federal Reserve Bank of St.Louis. We use a rolling window of 1 year for getting the interest rate $r$ and $\sigma$, and use them in DRBC and baseline methods for the following month's investment allocation.

We remark the choice of $\delta$ is more a managerial decision rather than a scientific choice, and too large $\delta$ will not give meaningful solutions. Here we follow Si et al. (2023) to use the existing data to estimate the distributional shift, which implies a choice of $\delta$. Over the 326 stocks, the mean of $\delta \approx 0.15$, thus we choose it for the experiments. For the prior, we choose two fixed finite priors with supports inspired by Wang & Zhou (2020). Prior 1 is [-0.08,0.16,-0.02,0.04,0.10], with probability [0.35,0.08,0.25,0.22,0.10]. Prior 2 is [-0.05,0.15,0.00,0.05,0.10], with probability [0.45,0.05,0.25,0.15,0.1]. The histograms of sharpe ratios for two priors are shown in figure 2. We documented that using a single Intel Skylake 6148, 20-core, 2.4GHz, 150W processor, looping over all stocks for a single prior takes about 30 hours.

### I.3 DETAILS OF EXPERIMENTS IN SECTION H.2

#### I.3.1 DETAILS OF SETTING 1

For experiments in Section H.2, we run full DRBC Algorithm 3 to get $\hat{\pi}_{\text{DRBCKL}}$. Most hyperparameters are the same as in previous settings. One change here is the initialization of $\lambda$. Due to the hardness of convergence of Algorithm 3, we need finer pre-condition of $\lambda$. Based on Faury et al. (2020), which proves $\lambda = \mathcal{O}(\frac{1}{\sqrt{\delta}})$, we make our $\lambda = \frac{0.33}{\sqrt{\delta}}$. Another change here is we shift the prior distri-

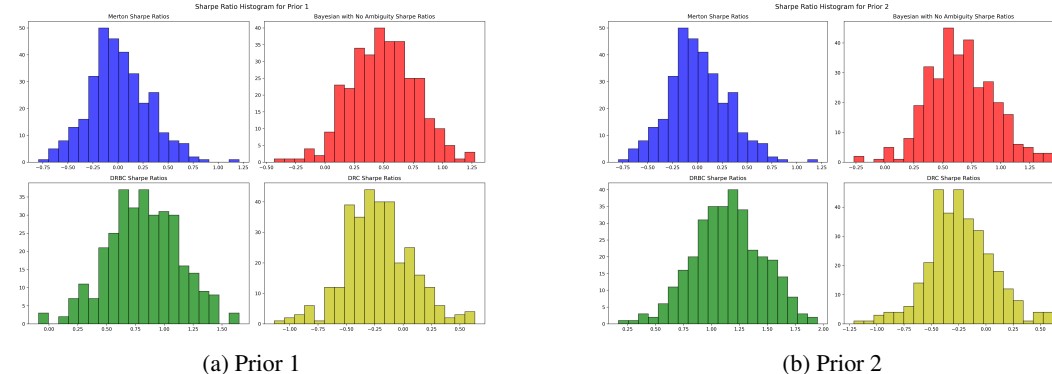

(a) Prior 1        (b) Prior 2

Figure 2: Histogram of Sharpe Ratios under different priors.

bution, which is also noted as incorrect prior in Table 4 to $[0.5, 0.05, 0.2, 0.15, 0.1]$, with support of the prior random variable $B$ unchanged. The correct prior distribution is $[0.05, 0.5, 0.1, 0.15, 0.2]$. The convergence condition is set to be $|\lambda_k - \lambda_{k-1}| < 10^{-3}$.

We regard the full DRBC process in Algorithm 3 as the pre-training phase to get the optimal policy. Then we run the evaluation process as follows: First, we generate market data $\{S_t\}_{t \in [0,T]}$ 200 times by Equation (1) to get 200 paths. Then use Equation (13) to transform $\{S_t\}_{t \in [0,T]}$ to $\{Y_t\}_{t \in [0,T]}$. Thirdly, calculate optimal fractions $\pi$ for BIP, BCP and DRBC cases using Theorem 28 with transformed $Y_t$. Fourthly, simulate Equation (2) or equivalently Equation (40) with $\pi_{\text{BIP}}$, $\pi_{\text{BCP}}$ and $\pi_{\text{DRBC}}$ with the same Brownian terms $W_t$ as the first step to get the wealth paths $X_t$. Finally, we use two metrics to evaluate the performances of BIP, BCP and DRBC. Since DRBC only focuses on terminal wealth, to remove other randomness, we choose the Sharpe ratio definition as in Wang & Zhou (2020). Another metric is expected terminal utility, as defined in Equation (3), which is also the value function. We collect terminal utility from all 200 paths to get the mean and we report them in Table 4. We document that using 2 Intel Skylake 6148, 20-core, 2.4GHz, 150W processors, the whole process takes about half an hour.

### I.3.2 DETAILS OF SETTING 2

Recall that when $B$ degenerates to a constant, then the Bayesian problem (3) becomes the Merton's problem. By applying the dynamic programming principle, it suffices to consider the terminal value problem with $V(T) = 1$ and

$$\frac{dV}{dt} + \alpha V \sup_{\pi} \left\{ \frac{1}{2} \sigma^2 \pi^2 (\alpha - 1) + (B - r)\pi + r \right\} = 0.$$

An verification argument shows that it suffices to solve the supremum problem in the ordinary differential equation and the optimal fraction invested in the stock is a constant over time. Based on the theory from Hansen & Sargent (2001), the Hamilton-Jacobi-Bellman-Isaacs (HJBI) equation for the distributionally robust control (DRC) is

$$\frac{dV}{dt} + \alpha V \sup_{\pi} \inf_{\nu \in \mathcal{U}_\delta} \left\{ \frac{1}{2} \sigma^2 \pi^2 (\alpha - 1) + (E_\nu[B] - r)\pi + r \right\} = 0,$$

where we denote the distribution of $B$ as $\mu$, the uncertainty set is $\mathcal{U}_\delta = \{\nu : D_{\text{KL}}(\nu \parallel \mu) \le \delta\}$, and the notation $E_\nu[B]$ denotes the mean of random variable $B$ if its distribution is $\nu$. Similarly as the non-robust Merton's problem, it suffices to solve the sup-inf problem and get the optimal fraction invested in the stock.

For the inner infimum problem, we formulate it as below. Distributions $\nu$ and $\mu$ share same finite support $\{b_i\}_{i=1}^d$. We denote the probability mass of two distributions $\{q_i\}_{i=1}^d$ and $\{p_i\}_{i=1}^d$ respectively.

$$\text{minimize} \quad \sum_{i=1}^{d} q_i b_i,$$

$$\text{subject to} \quad \sum_{i=1}^{d} q_i \ln\left(\frac{q_i}{p_i}\right) \leq \delta, \tag{47}$$

$$\sum_{i=1}^{d} q_i = 1, \quad q_i \geq 0 \quad \forall i.$$

After solving the infimum problem, we can directly get the optimal $\pi_{\text{DRC}}$ as in Merton problem, which is a constant across time. For the degenerate prior cases, we can get $\pi_{\text{BCPD}}$ as above.

The experiment here is similar to Section I.3.1. In experiment, we choose the prior with $d = 5$. We choose the same incorrect prior. First and second steps are the same, except for the prior now is a point mass at 0.46. Then we follow the third step to get $\pi_{\text{DRBC}}$. Finally, we run the final step with $\pi_{\text{DRC}}$, $\pi_{\text{BCPD}}$ and $\pi_{\text{DRBC}}$ to get performance metrics. Using the same hardware as setting 1, we document similar time for the whole process.

### I.4  DETAILS OF HIGH DIMENSIONAL EXPERIMENT RESULTS

We first randomly generate high dimensional SDE, then for DRBC, we use theorem 27 to calculate the empirical centers of $B$ with formula in Appendix B by optimizing $V(x_0)$. For DRC, we slightly modify the one dimensional implementation in section I.3.2 with below.

Given: $P = (p_1, \ldots, p_m), \quad z_1, \ldots, z_m \in \mathbb{R}^d, \quad B(z) = \sum_{k=1}^{d} z_k, \quad \delta > 0.$

Define the tilted distribution: $q_i(\alpha) \;=\; \dfrac{p_i\, e^{\alpha B(z_i)}}{\sum_{j=1}^{m} p_j e^{\alpha B(z_j)}}.$

Then: $\mu_Q(\alpha) \;=\; \sum_{i=1}^{m} q_i(\alpha)\, z_i \;=\; \dfrac{\sum_{i=1}^{m} p_i\, z_i\, e^{\alpha B(z_i)}}{\sum_{j=1}^{m} p_j e^{\alpha B(z_j)}}.$

$D_{\text{KL}}(Q(\alpha)\,\|\,P) \;=\; \alpha\, \mathbb{E}_{Q(\alpha)}[B(Z)] \;-\; \log\left(\sum_{j=1}^{m} p_j e^{\alpha B(z_j)}\right).$

Find $\alpha^*$ such that $D_{\text{KL}}(Q(\alpha^*)\,\|\,P) = \delta.$

Get: $\alpha^*, \quad \mu_{Q^*} \;=\; \mu_Q(\alpha^*).$

Using Merton Style Formula to get Optimal Fraction: $\pi^* = \dfrac{1}{1-\alpha}\Sigma^{-1}(\mu_{Q^*} - r)$

In both DRBC and DRC case, we choose $\delta = 0.4$. All other settings are the same as section I.3.2.

*Remark* 45. Wang et al. (2023b) discusses the distributionally robust control (choose the worst case in every step) formulation in the discrete state space case, and derive conditions to apply the dynamic programming approaches similar to Hansen & Sargent (2001). We remark that these conditions are assumed in Hansen & Sargent (2001) rather than derived. A takeaway of this is that we may also do the similar theoretical foundation as in Wang et al. (2023b) and then do the similar steps as in Hansen & Sargent (2001): derive the HJB equation for the Bayesian problem and then get the HJBI equation for the DRBC formulation, which will have super complicated form and will be hard to solve. This is why we say the DRBC formulation looses the dynamic programming principle and another efficient method is needed to get the optimal policy.

## J  USAGE OF LLM

LLM is used to polish some of the writings of this paper.

