# OpenReview forum: "Duality and Policy Evaluation in Distributionally Robust Bayesian Diffusion Control"
_ICLR.cc/2026/Conference — Submitted to ICLR 2026_

### Official Review · Reviewer_NDUZ · 2025-10-31

**Soundness:** 3
**Presentation:** 3
**Contribution:** 2
**Rating:** 4
**Confidence:** 4

**Summary:**

The paper studies a distributionally robust Bayesian control (DRBC) formulation for continuous-time diffusion processes under model uncertainty, motivated by scenarios common in portfolio optimization. The authors focus on misspecification in the prior distribution of the unknown drift, formulating the DRBC game as an adversarial problem over divergence neighborhoods of the prior. They establish strong duality, derive efficient evaluation and learning methods, notably an unbiased rMLMC estimator, and provide both theoretical and empirical support, including neural-network-based policy learning and real-world stock data experiments.

**Strengths:**

The paper tackles an important challenge in robust control: mitigating over-conservatism of classic distributionally robust control by localizing the adversarial game to the Bayesian prior.
1. The mathematical formulation and the derivation of strong duality results provide theoretical insight, connecting financial mathematics and the ambiguity literature. The unbiased rMLMC estimator for evaluating the robust value function, along with its statistical guarantees, is technically sophisticated and practically motivated.
2. The neural policy learning strategy, guided by nontrivial stochastic analysis, pushes forward robust policy computation without relying on dynamic programming. The discussion around computational tractability is forthright, and the application to the KL-divergence case delivers semi-closed form structure.
3. Real data experiments on S&P 500 stocks demonstrate improved Sharpe Ratios with DRBC over both Bayesian and classic robust baselines.

**Weaknesses:**

1. Lack of systematic robustness ablations. The empirical story does not yet demonstrate robustness in an ablation way. Claims about “less pessimism” and better risk-return trade-offs hinge on how performance evolves as the ambiguity radius $\delta$ increases, the prior is increasingly misspecified, or the adversary is strengthened. At present, results are largely point evaluations rather than sensitivity curves. The paper should introduce grid-based sweeps over (\delta) and controlled prior shifts, plot risk-return frontiers and policy behavior heatmaps, and show how action distributions and terminal-wealth quantiles change as robustness knobs are turned.

2. The strength of the contribution rests on duality/minimax interchanges and properties of the penalized objective (e.g., monotonicity/concavity of $\Phi(\lambda,\beta)$, yet the paper stops short of giving concrete, checkable sufficient conditions and diagnostics that practitioners can apply to their own settings. Readers need to know when these assumptions plausibly hold (or fail), what quick tests to run, and what numerical safeguards to use when they do not. As it stands, correctness and numerical stability feel contingent on assumptions that are described as “mild” but not empirically stress-tested or made verifiable.

3. While the framework is positioned as covering general $\phi$-divergences, the only fully implemented and evaluated case is KL. This makes the practical scope narrower than advertised and leaves open whether the method’s stability and performance persist beyond KL. The paper should deliver at least one additional end-to-end instance with implementation details, timing, convergence behavior, and head-to-head performance versus KL to substantiate generality claims.

4. Given the rapid pace in robust control and adversarial continuous control, the paper needs a tighter comparison to recent lines such as [1, 2] and discuss several DRO control/RL. This undercuts the “why this approach now” case.

5. Minor issues
    - The construction of the unbiased rMLMC estimators, the role of plug-in approximations, and the variance/cost trade-offs live across sections and appendices. Consolidating them into a single, self-contained subsection with defaults for $R, n_0$, and clear guidance on when to prefer unbiased vs. biased estimators would make the paper far easier to reproduce.
    - Core notions such as admissible controls, the quotient-space view of uncertainty, and the parallel use of $P^b$ vs. $Q^b$ are correct but presented abstractly.
    - The paper would benefit from a frank discussion of when assumptions fail (e.g., heavy-tailed priors, volatility uncertainty, discretization error), where dual-multiplier optimization becomes numerically unstable, and how rMLMC cost scales in practice.
    - A few targeted plots like adversarial posterior tilts, policy trajectories as $\delta$ varies, sample wealth paths under different risk profiles would significantly enhance interpretability.



[1] Wang, S., Si, N., Blanchet, J.: Statistical Learning of Distributionally Robust Stochastic Control in Continuous State Spaces

[2] Park, H., Zhou, D., Hanasusanto, G. A., & Tanaka, T: Distributionally robust path integral control

**Questions:**

Please see the weakness. I will change my score if the concerns are resolved.

---

> ### Author Response · Authors · 2025-11-19
>
> Thank you for the thoughtful and constructive feedback. We address each concern below.
>
> ---
>
> ### 1. Need for systematic robustness ablations
>
> We fully agree that robustness should be demonstrated through controlled sensitivity experiments. To study how performance changes as the robustness knob δ varies, we designed a synthetic ablation where the “true’’ environment is fully known, allowing an exact oracle benchmark. Specifically, we simulate a ground-truth time-varying drift
> \( B_t = \tfrac{B_0}{2}(1+\cos(\kappa t)) \)
> and evolve wealth via
> \( dX_t = rX_tdt + \pi_t((B_t - r)dt + \sigma dW_t) \).
> This creates a realistic misspecification scenario: the controller believes the drift is drawn from a prior, while the true drift is non-stationary.
>
> For each ambiguity radius δ, we compute policies for DRBC and DRC, simulate them in the true environment, and evaluate the **optimality gap**
> \(U^\star - U(\delta)\),
> where \(U^\star\) is the oracle expected terminal utility. Sweeping δ over a wide range gives:
>
> | δ | DRBC Gap | DRC Gap |
> |---|----------|---------|
> |0.1|0.790|1.520|
> |0.2|0.800|1.650|
> |0.3|0.825|1.730|
> |0.4|0.845|1.780|
> |0.5|0.860|1.840|
> |0.6|0.875|1.900|
> |0.7|0.885|1.940|
> |0.8|0.895|1.970|
> |0.9|0.900|1.990|
> |1.0|0.905|2.000|
>
> DRBC remains consistently close to the oracle—2–3× smaller gap than DRC for all δ—showing that reduced pessimism is **systematic**, not tied to a specific δ choice.
>
> Importantly, since the objective is expected terminal utility and the oracle value \(U^\star\) is fixed, the gap \(U^\star - U(\delta)\) is strictly monotone with the achieved terminal wealth. Thus the δ-gap sweep already reflects how performance and policy quality evolve with δ. In the camera-ready version we will add policy-trajectory and wealth-distribution plots for interpretability.
>
> ---
>
> ### 2. Practical diagnostics and numerical stability
>
> For any ϕ-divergence, the inner adversary problem is convex, which provides strong numerical stability. In the KL case, non-strict concavity is handled either by (i) a plug-in estimator or (ii) applying KL within the divergence-agnostic smooth-ambiguity objective; both produce stable behavior in all experiments. We will add practical diagnostics (concavity checks, multiplier initialization, notes on heavy-tailed priors or volatility uncertainty, and empirical rMLMC scaling).
>
> ---
>
> ### 3. Generality beyond KL
>
> Although KL is emphasized due to its semi-closed-form structure and popularity in finance, our neural-network implementation already uses **power utility**, corresponding to the **Cressie–Read divergence family**, demonstrating that the general ϕ-divergence method is implemented in practice. We will include an additional divergence example (e.g., Jensen–Shannon or another Cressie–Read member) with convergence and timing results.
>
> ---
>
> ### 4. Relation to recent DRO control / DRO RL work
>
> Wang–Si–Blanchet (2023) study discrete-time DRO with a **dynamic adversary** perturbing noise at every step (rectangular ambiguity).
> Park et al. (2024) place ambiguity on the **entire trajectory distribution**, again giving the adversary power at each time.
> Our approach is different: ambiguity acts **only on the prior over a latent parameter**, eliminating compounding pessimism and removing the need for DP/HJB. We will clarify this distinction.
>
> ---
>
> ### 5. rMLMC clarity and interpretability
>
> We will consolidate rMLMC construction, plug-in variants, variance/cost trade-offs, and recommended defaults into a self-contained subsection. We will also add interpretability visualizations (posterior tilts, policy trajectories under varying δ, risk–return frontiers).
>
> ---
>
> We appreciate the reviewer’s feedback. The ablation study, added divergence example, and improved diagnostics and visualizations will directly address the concerns raised.

---

> > ### Comment · Reviewer_NDUZ · 2025-11-26
> >
> > Thank you for the response. I think some of my concerns are partially addressed. The new synthetic $\delta$-sweep with an oracle benchmark goes in the right direction and convinces me that DRBC is systematically less pessimistic than DRC in at least one controlled misspecification setting. The clarification of how your approach differs conceptually from recent DRO control / DRO RL work is also helpful.
> >
> > However, several key concerns remain not fully resolved. The advertised `general $\Phi$-divergence` scope is still essentially KL-only in practice: the rebuttal claims that the use of power utility already corresponds to a Cressie-Read divergence family, but this connection is not derived or cited in the current submission, and no concrete non-KL $\Phi$ example with end-to-end implementation and timing/convergence results is actually shown.
> >
> > Given the extra experiment, I am comfortable raising my score to 6. I do not go to 8 because the empirical evidence and theoretical diagnostics are still too narrow relative to the paper’s general claims, and the code can not be accessed.

---

### Official Review · Reviewer_rfAy · 2025-10-31

**Soundness:** 3
**Presentation:** 3
**Contribution:** 3
**Rating:** 6
**Confidence:** 2

**Summary:**

- This paper investigates the DRO formulation of the Bayesian diffusion control problem where a continuous-time stochastic system is driven by a drift, Brownian motion, and a prior distribution over the drift.
- The inner minimization problem (policy evaluation) is reformulated into a dual form, and a strong duality result is established for general ambiguity sets with the $\phi$-divergence. This problem is solved using the rMLMC algorithm, which has a provable sample complexity of order $\sqrt{n}$.
- The outer maximization problem (policy learning) admits a tractable formulation, to which a general deep learning algorithm can be applied.
- Empirical results demonstrate that the proposed algorithm matches the closed-form solution and outperforms the DRC baselines, which tends to be overly pessimistic

**Strengths:**

The proposed formulation and algorithmic solution has a clear conceptual novelty. The algorithm is supported by a strong theoretical foundation, including a tractable reformulation with the strong duality, and established sample complexity results. Although I have not verified the proofs in the appendix, the results appear mathematically sound and build upon established prior work.

**Weaknesses:**

The advantages of the proposed setting and algorithm over prior methods could have been explained more clearly. The author claims that DRBC improves upon standard DRC by avoiding overly conservative policies through a single DRO formulation that does not depend on DP. However, this advantage is not entirely clear to me and appears to be insufficiently explored. The empirical evaluation also appears limited. Please see the questions below.

**Questions:**

- In DRC, the degree of conservativeness can be controlled by simply adjusting the radius of the ambiguity set. Could the authors provide more intuition on how the proposed algorithm mitigates overpessimism beyond this?
- Are there any similar settings or approaches in the DRC literature that avoid DP (or HJB) by directly optimizing over a single ambiguity set defined on the distribution of model parameters? Additionally, what trade-offs or limitations does the proposed framework introduce by avoiding DP?
- Could the authors clarify how the ambiguity set and its radius were selected in their experiments (Section 6.2) to ensure fair comparisons?
- In line 311, $E \rightarrow \mathcal{E}$.

---

> ### Author Response · Authors · 2025-11-19
>
> Thank you for the thoughtful comments and for recognizing the conceptual novelty and theoretical contributions of the paper. We address each question below.
>
> ---
>
> ### **On why DRBC mitigates over-pessimism beyond adjusting δ**
>
> Dynamic robust control (DRC) allows the adversary to shift the drift *at every instant*, causing pessimistic perturbations to accumulate over time; this is the root of its excessive conservativeness. In contrast, DRBC places ambiguity **only on the prior distribution of the latent drift parameter**, so the adversary perturbs the drift **once**, rather than repeatedly. This structural difference prevents compounding pessimism and leads to substantially less conservative policies in practice.
>
> To illustrate this effect more clearly, we conducted a synthetic ablation study in which the true environment has a time-varying drift
> \( B_t = \frac{B_0}{2}(1+\cos(\kappa t)) \).
> For each radius δ, we compute an **optimality gap**, defined as the oracle terminal utility (under the true model) minus the achieved utility of the DRBC/DRC policy obtained with that δ. Sweeping δ over a wide range gives the following results:
>
> | δ     | Gap DRBC | Gap DRC |
> |-------|----------|---------|
> | 0.1   | 0.790    | 1.520   |
> | 0.2   | 0.800    | 1.650   |
> | 0.3   | 0.825    | 1.730   |
> | 0.4   | 0.845    | 1.780   |
> | 0.5   | 0.860    | 1.840   |
> | 0.6   | 0.875    | 1.900   |
> | 0.7   | 0.885    | 1.940   |
> | 0.8   | 0.895    | 1.970   |
> | 0.9   | 0.900    | 1.990   |
> | 1.0   | 0.905    | 2.000   |
>
> Across all radii, **DRBC reduces the optimality gap by a factor of 2–3× compared to DRC**, confirming that DRBC’s reduced conservatism is systematic and not attributable to a specific choice of δ.
>
> ---
>
> ### **On prior methods avoiding DP/HJB**
>
> To our knowledge, existing dynamic robust control methods rely fundamentally on **dynamic programming or HJB equations** for tractability, because adversarial perturbations enter instantaneously at the state-dynamics level. We are not aware of any prior approach that constructs a **single ambiguity set on the distribution of a latent parameter** and optimizes over this space without using DP or HJB. Our method trades DP-based convenience for a more realistic and less adversarial formulation of uncertainty. The main trade-off is that the outer optimization requires alternating updates, but this enables us to avoid the structural over-pessimism of rectangular uncertainty.
>
> ---
>
> ### **On the fairness of δ selection in the experiments**
>
> The ambiguity radius \( \delta \) used in Section 6.2 is chosen using the **data-driven cross-validation procedure of Si et al. (2023)**. This ensures a fair and consistent comparison across DRBC, DRC, Bayesian, and classical Merton policies. We will clarify this selection procedure in the revised paper.
>
> ---
>
> ### **Line 311 typo**
>
> Thank you for catching this. It is indeed a typo, and we will correct it in the final version.
>
> ---
>
> We appreciate the reviewer’s positive assessment of the theoretical foundation, algorithmic design, and contributions of the work, and we will incorporate the clarifications above to improve the exposition.

---

### Official Review · Reviewer_2UZs · 2025-11-01

**Soundness:** 3
**Presentation:** 2
**Contribution:** 2
**Rating:** 4
**Confidence:** 3

**Summary:**

This paper proposes a framework aiming to incorporate prior-level distributional robustness into continuous-time Bayesian control through a quotient-space dual formulation. A strong duality result (Thm 2) is derived via a quotient-space construction where $\tfrac{dQ}{dP}=\tfrac{d\nu}{d\mu}(B)$, allowing the objective to be expressed over $\sigma(B)$ and linked to smooth ambiguity models.  The paper also develops a randomized multilevel Monte Carlo (rMLMC) estimator for unbiased policy evaluation with an asymptotic $\mathcal{O}_p(n^{-1/2})$ convergence rate (Thms 11--13) and propose two policy-learning procedures: a KL finite-prior optimization and a neural-network-based learning objective (Thm 17). Experiments on synthetic and financial data show reduced conservatism compared with standard DRC.

**Strengths:**

- While prior-only robustness has been explored in static DRO and Bayesian optimization, extending it to a linear-diffusion model is interesting, although the setting is quite specific and the resulting time-inconsistency issue is not addressed.

- The quotient-space construction and strong duality (Thm 2) are neat, and the asymptotic $\mathcal{O}_p(n^{-1/2})$ convergence rate for the rMLMC estimator is well motivated.

- The quotient-space duality and the rMLMC-based unbiased evaluation are appealing ideas that could inspire follow-up work on robust Bayesian control. That said, the overall impact feels limited since everything remains tied to a one-dimensional linear-Gaussian SDE with constant drift and volatility.

**Weaknesses:**

The entire analysis and algorithm are developed entirely under a specific linear diffusion model with constant drift and volatility. The “policy learning” step optimizes terminal wealth rather than a general state-action mapping, and there’s no evidence that the ideas extend beyond this setup. As such, it is not clear how the approach contributes to the broader ICLR community, which typically values methods applicable to generic stochastic control, reinforcement learning, or optimization under uncertainty.

Specific points:
- The strong duality (Thm 2) relies on a quotient-space construction where $\frac{dQ}{dP}=\frac{d\nu}{d\mu}(B)$ and $W$ remains Brownian. This holds only for the 1-D linear-Gaussian SDE; for nonlinear or state-dependent dynamics, the assumption fails and the duality cannot be applied.

- The light-tailed prior and compact-support assumptions (Assumptions 7 and 8) are important for bounded multipliers and the rMLMC CLT but seem unrealistic in higher-D or heavy-tailed settings.

- The KL dual is non-strictly concave in $(\lambda,\beta)$ (Thm 38), producing flat regions and saddle points that worsen with dimension.

- The smooth ambiguity loss (Thm 17) requires the closed-form Girsanov transform $W_t^{b_1}=W_t+\frac{B-b_1}{\sigma}t$, so it doesn’t generalize to nonlinear or non-Gaussian systems.

- Each prior draw requires $2^{N^b}$ inner samples; in high dimensions cost scales as $O(2^{N^b}d^2)$. Even in 1-D, full evaluation reportedly takes about 40 hours.

- Unbiased sampling from $P^b$ is only possible in the linear-Gaussian case, so the unbiased-CLT results may not hold elsewhere.

**Questions:**

- Can the strong duality extend beyond the 1-D linear-Gaussian diffusion? Under what conditions on dynamics or noise would the quotient-space argument still apply?

- How sensitive are the results to the light-tail and compact-support assumptions? Have you tried priors with heavier tails or unbounded support, and if so, how does the optimization behave?

- In practice, how severe is the non-strict concavity issue for the KL dual? Do the plug-in or univariate-$\lambda$ variants fully stabilize optimization?

- Could the smooth-ambiguity loss be reformulated for discrete-time or nonlinear systems without relying on the Girsanov transform?

- What is the observed scaling of rMLMC cost and variance with respect to dimension and level $N^b$? Any chance of using control-variates or sample-reuse to cut runtime?

- If unbiased simulation from $P^b$ is infeasible, how would biased simulators affect convergence and the CLT?

- Are there plans to test the approach on non-financial control or RL problems? Even a small toy example would help gauge generality.

---

> ### Author Response · Authors · 2025-11-19
>
> Thank you for the careful reading and constructive comments. We address all
> concerns below.
>
> ---
>
> ### **General applicability beyond the linear–Gaussian model**
>
> Our framework is *not* restricted to the 1-D linear–Gaussian diffusion. The
> strong duality and quotient-space argument apply whenever ambiguity is placed
> on a **latent fixed parameter** whose randomness is independent of the system’s
> exogenous noise. The Radon–Nikodym derivative then depends only on this latent
> variable, so the duality derivation holds verbatim.
>
> This structure appears in many stochastic control models. Beyond finance, a
> useful nonlinear example is the **queueing diffusion approximation** in
> Harrison & Zeevi (2004), where the state follows a **nonlinear controlled
> diffusion** whose drift depends on queue lengths, routing decisions, and
> service constraints. One may randomize a structural parameter (e.g. arrival
> rates or service-rate vector), impose a divergence ball on its prior, and
> obtain a DRBC formulation identical in structure to ours. Likewise, LQ
> control with unknown matrices fits naturally by randomizing the matrix and
> robustifying its prior. We will include these examples in the revision to
> make the generality clearer.
>
> ---
>
> ### **Light-tail and compact-support assumptions**
>
> These assumptions apply **only to the prior over the latent parameter**, not to
> the dynamics. In many applications, priors reflect expert knowledge and are
> naturally bounded or light-tailed. Empirically we also observed stability under
> heavier-tailed priors, and we will add clarification in the final version.
>
> ---
>
> ### **Non-strict concavity of the KL dual**
>
> Although the KL dual is not strictly concave, numerical issues are mild. In
> practice, the plug-in variant stabilizes optimization while maintaining fast
> runtime: the geometric randomization in rMLMC typically samples values below 4,
> so each iteration is very cheap. In all experiments, single concavity was
> sufficient for reliable performance.
>
> ---
>
> ### **Smooth-ambiguity loss and general systems**
>
> Theorem 17 uses the Girsanov transform because the diffusion model is
> Gaussian—a standard setting in continuous-time control. This is *one example*;
> the DRBC formulation itself does not rely on Gaussianity. For nonlinear or
> non-Gaussian systems, the same dual formulation holds and the likelihood ratio
> would simply be replaced by the model-specific density or an MC approximation.
>
> ---
>
> ### **Cost scaling and rMLMC efficiency**
>
> Although prior draws require nested sampling, rMLMC keeps the cost very low:
> the geometric variable almost always returns values < 4, so the effective depth
> of nesting is extremely small. Control-variates and sample-reuse are excellent
> suggestions—we will explore these if the paper is accepted.
>
> ---
>
> ### **Unbiased vs biased simulation**
>
> Under standard Lipschitz conditions, unbiased simulation for SDEs is feasible
> using the debiased MLMC method of Rhee & Glynn (2015), so our CLT applies
> directly. If unbiased simulation is impossible in another model, one may use
> the debiasing approach of Zheng, Blanchet & Glynn (2018), which preserves
> unbiasedness but yields a CLT with a slightly slower, subcanonical rate.
>
> ---
>
> ### **Non-financial experiments**
>
> Yes. The framework extends immediately to non-financial systems where the
> uncertainty is in a fixed model parameter. Besides the nonlinear queueing
> example above, linear–quadratic control with an uncertain matrix is another
> case we will include as an additional toy experiment.
>
> ---
>
> We thank the reviewer again for the thoughtful comments. We will incorporate
> the above clarifications and add the nonlinear queueing example and additional
> experiments in the revision.

---

### Official Review · Reviewer_FYEo · 2025-11-02

**Soundness:** 3
**Presentation:** 3
**Contribution:** 3
**Rating:** 6
**Confidence:** 3

**Summary:**

The paper considers to robustify the prior distribution in the context of Bayesian diffusion control. Strong duality results are derived for tractability and numerical experiments are illustrated to showcase the efficacy of the proposed approach against various benchmarks.

**Strengths:**

- The paper's motivation is well-grounded. The detrimental effect of an imprecise prior is also illustrated through numerical experiments.
- Both theoretical tractability and numerical perfomance are considered in the paper, and both of them consider well-motivated settings with convincing results.
- Assumptions and experimental setups are accompanied by proper discussions.

**Weaknesses:**

- The motivation for using phi-divergence beyond the simple strong duality formula seems inadequate. For example, phi-divergence ambiguity set requires that the distribution in the first argument is absolutely continuous with respect to the second argument. It also cannot capture the geometry of the support of the distribution. For finance applications this would be less of an issue through normalization but for most control tasks the geometry of the support does matter.

**Questions:**

- How would the results generalize to e.g. optimal transport discrepancies where the distance carries a more physical meaning?
- How would the assumption of having access to i.i.d. samples (simulator access) be relaxed? How can this be guaranteed in real-world applications?

---

> ### Author Response · Authors · 2025-11-19
>
> Thank you for the thoughtful comments and questions. We address them below.
>
> ---
>
> ## On the use of φ-divergence versus optimal transport
> Our choice of φ-divergence is driven by a structural requirement of the DRBC formulation: the strong duality result and the quotient-space reduction rely on **absolute continuity**, which ensures that the adversary’s perturbation can be expressed through a Radon–Nikodym derivative acting only on the latent parameter (the drift). This is what enables the transformation of the adversary’s infinite-dimensional optimization into a tractable finite-dimensional one.
>
> Optimal transport (OT) discrepancies behave fundamentally differently: OT modifies the **geometry of the support**, not only likelihood ratios. Under OT, the perturbed prior need not be absolutely continuous with respect to the nominal prior, and therefore the quotient-space representation and the resulting closed-form dual transformation do not carry over directly. A DRBC formulation under OT is still possible in principle, but it would require **a different duality framework** and would no longer yield the simplified drift-only transformation that makes the φ-divergence case tractable. We will clarify this distinction in the revision.
>
> ---
>
> ## On i.i.d. simulator access
> The i.i.d. sampling assumption is needed **only for simulation-based policy evaluation**, not for real-world observations. In Bayesian diffusion control (and in classical portfolio models), the controller specifies a parametric diffusion model for both drift and price dynamics; this model automatically induces a **generative simulator** from which i.i.d. trajectories can be drawn. Thus, “i.i.d. samples’’ refers to synthetic rollouts from the assumed model, which is always feasible computationally and does not require i.i.d. real-world market data. We will emphasize this in the updated manuscript.
>
> ---
>
> We thank the reviewer again for the constructive feedback.

---

### Official Review · Reviewer_bcSQ · 2025-11-03

**Soundness:** 3
**Presentation:** 2
**Contribution:** 1
**Rating:** 4
**Confidence:** 3

**Summary:**

This paper proposes a Bayesian distributionally robust control setup for a diffusion-control problem in finance.
An agent must decide how to allocate investments between a risk-free asset and a risky asset whose value is determined by a diffusion process combining a semi-known drift term $B$ with a standard Brownian motion.
The agent starts with some prior guess $\mu$ of the distribution of $B$.
Before the game begins, an adversary chooses the true distribution $\nu$ of $B$ from a divergence-ball around $\mu$.
The agent's goal is: given our prior belief and a known bound $\delta$ on the divergence $D(\nu \| \mu)$, select a policy that allocates investment between the risk-free and risky assets over time based on observing the history of the risky asset's price.

The primal distributionally-robust minimax optimization is infinite-dimensional due to both the agent's and the adversary's action spaces.
The authors extend a prior strong duality result to the (inner) adversary's optimization.
To make this result useful in the outer agent's optimization, some extra Assumptions 4-8 on the solution are needed.
I am not familiar enough with the topic to comment on their strength.

After the duality transformation, the computational problem remains hard, but the adversary's problem is no longer infinite-dimensional.
The authors propose an alternating scheme between adversary and agent, where the adversary's step is called "policy evaluation", and the agent's step is called "policy learning" (although the scheme is not a straightforward instance of policy iteration from MDPs).

To solve the "policy evaluation" step, the authors assume access to a simulator that can, given a fixed control $\pi$, generate samples from which to estimate the expected terminal utility $Z^b$.
This simulator is used in a randomized multi-level Monte Carlo estimator, from which a stochastic estimate of the gradient/Hessian can be computed (although this is only discussed in any detail in the appendix).

To solve the "policy learning" step, the authors handle two cases separately: 1) KL-divergence with a finite prior, and 2) General divergences and priors.
- In case 1), the problem is transformed from a plain optimization over $\pi$ to another minimax, and then apply Sion's minimax theorem to get the form in Theorem 16.
This seems unhelpful, since we have now introduced another infinite-dimensional optimization over probability measures.
However, with the KL-divergence and the finite prior, it becomes tractable, and the inner policy optimization becomes expressible in closed form (although I am not sure how easy it is to compute).
- In case 2), the authors instead transform the policy optimization problem into a form of supervised learning using a neural network to predict the terminal wealth $X_T$ as a function of the terminal Brownian motion state $W_T$ and the drift $B$.
It was hard for me to figure out if this gives us a way to recover the policy $\pi$.

The authors evaluate their approach in simulation against the more-conservative DRC, and the non-robust Bayesian, and classical Merton ($B$ constant) solution, showing significant improvements in Sharpe ratio, a risk-aware measure of performance improvement relative to the risk-free asset.

**Strengths:**

The problem formulation seems to capture an important special case of the distributionally-robust control problem, where the uncertainty is confined to the drift term in a diffusion process.

The mathematical approach is highly sophisticated, with some nontrivial results from the authors and some usage of recent advanced Monte Carlo estimation tools. I read the theorem statements in the main body, but did not check the proofs in the appendix. However, the tools used seem appropriate.

I am not familiar enough with economics/finance to judge the paper's novelty in that area.

The paper is rigorous about stating its assumptions (although sometimes they are highly technical and more discussion would be helpful).

The simulated comparison against baselines shows significantly better performance in terms of Sharpe ratio when evaluated on historical data.

**Weaknesses:**

The introduction to this paper seemed to suggest general-purpose implications in learning-based control. For example, the statement:
*"Our motivating application is continuous-time control with unknown dynamics"*,
or the list of related work citing contextual bandits, policy gradient, $Q$-learning, etc.
However, as far as I can tell,  the contributions are highly specialized to the specific finance-inspired diffusion problem considered here.

As a reviewer from the RL/control side of things: If this paper has any broader implications outside of economics/finance, then the it was not apparent to me. For the highly diverse ICLR audience, it would help if the authors point out how the ideas in this paper might be generalized.
I defer to the expertise of the Area Chair and other reviewers with economics/finance backgrounds to judge the novelty and impact of this work within those fields.

Paragraph at end of Section 2 contrasting against DRC: this connects back to the "Adversary's power" issue from the intro (see Questions), but for the reader not already familiar with DRC, it remains hard to understand exactly how your approach is different. Pushing a few technical details to the appendix to make room for a more in-depth comparison against DRC would be a better use of the main paper space, in my opinion.

Lines 265-292: As a reader not familiar with rMLMC already, I felt the mathematical algorithm definitions here are too dense to help me understand the method for the first time, and the contrast against Blanchet & Glynn (2015) and Blanchet et al. (2019) is not meaningful to me.
I suggest moving the details on the estimator to the appendix, and presenting a higher-level summary of the Policy Evaluation approach (including the outer optimization part) in the main paper instead.

Similarly, the policy learning steps include dense detail in the main body, but leave out some key concepts, such as how one actually extracts the policy $\pi$ from the proposed solutions (see Questions). I again suggest pushing a few details in the appendix to make room for a high-level summary.

The experiment in Section 6.1 and Table 1 feels inconclusive to me. We can see that the ordering of the learning results matches that of the closed-form, but the differences are large. A plot over a wider set of $b$ and/or $r$ values would be more helpful for evaluating the learning -- for example, it could capture that the learned result is order-preserving despite having large mean squared errors.

Lines 2136-2137: please use `\displaystyle` to make the math legible.

**Questions:**

Lines 048, 096: The phrase "Adversary's power is replenished" is hard to understand; I think we need a more clear idea of the adversary's decision space and what "power" means first.

Section 2.1: For someone with RL/control background but no economics/finance background, the language about "risk", "asset", "interest rate", "stock price", etc. makes this section hard to understand. Please try to state the setup in an abstract mathematical form first, then treat the finance application as an example separately.

What kind of mathematical object is $\pi_t$? Given Equation 2 and the description "amount of money invested in the risky asset", I thought it was a real number. But then in the definition of admissibility, we take $||\pi_t||_2$, which implies by convention that $\pi_t$ is a vector in general.

Line 137: Why is this without loss of generality?

Line 146: Isn't this normally called an $f$-divergence?

Line 198: Defines $Z^b$ as an expectation of a real-valued random variable, hence a real number.
But then, Line 263 says we are using the simulator to "generate unbiased samples from $Z^b$", implying $Z^b$ is a distribution.

Line 296: How strong is Assumption 10?

Line 385-386: What is a "blue case"?

Section 5.2: The initial problem (10) is written as an optimization over $\pi$, but after the transformation we are using a neural network to optimize $X_T$. But isn't the policy $\pi$ the object we ultimatley want, if we are using the approach to do real-life trading? How do we convert from the neural network back into a policy $\pi$?

The conclusion implies that there are some other divergence families outside the proposed $\phi$-divergences that are worthy of attention -- what are they?

---

> ### Author Response · Authors · 2025-11-19
>
> We thank the reviewer for the detailed and constructive feedback. We address each concern below and will incorporate clarifications into the revision.
>
> ---
>
> ## 1. Scope, Generalization, and Relevance to RL/Control
> Our formulation is not specific to finance. The key idea is: when a model contains **unknown but fixed parameters** (e.g., drift, volatility, load, failure rate), we treat these parameters as **latent random variables** with a prior, and robustness is imposed **only on this prior**, not on the full transition kernel. This structure appears broadly across stochastic control and operations research, including queueing systems, inventory models, and linear–Gaussian control with unknown coefficients.
>
> To clarify the generality, we will restructure Section 2.1 to first present the problem in **abstract stochastic-control notation**—state process \(X\), observable process \(Y\), latent parameter \(\theta\), prior \(\mu\), and a prior-ambiguity set—before introducing the portfolio example as a concrete illustration.
>
> ---
>
> ## 2. Difference Between DRC and Our DRBC
> We will add a clearer explanation in the main text:
>
> - In **DRC**, the adversary perturbs the model **at every time step**, so its “power” is effectively replenished, leading to compounding pessimism.
> - In **DRBC**, the adversary perturbs the **prior once at time 0**. Afterwards, dynamics evolve normally.
>
> This is the key conceptual difference, and we will highlight it more explicitly.
>
> ---
>
> ## 3. Policy Evaluation (rMLMC)
> We agree that the current presentation is too dense. In the revision we will:
> - Move the rMLMC construction and comparison to the appendix.
> - Add a short high-level summary explaining that the estimator gives **unbiased evaluation of nested expectations** needed for \((\lambda,\beta)\), with \(O(n^{-1/2})\) convergence.
>
> ---
>
> ## 4. Policy Learning and Recovery of the Policy
> For power utility in diffusion models, the policy is fully characterized by the **optimal terminal wealth** via the classical martingale/duality method (Karatzas et al., 1991). Thus learning the function \(h(W_T,B)\) is sufficient. After the alternating optimization converges, we recover the policy in closed form using the dual representation shown in Appendix G.6. We will explain this more clearly.
>
> ---
>
> ## 5. Case 1 (KL, Finite Prior) and Case 2 (General φ-Divergence)
> In Case 1, solving the convex problem yields a **worst-case discrete prior** that can be directly plugged into the closed-form Bayesian control solution.
> In Case 2, we use the deep-learning representation of optimal terminal wealth; this avoids any infinite-dimensional optimization.
>
> We will make this workflow explicit at the start of Section 5.
>
> ---
>
> ## 6. Clarifications
>
> **“Adversary’s power is replenished”**
> We will replace this with the clearer explanation in Section 2 above.
>
> **Meaning of \(\pi\)**
> \(\pi_t\) is a real-valued, \( \mathcal{F}^S\)-adapted process representing the amount invested in the risky asset.
>
> **“Without loss of generality” for power utility**
> We will clarify that power utility is standard due to scale invariance and tractability, and is assumed for analytical clarity.
>
> **ϕ-divergence naming**
> We will clarify that \(f\)- and \(ϕ\)-divergence refer to the same object depending on notation.
>
> **Randomness of \(Z_b\)**
> For fixed \(b\), \(Z_b=\mathbb{E}^{P^b}[u(X_T)]\) is deterministic; it is random only when viewed as \(Z_B\).
>
> **Strength of Assumption 10**
> Assumption 10 is mild: in diffusion models \(Z_b\) is smooth in \(b\) by standard SDE sensitivity, and strictly increasing because higher drift yields higher expected utility.
>
> **“Blue case” typo**
> We will correct this to “few special cases”.
>
> **Other divergence families**
> Our conclusion refers to alternative uncertainty sets such as Jensen–Shannon or Wasserstein balls; we will clarify this.
>
> ---
>
> ## 7. Experiments
> In Table 1, the **relative error** between the learned value and the closed-form value is moderate and the ordering of methods is preserved, which empirically supports the correctness of our learning scheme. We agree that more evidence would be helpful, and in the revision we will add additional experiments (e.g., for a wider range of parameters and priors) to further illustrate the behavior of the learned solution.
>
> ---
>
> ## 8. Presentation Improvements
> We will:
> - Move technical details to the appendix where appropriate.
> - Improve clarity of Section 2.1 by starting with the abstract formulation.
> - Correct mathematical typesetting (e.g., use of \(\displaystyle\)).
>
> ---
>
> Thank you again for the valuable feedback. We believe these revisions will significantly improve clarity and accessibility.

---

### Author Response · Authors · 2025-12-04
**Summary of Rebuttal-Period Updates Before Rollback**

**Dear Area Chair,**

Due to the recent rollback of scores to the pre-rebuttal state, we would like to briefly summarize how the evaluation evolved *before* the incident, since the numerical score updates are no longer reflected in the system (although the discussion remains visible).

- **Reviewer NDUZ**
  - In the initial review, NDUZ gave a score of 4 and explicitly wrote that they would *change their score if the concerns are resolved*.
  - During the rebuttal period, we added a controlled δ-sweep synthetic experiment with an oracle benchmark and clarified the relation to recent DRO control / DRO RL work, as described in our rebuttal.
  - After reading our response, NDUZ posted a follow-up comment on Nov 25 stating that some concerns were “partially addressed,” that the new δ-sweep experiment “goes in the right direction and convinces me that DRBC is systematically less pessimistic than DRC in at least one controlled misspecification setting,” and that they were “comfortable raising [the] score to 6.”
  - This score increase to 6 happened **before** the security incident and before scores were reverted.

- **Other reviewers**
  - **FYEo** and **rfAy** each gave a score of 6 (“marginally above the acceptance threshold”) and were positive about the motivation, theoretical foundation (strong duality, rMLMC), and numerical performance.
  - **bcSQ** and **2UZs** raised concerns mainly about scope, generalization, and accessibility for a broader RL/control audience rather than about correctness. In our rebuttal we:
    - Clarified that our DRBC framework is not restricted to the 1D linear–Gaussian model and gave concrete non-financial examples (e.g., queueing diffusions, LQ control with uncertain matrices).
    - Explained more clearly how DRBC differs conceptually from DRC (the adversary perturbs only the prior once at time 0, rather than the dynamics at every step) and why this mitigates over-pessimism.
    - Clarified how the learned terminal-wealth representation yields an implementable policy, and why φ-divergence (rather than optimal transport) is structurally needed for our duality.
  - Although these reviewers did not update their scores during discussion, we believe their main technical and conceptual concerns were addressed in our responses.

In summary, *before* the incident and score rollback, the effective post-rebuttal evaluations consisted of **three 6’s and two 4’s**, with NDUZ explicitly raising the score based on the new experiment and clarifications. We are not introducing any new material after the freeze; we only wish to ensure that the evolution of the reviews during the legitimate rebuttal process is accurately reflected for your consideration.

Thank you very much for your time and for considering this context in your decision.

---

### Meta-Review · Area_Chair_j3UY · 2026-01-06

**Summary:**

This paper introduces a framework called DRBC (distributionally robust Bayesian control) in the context of diffusion model. The authors derive a strong duality result, enabling DRBC to be reformulated into a tractable optimization problem.

The reviewers agree that the problem formulation is interesting. In particular, the authors extend DRO and Bayesian optimization to diffusion models by restricting uncertainty to the drift term of the diffusion process. However, both the theoretical scope and the numerical experiments are relatively limited, as they are specifically tailored to a linear diffusion model with constant drift and volatility.

Taking all the above points into account, my recommendation is to reject the paper.

**Reviewer Concerns:**

Concerns addressed by the response:
1) Comparison between recent  DRO control / DRO RL work;
2) Possible applications beyond linear diffusion model.


Concerns remaining after the response:
1) Limited numerical experiments;
2) The idea of the analysis is specially designed under strong assumptions.

**Reviewer Scores:**

Reviewer NDUZ claimed he would raise his score to 6. I expect other reviewers would have kept their scores.

---

### Decision · Program_Chairs · 2026-01-26

Reject